PREPARED FOR SUBMISSION TO JHEP

# Non-invertible Symmetries and Higher Representation Theory I

**Thomas Bartsch, Mathew Bullimore, Andrea E. V. Ferrari, Jamie Pearson**

*Department of Mathematical Sciences, Durham University,*
*Lower Mountjoy, Stockton Road, Durham, DH1 3LE, United Kingdom*

ABSTRACT: The purpose of this paper is to investigate the global categorical symmetries that arise when gauging finite higher groups in three or more dimensions. The motivation is to provide a common perspective on constructions of non-invertible global symmetries in higher dimensions and a precise description of the associated symmetry categories. This paper focusses on gauging finite groups and split 2-groups in three dimensions. In addition to topological Wilson lines, we show that this generates a rich spectrum of topological surface defects labelled by 2-representations and explain their connection to condensation defects for Wilson lines. We derive various properties of the topological defects and show that the associated symmetry category is the fusion 2-category of 2-representations. This allows us to determine the full symmetry categories of certain gauge theories with disconnected gauge groups. A subsequent paper will examine gauging more general higher groups in higher dimensions.

# 1   Introduction

There has been exciting recent progress in understanding the existence and implications of non-invertible global categorical symmetries [1–13]. A general mechanism to produce non-invertible symmetries in dimension $D = 2$ is to gauge a finite non-abelian symmetry group [14–17]. The purpose of this work is expand this construction to incorporate gauging a finite $n$-group symmetry in dimension $D > 2$ with $n = 1, \ldots, D - 1$.

This present paper will focus on gauging finite groups and split 2-groups in dimension $D > 2$, while subsequent work will explore more general finite 2-groups and higher groups in $D > 2$.

## 1.1   Motivation

A first motivation for this paper is to develop a systematic approach to constructing finite non-invertible symmetries in dimension $D > 2$, which incorporates the range of perspectives that have appeared in the literature and sheds some light on the relationship between different constructions and common structures.

A second motivation is to explore the mathematical structure of symmetries that result from gauging finite higher groups. In dimension $D$, the symmetry structure of a quantum field theory is expected to be captured by a fusion $(D - 1)$-category, which encodes the properties of extended topological operators in dimensions $p = 0, \ldots, D - 1$.

The proposal is that the symmetry category arising from gauging a finite higher group in $D$ dimensions is the category of $(D - 1)$-representations of that higher group. This mathematical structure encodes the properties of extended topological operators which are higher-dimensional analogues of topological Wilson lines. This generalises the well-known

result for gauging a finite symmetry group in $D = 2$ [14–17] and has interesting structure when $D > 2$ even when gauging an ordinary finite group.

## 1.2 Summary of Results

In dimension $D = 2$, gauging a finite symmetry group $G$ results in topological Wilson lines transforming in representations of $G$. This generates a $\mathsf{Rep}(G)$ fusion category symmetry, which has non-invertible simple objects if $G$ is non-abelian.

In $D > 2$, gauging an ordinary finite symmetry group again results in topological Wilson lines transforming in representations of $G$. However, there are also higher-dimensional topological defects arising from combinations of inserting SPT phases on submanifolds and condensation defects for the topological Wilson lines[1].

The idea is that the full spectrum of topological defects of dimensions $q = 0, \ldots, D - 1$ that arises when gauging a finite group $G$ is captured by the higher representation theory of $G$. We propose that the full symmetry category is the $(D-1)$-fusion category of $(D-1)$-representations of $G$. A large portion of this paper is dedicated to explaining and checking this proposal in dimension $D = 3$.

### 1.2.1 Groups

Let us consider a theory $\mathcal{T}$ with anomaly free finite group symmetry $G$. The strategy utilises the construction of correlation functions in $\mathcal{T}/G$ by summing over networks of symmetry defects inserted in correlation functions in $\mathcal{T}$. The topological defects in $\mathcal{T}/G$ are then defined operationally as topological defects in $\mathcal{T}$ together with instructions for how symmetry defects may end on them consistently. Spelling out this construction systematically leads to a generalisation of the construction of [14–17] to dimension $D > 2$.

In dimension $D = 3$, we will show that the simple topological surfaces are labelled by the following data

1. A $G$-orbit $O$.

2. A class $c \in H^2(G, U(1)^O)$.

They coincide with irreducible 2-representations of $G$. We will also compute the fusion, 1-morphisms, composition of 1-morphisms and fusion of 1-morphisms of simple objects. This provides an identification of topological surfaces with $|O| > 1$ with partial condensation defects for topological Wilson lines. The results are consistent with the symmetry category $2\mathsf{Rep}(G)$ of 2-representations of $G$, whose structure has been studied extensively in the mathematical literature [18–21].

We will also present an equivalent formulation where simple topological surfaces are labelled by instead by the following data:

1. A subgroup $H \subset G$.

2. An SPT phase $c \in H^2(H, U(1))$.

---

[1]From a mathematical perspective these are all condensations.

This provides a more direct physical construction of the simple topological surfaces in $\mathcal{T}/G$ in which the gauge symmetry is broken to $H \subset G$ and SPT phase is inserted for the unbroken gauge symmetry. This connects with and to some extent generalises the perspective on condensation defects in [4].

### 1.2.2  2-Groups

We extend this construction further to gauging a finite 2-group symmetry in $D = 3$. In this paper, we focus on split 2-groups with vanishing Postnikov class, which are specified by a 0-form symmetry group $H$, an abelian 1-form symmetry group $A$, and an action of the former on the latter by automorphisms. We write such a split 2-group as

$$G = A[1] \rtimes H \tag{1.1}$$

by analogy with a semi-direct product.

We first elucidate the full symmetry category $2\mathsf{Vec}(G)$ of a theory $\mathcal{T}$ with 2-group symmetry $G$, including the contribution of condensation defects for the 1-form symmetry. We then compute symmetry category of $\mathcal{T}/G$ by generalising the gauging procedure described above to show that it coincides with the fusion 2-category $2\mathsf{Rep}(G)$ of 2-representations of the 2-group $G$.

The simple topological surfaces are now labelled by the following data

1. A $H$-orbit $O$.

2. A class $c \in H^2(G, U(1)^O)$.

3. A collection of characters $\chi_j : A \to U(1)$ indexed by $j \in O$, satisfying

$$\chi_j(a^h) = \chi_{\sigma_h(j)}(a)$$

for all elements $a \in A$ and $h \in H$.

Here, $a^h$ denotes the action of $H$ on $A$ and $\sigma : H \to \mathrm{Aut}(O)$ denotes the permutation representation arising from the $H$-action on $O$. We compute the fusion, 1-morphisms, composition of 1-morphisms and fusion of 1-morphisms and show that they coincide with those in $2\mathsf{Rep}(G)$. The topological surfaces with $|O| > 1$ and characters $\chi_j : A \to U(1)$ that do not form a single $H$-orbit in $\widehat{A}$ involve at least a partial condensation of topological Wilson lines.

We again present an equivalent formulation in which simple topological surfaces are labelled by the following data:

1. A subgroup $K \subset H$.

2. A lass $c \in H^2(K, U(1))$.

3. A $K$-invariant character $\chi : A \to U(1)$.

This again provides a more direct physical interpretation of simple topological surfaces in $\mathcal{T}/G$ generalising the group case. From a mathematical perspective, it also provides a new description of simple objects in $2\mathsf{Rep}(G)$ for a split 2-group.

The construction in this paper is closely related but distinct from the gauging process in [5], which did not in the first instance output the SPT phases and condensation defects. The latter arose instead from an additional step of passing from local to global fusion[2]. Here, all of the simple topological surfaces arise uniformly from the construction and there is only one type of fusion. The distinction may be seen in the classification above by setting $c = 0$ and restricting to collections $\chi_j : A \to U(1)$ forming a single $H$-orbit. It would be interesting to clarify the precise relation.

### 1.2.3   Gauge Theories

The above results have applications to non-invertible categorical symmetries of gauge theories with disconnected gauge groups. We first consider a pure gauge theory $\mathcal{T}$ in $D = 3$ with a compact, connected, simple, simply connected gauge group $\mathbf{G}$, such as $\mathrm{Spin}(2N)$. This has a split-2 group global symmetry

$$Z(\mathbf{G})[1] \rtimes \mathrm{Out}(\mathbf{G}) \tag{1.2}$$

where $\mathrm{Out}(\mathbf{G})$ is the 0-form symmetry of outer automorphisms and $Z(\mathbf{G})$ is the electric 1-form center symmetry.

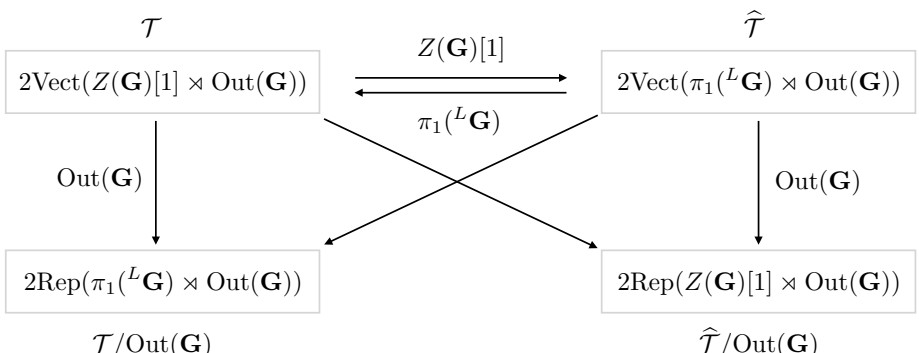

**Figure 1**.

Independently gauging the 0-form and 1-form components of the 2-group symmetry generates to a commuting square of gauge theories shown in figure 1. Gauging the 1-form centre symmetry $Z(G)$ results in a theory $\widehat{\mathcal{T}}$ with the Langlands dual gauge group $^L G$ and semi-direct product 0-form symmetry

$$\pi_1(^L\mathbf{G}) \rtimes \mathrm{Out}(\mathbf{G}) \,. \tag{1.3}$$

Then gauging outer automorphisms leads to a gauge theory with disconnected gauge groups and non-invertible categorical symmetries given by 2-representations. This reproduces and extends examples considered in [5] with a systematic inclusion of condensation defects and description of the full symmetry category.

---

[2]This perspective was changed in version 2 of [5].

## 1.3 Outline

The structure of the paper is as follows.

Section 2 reviews aspects of gauging a finite group in two dimensions. In section 3, we consider gauging a semi-direct product group in two dimensions by sequentially gauging subgroups, which serves as a warm-up for later sections. In sections 4 and 5 we consider gauging a finite group and finite split 2-group respectively in three dimensions. In section 6, we apply these results to compute the symmetry categories of gauge theories in three dimensions. Finally, in section 7, we outline generalisations to higher dimensions and more general higher groups.

*Note added: in the course of this project we were informed of overlapping papers [22] and [23]. We are grateful to the authors of these papers for coordinating release and agreeing to a delay to accommodate the second author's paternity leave.*

## 2 Two dimensions: finite group

Finite global symmetries and their 't Hooft anomalies in two dimensions are described by a unitary fusion category that captures the spectrum and properties of topological line defects [24–29]. In this section, we review aspects of the fusion categories associated to a finite group and its gauging following [14–17]. This will introduce notation and useful ingredients and set the stage for higher dimensions.

### 2.1 Finite groups

Consider a theory $\mathcal{T}$ with finite group symmetry $G$ that is free from 't Hooft anomalies. The associated symmetry category is $\mathsf{Vec}(G)$. The simple objects are topological lines labelled by group elements $g \in G$. They have morphisms

$$\mathrm{Hom}_{\mathcal{T}}(g, g') = \begin{cases} \mathbb{C} & \text{if} \quad g = g' \\ \emptyset & \text{if} \quad g \neq g' \end{cases} \tag{2.1}$$

and satisfy

$$g \otimes g' = gg' \qquad g^* = g^{-1} \tag{2.2}$$

with trivial associator. The dimensions of all simple objects is 1. These properties are summarised in figure 2.

**Figure 2**.

A general object is a direct sum of symmetry lines

$$V = \bigoplus_{g \in G} n_g\, g \qquad n_g \in \mathbb{Z}_+\,, \tag{2.3}$$

or equivalently a $G$-graded vector space

$$V = \bigoplus_{g \in G} V_g \tag{2.4}$$

under the identification $V_g = \mathbb{C}^{n_g}$.

The sum and product of general objects in the symmetry category are then identified with direct sum and tensor product of graded vector spaces,

$$\begin{aligned}
(V \oplus W)_g &= V_g \oplus W_g \\
(V \otimes W)_g &= \bigoplus_{hh'=g} V_h \otimes W_{h'} \,.
\end{aligned} \tag{2.5}$$

The morphisms are homogeneous linear transformations

$$\mathrm{Hom}_{\mathcal{T}}(V, V') = \bigoplus_{g \in G} \mathrm{Hom}(V_g, V'_g) \,. \tag{2.6}$$

The composition of morphisms is then induced by matrix multiplication.

## 2.2 Gauging a finite group

Let us now gauge the finite group $G$. The resulting theory $\mathcal{T}/G$ has topological Wilson lines transforming in linear representations of $G$.

The associated symmetry category is denoted by $\mathsf{Rep}(G)$. The objects are topological Wilson lines in linear representations $\Phi : G \to GL(W)$. The morphisms $\mathrm{Hom}_{\mathcal{T}}(\Phi, \Phi')$ are intertwiners between representations, while sum and product are direct sum $\Phi \oplus \Phi'$ and tensor product $\Phi \otimes \Phi'$ of representations and duals $\Phi^*$ are complex conjugate representations. The dimension of an object is the dimension of the representation $\dim \Phi$. The simple objects correspond to Wilson lines in irreducible representations of $G$. These properties are illustrated in figure 3.

**Figure 3**.

Let us now summarise how to reproduce the symmetry category $\mathsf{Rep}(G)$ of the theory $\mathcal{T}/G$ starting from the symmetry category $\mathsf{Vec}(G)$ of $\mathcal{T}$ and gauging $G$. The construction proceeds via the object

$$A = \bigoplus_{g \in G} g \,, \tag{2.7}$$

which is equivalently the graded vector space with $A_g = \mathbb{C}$ for all elements $g \in G$. This inherits from group multiplication the structure of a Frobenius algebra in $\mathsf{Vec}(G)$, with a multiplication morphism $\mu : A \otimes A \to A$ and unit $u : 1 \to A$.

The correlation functions in $\mathcal{T}/G$ are then defined by correlation functions in $\mathcal{T}$ with a network of topological lines $A$ inserted. Expanding into components, insertion of this network implements a summation over networks of $G$-symmetry lines or equivalently flat connections for $G$.

Similarly, a topological line in $\mathcal{T}/G$ is defined as a topological line in $\mathcal{T}$ together with a specification of how networks of the topological line $A$ may consistently end on it from the left and right. This is encoded in the structure of a $(A, A)$-bimodule. Starting from a topological line $V$ in $\mathcal{T}$, one specifies morphisms

$$l \in \operatorname{Hom}_{\mathcal{T}}(A \otimes V, V)$$
$$r \in \operatorname{Hom}_{\mathcal{T}}(V \otimes A, V) \tag{2.8}$$

satisfying compatibility conditions involving the multiplication $\mu : A \otimes A \to A$ and unit $u : 1 \to A$, which define the structure of an $(A, A)$-bimodule.

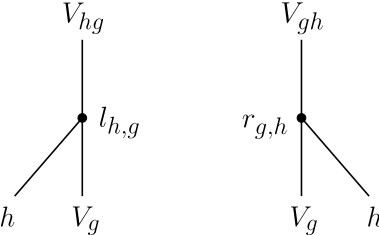

**Figure 4**.

The components of these morphisms are

$$l_{h,g} \in \operatorname{Hom}_{\mathcal{T}}(h \otimes V_g, V_{hg}) \qquad r_{g,h} \in \operatorname{Hom}_{\mathcal{T}}(V_g \otimes h, V_{gh}) \tag{2.9}$$

and specify how individual symmetry defects end on the line, as illustrated in figure 4. The component morphisms are subject to the relations

$$l_{hh',g} = l_{h,h'g} \circ l_{h',g} \qquad r_{g,hh'} = r_{gh,h'} \circ r_{g,h} \tag{2.10}$$

and

$$l_{h,gh'} \circ r_{g,h'} = r_{hg,h'} \circ l_{h,g} \tag{2.11}$$

together with the normalisations $l_{e,g} = 1$ and $r_{g,e} = 1$.

There are many ways to present solutions to these conditions. We choose a presentation that is convenient for our purposes. The equations (2.10) imply the components $l_{h,g}$, $r_{g,h}$ are invertible and identifies $V_g \cong W = \mathbb{C}^n$ for all $g \in G$. They may be determined by the components $l_{h,e}$, $r_{e,g}$ via the formula

$$l_{h,h'} = l_{hh',e} \circ (l_{h',e})^{-1} \qquad r_{h,h'} = r_{e,hh'} \circ (r_{e,h})^{-1}. \tag{2.12}$$

To formulate the remaining conditions on $l_{h,e}$, $r_{e,g}$, we introduce the combination

$$\Phi_g := (r_{e,g})^{-1} \circ l_{g,e} \in \operatorname{Hom}(W, W). \tag{2.13}$$

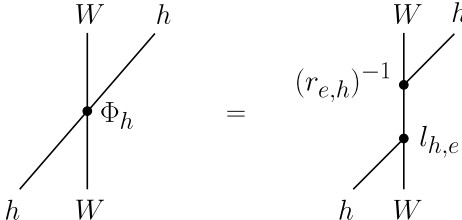

**Figure 5**.

This determines the phases attached to a symmetry generator crossing the line, as illustrated in figure 5. The remaining equations (2.11) imply that

$$\Phi_g \circ \Phi_h \ = \ \Phi_{gh} \,. \tag{2.14}$$

This equation encodes the requirement that in order to define a topological line in $\mathcal{T}/G$, $V$ must be moveable through networks of symmetry defects in $\mathcal{T}$. This is illustrated in figure 6. The isomorphism class of the resulting line operator in $\mathcal{T}/G$ depends only on the combination $\Phi_g$, rather than individual $l_{g,e}$, $r_{e,g}$ [3].

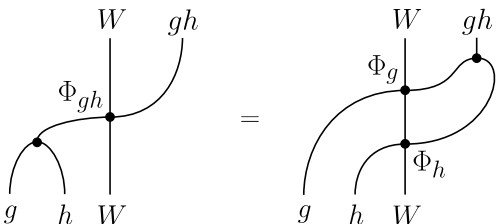

**Figure 6**.

Let us summarise the situation. Topological lines in $\mathcal{T}/G$ are labelled by linear representations $\Phi : G \to GL(W)$. This reproduces the classification of Wilson lines in a manner that will generalise to topological surfaces in higher dimensions.

The sum, product and morphisms of topological lines in $\mathcal{T}/G$ may also be computed from those of the parent topological lines in $\mathcal{T}$. It is straightforward to check that sum and product reproduce the direct sum and tensor product of representations. Let us briefly summarise the computation of morphisms. The morphisms $\mathrm{Hom}_{\mathcal{T}/G}(\Phi, \Phi')$ arise from morphisms $m \in \mathrm{Hom}_{\mathcal{T}}(W, W')$ subject to

$$m \circ \Phi_g \ = \ \Phi'_g \circ m \tag{2.15}$$

This arises from commutation with symmetry lines as illustrated in figure 7. This reproduces the intertwiners between representations $\Phi, \Phi'$. The symmetry category may therefore be identified with $\mathsf{Rep}(G)$.

---

[3]In reference [17] it is shown that it is always possible to choose $r_{e,g} = 1$ within an isomorphism class whereupon $\Phi_g = l_{g,e}$.

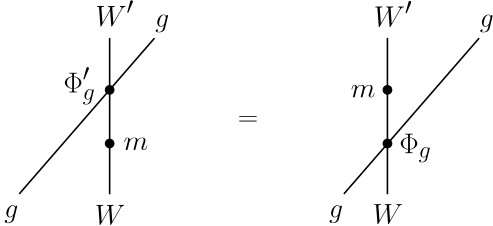

**Figure 7**.

## 2.3 Discrete torsion

A generalisation is to gauge $G$ with discrete torsion $c \in H^2(G, U(1))$, resulting in a theory $(\mathcal{T}/G)_c$. This is accomplished by twisting the multiplication morphism $\mu : A \otimes A \to A$ by a representative of the class $c$ and sums over networks of symmetry defects where vertices $g \otimes h \to gh$ contribute with an additional phase $c(g, h)$.

Let us consider a general situation of topological interfaces between pairs of theories with discrete torsion $c^l$, $c^r$. The topological interfaces are constructed analogously to above, with the result that now

$$\Phi_{gh} \;=\; c(g, h) \cdot \Phi_g \circ \Phi_h \,, \tag{2.16}$$

where

$$c(g, h) \;=\; c^l(g, h) - c^r(g, h) \,. \tag{2.17}$$

The interpretation of this equation is illustrated in figure 8.

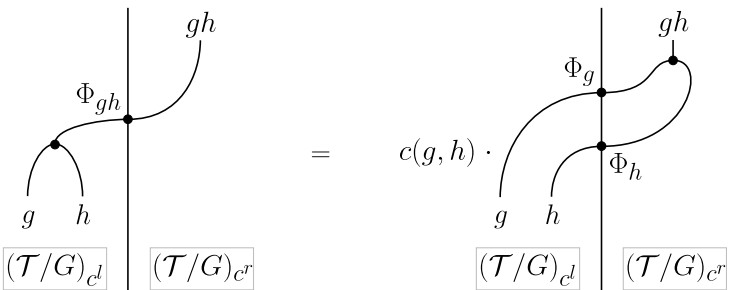

**Figure 8**.

In summary, the topological interfaces between theories $(\mathcal{T}/G)_{c^l}$ and $(\mathcal{T}/G)_{c^r}$ are labelled by projective representations $\Phi : G \to GL(W)$ with cocycle $c = c^l - c^r$. They are consistent topological Wilson lines in projective representations, whose anomalous gauge transformations are compensated by anomaly inflow to the interface from the SPT phases. The topological lines in a given theory $(\mathcal{T}/G)_c$ are Wilson lines in ordinary representations $\Phi : G \to GL(W)$.

Let us denote the associated category of projective representations by $\mathsf{Rep}^c(G)$. This does not generally have a fusion structure since cocycles are additive under tensor product.

However, there are functors

$$\mathsf{Rep}^c(G) \times \mathsf{Rep}^{c'}(G) \;\longrightarrow\; \mathsf{Rep}^{c+c'}(G) \tag{2.18}$$

arising from collision of topological interfaces. In particular, there are left and right actions of the fusion category $\mathsf{Rep}(G)$ on the categories $\mathsf{Rep}^c(G)$ arising from collision of topological lines with topological interfaces.

## 3 Two dimensions: semi-direct product

We remain in two dimensions and consider gauging a semi-direct product group $G = A \rtimes H$ with $A$ abelian. While this is a special case of the general construction in section 2, it is illuminating to gauge in two steps. The first step is to gauge $A$, resulting in an intermediate theory with semi-direct product symmetry $\widehat{G} = \widehat{A} \rtimes H$. The second is to gauge $H$. The combination of these steps is equivalent to gauging $G = A \rtimes H$.

This construction is in fact entirely symmetric between $A$, $\widehat{A}$ and results in the square symmetry categories illustrated in figure 9.

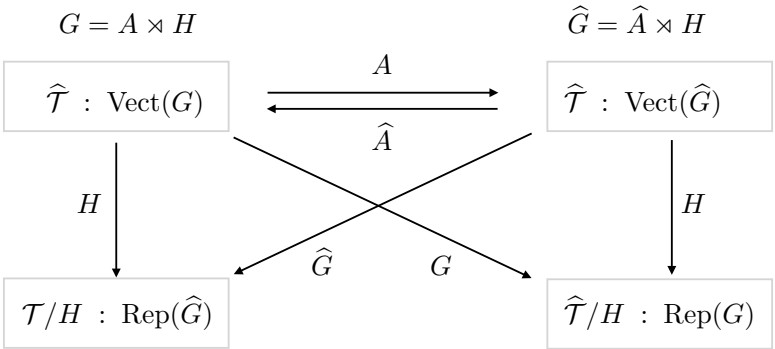

**Figure 9**.

While the final result must reproduce the symmetry category $\mathsf{Rep}(G)$, this will reproduce Mackey's construction of irreducible representations of semi-direct products $G = A \rtimes H$ by induction. Moreover, it will provide valuable insights into gauging higher groups in higher dimensions, which will be utilised in section 5.

### 3.1 Semi-direct product

We consider then a theory $\mathcal{T}$ with anomaly free finite symmetry group

$$G = A \rtimes_\varphi H \tag{3.1}$$

constructed from the following data:

- A finite group $H$.

- A finite abelian group $A$.

- A homomorphism $\varphi : H \to \mathrm{Aut}(A)$.

The group elements are pairs $g = (a, h)$ with group law

$$(a, h) \cdot (a', h') \;=\; (a\varphi_h(a'), hh') \,. \tag{3.2}$$

Introducing the notation $a = (a, 1)$ and $h = (1, h)$, we have $ah = (a, h)$ and $ha = (\varphi_h(a), h)$ and consequently $\varphi_h(a) = a^h$, where we define $a^h := hah^{-1}$. We often drop the homomorphism from notation and write $G = A \rtimes H$.

Gauging $A \subset G$ results in a theory $\widehat{\mathcal{T}} := \mathcal{T}/A$ with anomaly free finite symmetry

$$\widehat{G} = \widehat{A} \rtimes_{\widehat{\varphi}} H \tag{3.3}$$

where

$$\widehat{A} \;:=\; \mathrm{Hom}(A, U(1)) \tag{3.4}$$

is the Pontryagin dual of $A$ and $\widehat{\varphi} : H \to \mathrm{Aut}\,\widehat{A}$ is the dual homomorphism [17]. Elements of the Pontryagin dual are characters $\chi : A \to U(1)$ with dual action $\widehat{\varphi}_h(\chi) = \chi^h$ such that $\chi^h(a) = \chi(a^h)$. We again drop the homomorphism from notation and write $\widehat{G} = \widehat{A} \rtimes H$, which we emphasise is not the Pontryagin dual of $G$.

The situation is entirely symmetric under exchanging $A$, $\widehat{A}$: gauging $\widehat{A} \subset \widehat{G}$ in $\widehat{\mathcal{T}}$ reproduces the original theory $\mathcal{T}$ with symmetry $G$. This is summarised in the horizontal arrows in figure 9.

## 3.2 Gauging a semi-direct product

We now consider gauging the symmetry $H \subset G, \widehat{G}$ in $\mathcal{T}, \widehat{\mathcal{T}}$, represented by the vertical arrows in figure 9. This results in a pair of theories $\mathcal{T}/H, \widehat{\mathcal{T}}/H$. The combination of these operations is equivalent to gauging the whole symmetry $\widehat{G}, G$ of $\widehat{\mathcal{T}}, \mathcal{T}$ and must reproduce the symmetry categories $\mathsf{Rep}(\widehat{G})$, $\mathsf{Rep}(G)$. In other words,

$$\mathcal{T}/H = \widehat{\mathcal{T}}/\widehat{G} \qquad \widehat{\mathcal{T}}/H = \mathcal{T}/G \tag{3.5}$$

This is summarised by the commutativity of arrows in figure 9.

For concreteness, we will consider gauging $H \subset \widehat{G}$ in $\widehat{\mathcal{T}}$. This is a special case of gauging a general finite subgroup and the resulting symmetry categories in the general case have been studied in [30, 31]. In the situation considered here, we show that this reproduces the symmetry category $\mathsf{Rep}(G)$.

Our starting point is therefore theory $\widehat{\mathcal{T}}$ with symmetry $\widehat{G} = \widehat{A} \rtimes H$. Let us briefly summarise the associated aspects of the symmetry category $\mathsf{Vec}(\widehat{G})$. A general object is a $\widehat{G} = \widehat{A} \rtimes H$-graded vector space,

$$V \;=\; \bigoplus_{\chi, h} V_{\chi, h} \,, \tag{3.6}$$

where the summation runs over group elements $g = \chi h$ with $\chi \in \widehat{A}$, $h \in H$. The sum and product are direct sum and tensor product of graded vector spaces

$$
\begin{aligned}
(V \oplus W)_{\chi, h} \;&=\; V_{\chi, h} \oplus W_{\chi, h} \\
(V \otimes W)_{\chi, h} \;&=\; \bigoplus_{\substack{\chi_1 \cdot (\chi_2)^{h_1} = \chi \\ h_1 \cdot h_2 = h}} V_{\chi_1, h_2} \otimes W_{\chi_2, h_2} \,,
\end{aligned}
\tag{3.7}
$$

while morphisms are homogeneous linear maps,

$$\mathrm{Hom}_{\widehat{\mathcal{T}}}(V, W) = \bigoplus_{\chi, h} \mathrm{Hom}(V_{\chi, h}, W_{\chi, h}). \tag{3.8}$$

We now gauge $H \subset \widehat{G}$ and compute the symmetry category of $\widehat{\mathcal{T}}/H = \mathcal{T}/G$, generalising the construction in section 2.

### 3.2.1   Objects

A topological line in $\widehat{\mathcal{T}}/H$ is defined operationally as a topological line in $\widehat{\mathcal{T}}$ together with instructions for how networks of the Frobenius algebra object

$$A_H = \bigoplus_{h \in H} h \tag{3.9}$$

end on it consistently from the left and right. In particular, starting from a topological line $V$ in $\mathcal{T}$, one now specifies morphisms

$$\begin{aligned} l &\in \mathrm{Hom}_{\mathcal{T}}(A_H \otimes V, V) \\ r &\in \mathrm{Hom}_{\mathcal{T}}(V \otimes A_H, V) \end{aligned} \tag{3.10}$$

forming the structure of a $(A_H, A_H)$-bimodule.

The components of these morphisms

$$l_{h,g} \in \mathrm{Hom}_{\mathcal{T}}(h \otimes V_g, V_{hg}) \qquad r_{g,h} \in \mathrm{Hom}_{\mathcal{T}}(V_g \otimes h, V_{gh}) \tag{3.11}$$

are subject to the compatibility conditions

$$l_{hh',g} = l_{h,h'g} \circ l_{h',g} \qquad r_{g,hh'} = r_{gh,h'} \circ r_{g,h} \tag{3.12}$$

and

$$l_{h,gh'} \circ r_{g,h'} = r_{hg,h'} \circ l_{h,g} \tag{3.13}$$

together with the normalisations $l_{e,g} = 1$ and $r_{g,e} = 1$, where we now restrict attention to $h, h' \in H$ and $g \in \widehat{G} = \widehat{A} \rtimes H$.

We solve the equations analogously to section 2. First, equations (3.12) together with the normalisation conditions imply the morphisms are invertible and determined by the following two component morphisms

$$\begin{aligned} l_{h,\chi} &: \quad h \otimes V_{\chi,e} \;\rightarrow\; V_{\chi^h, h} \\ r_{\chi,h} &: \quad V_{h,e} \otimes h \;\rightarrow\; V_{\chi,h}, \end{aligned} \tag{3.14}$$

via the formulae

$$l_{h,\chi h'} = l_{hh',\chi} \circ (l_{h',\chi})^{-1} \qquad r_{\chi h,h'} = r_{\chi,hh'} \circ (r_{\chi,h})^{-1}. \tag{3.15}$$

Note that the right morphisms $r_{\chi,h}$ provide vector space isomorphisms $V_{\chi,h} \cong V_{\chi,e}$ for any $h \in H$. It is then convenient to define $W_\chi := V_{\chi,e}$ [4].

---

[4]The left morphism $l_{h,e}$ then further identifies $W_\chi = W_{\chi^h}$ and induces a decomposition into simple objects labelled by $H$-orbits in $\widehat{A}$. We postpone this step until our analysis of simple objects.

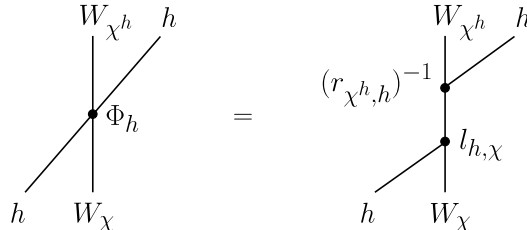

**Figure 10**.

We now introduce the combinations

$$\Phi_{h,\chi} = (r_{\chi^h,h})^{-1} l_{h,\chi} : W_\chi \to W_{\chi^h} \,, \tag{3.16}$$

which are the amplitudes associated to the intersection with a symmetry line, as in figure 10. The remaining equations (3.13) give

$$\Phi_{hh',\chi} = \Phi_{h,\chi^{h'}} \circ \Phi_{h',\chi} \,. \tag{3.17}$$

which directly encodes the topological nature of the resulting line. The isomorphism class of the line operator in $\mathcal{T}/G$ will depend only on the combination $\Phi_{h,\chi}$, rather than individual morphisms $l_{h,\chi}$, $r_{\chi,h}$.

In summary, objects are labelled by:

1. A collection of vector spaces $W_\chi$ indexed by $\chi \in \widehat{A}$.

2. A collection of invertible morphisms $\Phi_{h,\chi} : W_\chi \to W_{\chi^h}$ satisfying

$$\Phi_{hh',\chi} = \Phi_{h,\chi^{h'}} \circ \Phi_{h',\chi} \,.$$

This is the data of a linear representation $\Phi : G \to GL(W)$ with underlying vector space $W := \oplus_\chi W_\chi$ and action of group elements

$$\begin{aligned} \Phi_a(w_\chi) &= \chi(a) \cdot w_\chi \\ \Phi_h(w_\chi) &= \Phi_{h,\chi}(w_\chi) \,. \end{aligned} \tag{3.18}$$

It is straightforward to check that this defines a representation as a consequence of equation (3.17). We have therefore confirmed that objects in the symmetry category are indeed representations of $G = A \rtimes H$.

We note that this data can be framed more invariantly as follows:

1. A vector bundle $\pi : \mathcal{W} \to \widehat{A}$.

2. A homomorphism $\Phi : H \to \text{Aut}(\mathcal{W})$ satisfying

$$\pi \circ \Phi = \widehat{\varphi} \circ \pi \tag{3.19}$$

with the homomorphism $\widehat{\varphi} : H \to \text{Aut}(\widehat{A})$.

In other words, a $G$-equivariant vector bundle $\pi : W \to \widehat{A}$. This is a discrete analogue of the construction of representations of compact Lie groups from vector bundles on homogeneous spaces. The explicit description in terms of components is recovered by identifying fibers $\pi^{-1}(\chi) = W_\chi$ and the collection of vectors $\{w_\chi\}$ as a section.

### 3.2.2 Sum, Product, Morphisms

The sum, product and morphisms of objects in $\widehat{\mathcal{T}}/H$ may be computed from the operations of direct sum, tensor product and morphisms of graded vector spaces in $\widehat{\mathcal{T}}$ and coincide with direct sum, tensor products and intertwiners of representations of $G = A \rtimes H$. If we denote objects by pairs $(W, \Phi)$ then sum and tensor are

$$
\begin{aligned}
(W, \Phi) \oplus (W', \Phi') &= (W \oplus W', \Phi \oplus \Phi') \\
(W, \Phi) \otimes (W', \Phi') &= (W \otimes W', \Phi \otimes \Phi')
\end{aligned}
\tag{3.20}
$$

while morphisms are homogeneous linear maps $m : W \to W'$ satisfying $m \circ \Phi = \Phi' \circ m$.

### 3.3 Simple Objects

We now consider the decomposition of general objects in $\widehat{\mathcal{T}}/H$ into simple objects. This must reproduce the decompositions of representations of $G = A \rtimes H$ into irreducible representations.

### 3.3.1 Classification

First, the component morphisms $\Phi_{h,\chi} : W_\chi \to W_{\chi^h}$ mean a general object decomposes as a sum of objects supported on orbits of the $H$-action on $\widehat{A}$. We say that a representation is supported on an orbit $\mathcal{O} \subset \widehat{A}$ if

$$
W_\chi = 0 \quad \text{if} \quad \chi \notin \mathcal{O}.
\tag{3.21}
$$

Moreover, given a representation supported on an orbit $\mathcal{O}$, the collection of vector spaces $W_\chi$ with $\chi \in \mathcal{O}$ may decompose as direct sums with morphism $\Phi_{h,\chi} : W_\chi \to W_{\chi^h}$ acting in a block diagonal fashion preserving the direct sum decomposition.

In summary, a simple object is labelled by the following data:

1. A collection of vector spaces $W_\chi$ indexed by orbit elements $\chi \in \mathcal{O}$.

2. A collection of simple invertible morphisms $\Phi_{h,\chi} : W_\chi \to W_{\chi^h}$ satisfying

$$
\Phi_{hh',\chi} = \Phi_{h,\chi^{h'}} \circ \Phi_{h',\chi}.
\tag{3.22}
$$

This corresponds to an irreducible representation of the semi-direct product $G = A \rtimes H$.

Alternatively, the simple objects may be labelled by irreducible representations of stabilisers of orbits. That is, given collections $W$ and $\Phi$ as above, we can fix a representative $\chi_0 \in \mathcal{O}$ of the orbit and define $K := \mathrm{Stab}_H(\chi_0) \subset H$. Then, the morphisms $\Phi_{h,\chi_0} : W_{\chi_0} \to W_{\chi_0}$ with $h \in K$ define an irreducible representation $\Psi$ of $A \rtimes K \subset G$ by

$$
\Psi_a(w) := \chi_0(a) \cdot w
\tag{3.23}
$$

$$
\Psi_h(w) := \Phi_{h,\chi_0}(w).
\tag{3.24}
$$

Conversely, given an irreducible representation $\Psi$ of $A \rtimes K$, we can reconstruct the original irreducible representation as follows: For each orbit element $\chi \in \mathcal{O}$ we fix an element $h_\chi \in H$ such that

$$
\chi = \chi_0^{h_\chi}.
\tag{3.25}
$$

This determines $h_\chi$ up to right multiplication by $K$, and sets up an isomorphism of sets $\mathcal{O} \cong H/K$. It is then straightforward to check that the combination

$$\ell_{h,\chi} \ := \ h_{\chi^h}^{-1} \cdot h \cdot h_\chi \ \in \ K \tag{3.26}$$

lies in the stabiliser of the orbit representative. Then,

$$\Phi_{h,\chi} \ := \ \Psi(\ell_{h,\chi}) \tag{3.27}$$

solves the conditions (3.22) and determines an irreducible representation of the semi-direct product $G = A \rtimes H$. One can check that, up to isomorphism, the collection $\Phi$ does not depend on the choices of $\chi_0 \in \mathcal{O}$ and $h_\chi \in H$.

In summary, the simple objects are in 1-1 correspondence with

1. A character $\chi_0 \in \widehat{A}$ with stabiliser $K \subset H$.

2. An irreducible representation $\Psi$ of $A \rtimes K$.

This reproduces Mackey's construction and classification of irreducible representations of a semi-direct product $G = A \rtimes H$ by induction. Let us denote this induction by

$$(W, \Phi) \ = \ \mathrm{Ind}_{A \rtimes K}^{G}(\Psi) \,. \tag{3.28}$$

This presentation of the simple objects reflects a physical construction of topological lines in the finite gauge theory $\mathcal{T}/G$ by imposing Dirichlet boundary conditions that break the gauge symmetry to a subgroup $A \rtimes K \subset G$, supplemented by a topological Wilson line for the unbroken gauge symmetry.

### 3.3.2 Fusion of Simple Objects

We have shown that fusion corresponds to tensor product of representations of $G = A \rtimes H$. The construction of irreducible representations by induction provides a computational tool to compute the fusion ring of simple objects.

Let us denote the collection of $H$-orbits in $\widehat{A}$ by $\{\mathcal{O}_j\}$ and introduce a corresponding collection of orbit representatives $\{\chi_j\}$. We denote their stabilisers by $K_j := \mathrm{Stab}_H(\chi_j) \subset H$. Then, the simple objects or irreducible representations $\Phi_j : G \to GL(W_j)$ are constructed by induction as

$$(W_j, \Phi_j) \ = \ \mathrm{Ind}_{G_j}^{G}(\Psi_j) \,, \tag{3.29}$$

where $G_j := A \rtimes K_j$ and $\Psi_j$ is an irreducible representation of $G_j$ as above.

In order to compute their fusions rules, we must first understand how an irreducible representation $(W_j, \Phi_j)$ decomposes upon restriction to $G_i \subset G$. It is clear that this decomposition will involve a sum over $K_i$-orbits $\widetilde{\mathcal{O}} \subset \mathcal{O}_j$, whose summands we will determine in the following.

Given a $K_i$-orbit $\widetilde{\mathcal{O}} \subset \mathcal{O}_j$, we can fix a representative $\widetilde{\chi} \in \widetilde{\mathcal{O}}$ with stabiliser

$$\widetilde{K} \ := \ \mathrm{Stab}_{K_i}(\widetilde{\chi}) \ \equiv \ K_i \cap (h_{\widetilde{\chi}} K_j h_{\widetilde{\chi}}^{-1}) \,, \tag{3.30}$$

where, as before, we fixed elements $h_\chi \in H$ such that

$$\chi = \chi_j^{h_\chi} \tag{3.31}$$

for each $\chi \in \widetilde{\mathcal{O}}$ (in particular $\widetilde{\chi} = \chi_j^{h_{\widetilde{\chi}}}$). Let us now repeat the construction of induced representations discussed above. For each orbit element $\chi \in \widetilde{\mathcal{O}}$, we fix $\widetilde{h}_\chi \in K_i$ such that

$$\chi = \widetilde{\chi}^{\widetilde{h}_\chi} . \tag{3.32}$$

They are determined up to right multiplication by elements in $\widetilde{K}$ and this fixes an isomorphism of sets $\widetilde{\mathcal{O}} = K_i / \widetilde{K}$. It is now straightforward to check that $\widetilde{h}_\chi = h_\chi \cdot h_{\widetilde{\chi}}^{-1}$ solves condition (3.32) so that

$$\widetilde{\ell}_{h,\chi} := \widetilde{h}_{\chi^h}^{-1} \cdot h \cdot \widetilde{h}_\chi \equiv h_{\widetilde{\chi}} \cdot \ell_{h,\chi} \cdot h_{\widetilde{\chi}}^{-1} . \tag{3.33}$$

Consequently, upon restriction to elements $h \in K_i$, we find that

$$\begin{aligned} (\Phi_j)_{h,\chi} &\equiv \Psi_j(\ell_{h,\chi}) \\ &= \Psi_j(h_{\widetilde{\chi}}^{-1} \cdot \widetilde{\ell}_{h,\chi} \cdot h_{\widetilde{\chi}}) \\ &= \Psi_j^{\widetilde{\chi}}(\widetilde{\ell}_{h,\chi}) \end{aligned} \tag{3.34}$$

for all $\chi \in \widetilde{\mathcal{O}}$, where the last line corresponds to the induction of the linear representation $\Psi_j^{\widetilde{\chi}}$ of $\widetilde{K}$ defined by

$$\Psi_j^{\widetilde{\chi}}(h) := \Psi_j(h_{\widetilde{\chi}}^{-1} \cdot h \cdot h_{\widetilde{\chi}}) . \tag{3.35}$$

Thus, the restriction of $(W_j, \Phi_j)$ to $G_i$ is summarised by an instance of Mackey's decomposition formula,

$$\mathrm{Res}_{G_i}^G \mathrm{Ind}_{G_j}^G(\Psi_j) = \bigoplus_{[\widetilde{\chi}]} \mathrm{Ind}_{\widetilde{G}}^{G_i}(\Psi_j^{\widetilde{\chi}}) , \tag{3.36}$$

where the summation is over representatives $\widetilde{\chi}$ of $K_i$-orbits in $\mathcal{O}_j$ and $\widetilde{G} = A \rtimes \widetilde{K}$.

By combining this result with the push-pull formula for induction and restriction, we obtain a convenient method to compute the fusion of simple objects,

$$\begin{aligned} \mathrm{Ind}_{G_i}^G(\Psi_i) \otimes \mathrm{Ind}_{G_j}^G(\Psi_j) &= \mathrm{Ind}_{G_i}^G(\Psi_i \otimes \mathrm{Res}_{G_i}^G \mathrm{Ind}_{G_j}^G(\Psi_j)) \\ &= \bigoplus_{[\widetilde{\chi}]} \mathrm{Ind}_{G_i}^G\big(\Psi_i \otimes \mathrm{Ind}_{\widetilde{G}}^{G_i}(\Psi_j^{\widetilde{\chi}})\big) \\ &= \bigoplus_{[\widetilde{\chi}]} \mathrm{Ind}_{\widetilde{G}}^G\big(\Psi_i \otimes \Psi_j^{\widetilde{\chi}}\big) , \end{aligned} \tag{3.37}$$

where the summation again runs over representatives $\widetilde{\chi}$ of $K_i$-orbits in $\mathcal{O}_j$. The representation $\Psi_i \otimes \Psi_j^{\widetilde{\chi}}$ of $\widetilde{G}$ may be reducible and admit a further decomposition into irreducible representations. Nevertheless, this provides a concrete computational tool and we will see analogues for 2-representations in sections 4 and 5.

### 3.4 Example

Consider a theory $\mathcal{T}$ with finite symmetry group

$$G = D_{2n} \cong \mathbb{Z}_n \rtimes \mathbb{Z}_2, \tag{3.38}$$

with $n$ even. In other words, $H \cong \mathbb{Z}_2$ with group elements $\{1, h\}$ and $A \cong \mathbb{Z}_n$ with group elements $\{1, a, \ldots, a^{n-1}\}$, which are acted upon by $H$ through $h : a \mapsto a^{-1}$. Gauging $A$ generates another theory $\widehat{\mathcal{T}}$ with isomorphic symmetry group $\widehat{G} = D_{2n}$ constructed from $\widehat{A} \cong \mathbb{Z}_n$ with elements $\{1, \chi, \ldots, \chi^{n-1}\}$ and $H$-action $h : \chi \mapsto \chi^{-1}$.

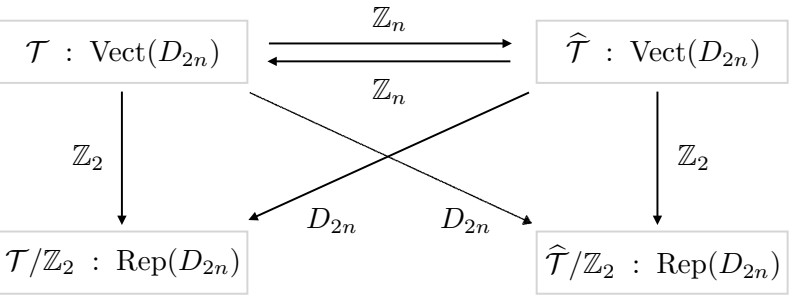

**Figure 11**.

Gauging $H \cong \mathbb{Z}_2$ produces a pair of theories with symmetry category $\mathsf{Rep}(D_{2n})$, as shown in figure 11. Let us reproduce the symmetry category starting from $\widehat{\mathcal{T}}$. There are the following simple objects:

- The 1-dimensional orbit $1 = \{1\}$ may be supplemented by irreducible representations $1$, $w$ of its stabiliser $\mathbb{Z}_2$. We denote the corresponding simple objects by $1$, $w$. [5]

- The 1-dimensional orbit $o = \{\chi^{\frac{n}{2}}\}$ may be supplemented by irreducible representations $1$, $w$ of its stabiliser $\mathbb{Z}_2$. We denote the corresponding simple objects by $o, ow$.

- The 2-dimensional orbits $\{\chi^i, \chi^{n-i}\}$ with $j = 1, \ldots, \frac{n}{2} - 1$ have trivial stabilisers. We denote the corresponding simple objects by $\mathcal{O}_j$, $j = 1, \ldots, \frac{n}{2} - 1$.

The fusion rules for irreducible representations may be computed following the recipe above and are given by

$$w \otimes w = 1 \qquad o \otimes o = 1 \qquad o \otimes w = ow \tag{3.39}$$

$$w \otimes \mathcal{O}_j = \mathcal{O}_j \qquad o \otimes \mathcal{O}_j = \mathcal{O}_j \tag{3.40}$$

$$\mathcal{O}_i \otimes \mathcal{O}_j = \mathcal{O}_{i+j} \oplus \mathcal{O}_{i-j}, \tag{3.41}$$

where in the final line it is understood that $\mathcal{O}_0 = 1 \oplus w$ and $\mathcal{O}_{\frac{n}{2}} = o \oplus ow$ and $\mathcal{O}_j = \mathcal{O}_{\frac{n}{2}+j}$ for $j \neq 0, \frac{n}{2} \bmod n$.

---

[5] They are pure topological Wilson lines for $H \cong \mathbb{Z}_2$.

For $n = 4$ this simplifies to

$$w \otimes w = 1 \qquad o \otimes o = 1 \qquad o \otimes w = ow \tag{3.42}$$

$$w \otimes \mathcal{O} = \mathcal{O} \qquad o \otimes \mathcal{O} = \mathcal{O} \tag{3.43}$$

$$\mathcal{O} \otimes \mathcal{O} = 1 \oplus w \oplus o \oplus ow \tag{3.44}$$

and the symmetry category $\mathsf{Rep}(D_8)$ is a Tambara-Yamagami fusion category based on the abelian group $\mathbb{Z}_2 \times \mathbb{Z}_2$ [32].

## 4 Three dimensions: groups

In this section, we consider gauging a finite group symmetry $G$ in three dimensions. As in section 2, the resulting theory has topological Wilson lines in representations of $G$ and generating a $\mathsf{Rep}(G)$ 1-form symmetry. In addition, there are now topological surface operators arising from combinations of two-dimensional condensation defects and SPT phases. The purpose of this section is to show that the full spectrum of topological defects is captured by a fusion 2-category $2\mathsf{Rep}(G)$ whose objects consist of 2-representations of the finite group $G$.

An output of the construction is a systematic derivation of properties of condensation defects associated to topological Wilson lines generating a $\mathsf{Rep}(G)$ 1-form symmetry. This analysis applies for general non-abelian finite groups $G$, and in this sense generalises the analysis of condensation defects that arise from higher gauging of invertible 1-form symmetries on surfaces in [4].

### 4.1 Finite group symmetry

Consider a three-dimensional theory $\mathcal{T}$ with finite group symmetry $G$ without 't Hooft anomalies. The associated symmetry category $2\mathsf{Vec}(G)$ is a fusion 2-category [33–35]. Let us summarise some of the important data. The simple objects are topological surfaces labelled by group elements $g \in G$ with

$$g \otimes g' = gg' \qquad g^\# = g^{-1} \tag{4.1}$$

where $g^\#$ denotes a topological surface with the opposite orientation. The dimension of all simple objects is 1. These properties are illustrated in figure 12.

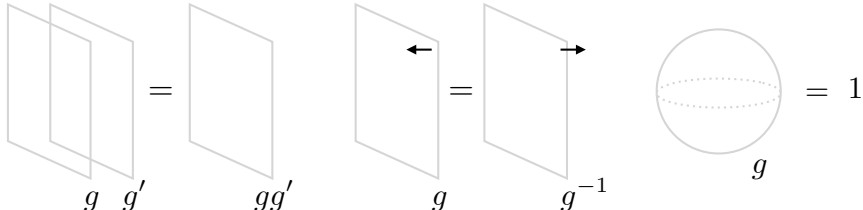

**Figure 12**.

The categories of 1-morphism capture topological lines at junctions between surfaces, and are given by

$$\text{Hom}_{\mathcal{T}}(g, g') = \begin{cases} \text{Vect} & g = g' \\ 0 & g \neq g' \end{cases}. \tag{4.2}$$

In other words, there are only 1-endomorphisms consisting of vector spaces spanned by sums of the identity line operator on a symmetry generating surface. The composition and fusion of 1-endomorphisms is determined by tensor product of vector spaces. The composition and fusion of 1-morphisms are illustrated in figure 13.

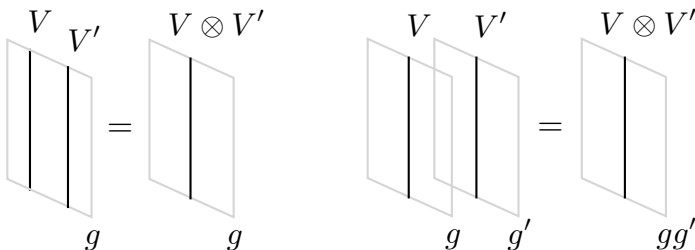

**Figure 13**.

A general object can be expressed as a sum

$$\mathcal{R} = \bigoplus_{g \in G} n_g\, g \tag{4.3}$$

with non-negative integers $n_g \in \mathbb{Z}_+$. This is represented by a $G$-graded set

$$\mathcal{R} = \bigsqcup_{g \in G} \mathcal{R}_g, \tag{4.4}$$

under an identification $\mathcal{R}_g \cong \{1, \ldots, n_g\}$ where elements of the set index the copies of the symmetry defect $g$ in (4.3). The sum and product of general objects are then disjoint union and cartesian product of $G$-graded sets,

$$
\begin{aligned}
(\mathcal{R} \oplus \mathcal{R}')_g &= \mathcal{R}_g \sqcup \mathcal{R}'_g \\
(\mathcal{R} \otimes \mathcal{R}')_g &= \bigsqcup_{g = hh'} \mathcal{R}_h \times \mathcal{R}_{h'}.
\end{aligned}
\tag{4.5}
$$

The 1-morphisms are

$$\text{Hom}_{\mathcal{T}}(\mathcal{R}, \mathcal{R}') = \bigoplus_{g \in G} \text{Vect}_{\mathcal{R}_g \times \mathcal{R}'_g} \tag{4.6}$$

whose summands are categories of $\mathcal{R}_g \times \mathcal{R}'_g$-graded vector spaces or alternatively $n_g \times n'_g$ 2-matrices whose components are vector spaces. The composition of 1-morphisms is determined by matrix multiplication and tensor product of vector spaces. Fusion of morphisms is determined by tenor product of matrices and vector spaces. The 2-morphisms are homogeneous linear maps between graded vector spaces.

## 4.2 Gauging a finite group

Now consider gauging the finite symmetry $G$ of $\mathcal{T}$. We compute the symmetry category of $\mathcal{T}/G$ by gauging an appropriate algebra object in the symmetry category $2\mathsf{Vec}(G)$ of $\mathcal{T}$. We classify the topological surfaces and explain their physical interpretation as condensation defects. We show that the topological surfaces are in 1-1 correspondence with 2-representations of $G$ and derive their fusion and 1-morphisms, which identifies the symmetry category with the fusion 2-category $2\mathrm{Rep}(G)$.

### 4.2.1 Objects

Following section 2, the strategy is to define topological surfaces in $\mathcal{T}/G$ as topological surfaces in $\mathcal{T}$ together with instructions for how networks of symmetry defects end on them in a manner that is consistent with their topological nature.

This construction again proceeds via the algebra object in $\mathcal{T}$,

$$\mathcal{A} = \bigoplus_{g \in G} g \,, \tag{4.7}$$

corresponding to the $G$-graded set with $\mathcal{A}_g \cong \{1\}$ for all elements $g \in G$. A topological surface in $\mathcal{T}/G$ is then specified by a topological surface in $\mathcal{T}$ together with instructions for how $\mathcal{A}$ ends on it inside correlation functions, which need to satisfy various compatibility conditions to ensure that the resulting surface is indeed topological.

The starting point is a general topological surface in $\mathcal{T}$ labelled by a $G$-graded set $\mathcal{R}$. This is supplemented by 1-morphisms

$$\begin{aligned} l &\in \mathrm{Hom}_{\mathcal{T}}(\mathcal{A} \otimes \mathcal{R}, \mathcal{R}) \\ r &\in \mathrm{Hom}_{\mathcal{T}}(\mathcal{R} \otimes \mathcal{A}, \mathcal{R}) \end{aligned} \tag{4.8}$$

that specify how topological surfaces $\mathcal{A}$ end on it from the left and right. To formulate the additional data and constraints concretely, we consider the component 1-morphisms

$$\begin{aligned} l_{h,g} &\in \mathrm{Hom}_{\mathcal{T}}(h \otimes \mathcal{R}_g, \mathcal{R}_{hg}) \\ r_{g,h} &\in \mathrm{Hom}_{\mathcal{T}}(\mathcal{R}_g \otimes h, \mathcal{R}_{gh}) \,, \end{aligned} \tag{4.9}$$

which are topological lines specifying how individual symmetry defects end on the surface. The interpretation of these 1-morphisms is illustrated in figure 14.

As in two dimensions, these topological lines must satisfy compatibility conditions to ensure consistency with topological manipulations of networks of surfaces in the bulk. However, in three dimensions, the conditions are not equalities but implemented by invertible topological local operators, which are 2-isomorphisms in the symmetry category of $\mathcal{T}$.

In particular, the component 1-morphisms are supplemented by the following topological local operators or 2-isomorphisms:

- There are normalisation 2-isomorphisms

$$\begin{aligned} \Psi_g^l &: \quad 1_{\mathcal{R}_g} \Rightarrow l_{e,g} \\ \Psi_g^r &: \quad 1_{\mathcal{R}_g} \Rightarrow r_{g,e} \end{aligned} \tag{4.10}$$

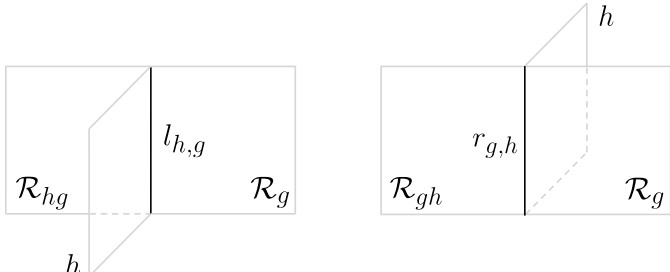

**Figure 14**.

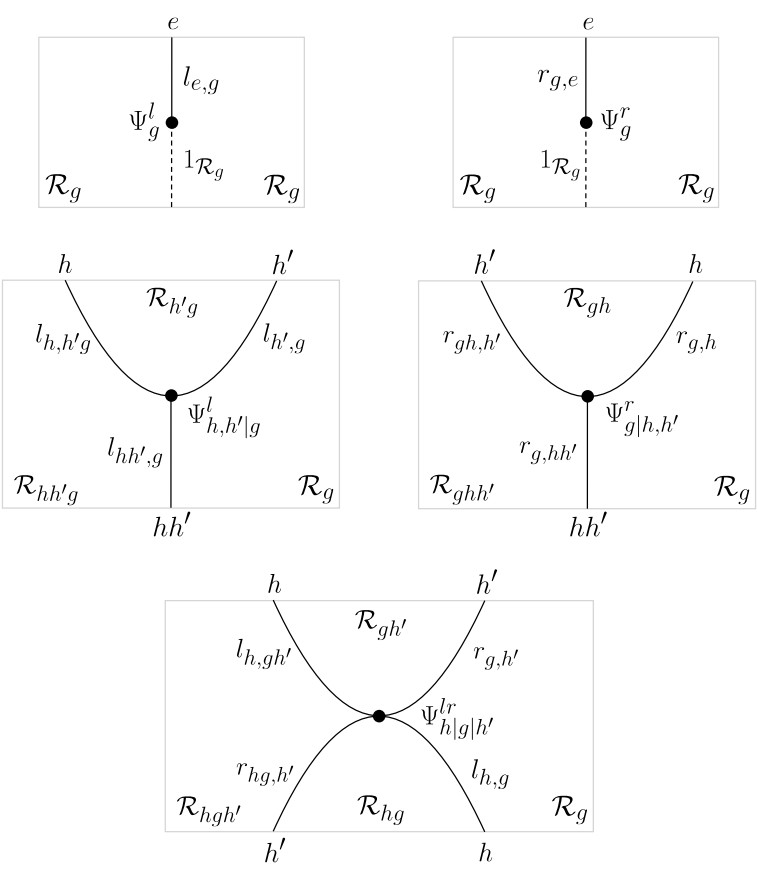

**Figure 15**.

and correspond to topological local operators on which the topological line operators $l_{e,g}, r_{g,e}$ may end.

- There are left and right 2-isomorphisms

$$
\begin{aligned}
\Psi^l_{h,h'|g} &: \quad l_{hh',g} \Rightarrow l_{h,h'g} \otimes l_{h',g} \\
\Psi^r_{g|h,h'} &: \quad r_{g,hh'} \Rightarrow r_{gh,h'} \otimes r_{g,h}
\end{aligned}
\tag{4.11}
$$

implementing compatibility with fusion of symmetry defects.

- There are 2-isomorphisms

$$\Psi^{lr}_{h|g|h'} : l_{h,gh'} \otimes r_{g,h'} \Rightarrow r_{hg,h'} \otimes l_{h,g} \,. \tag{4.12}$$

implementing compatibility of left and right 1-morphisms.

The interpretation of these 2-isomorphisms is illustrated in figure 15. For clarity, we have flattened the surfaces and the attached symmetry defects are omitted: one must imagine symmetry defects attached to $l_{h,g}/r_{g,h}$ pointing out of/into the page.

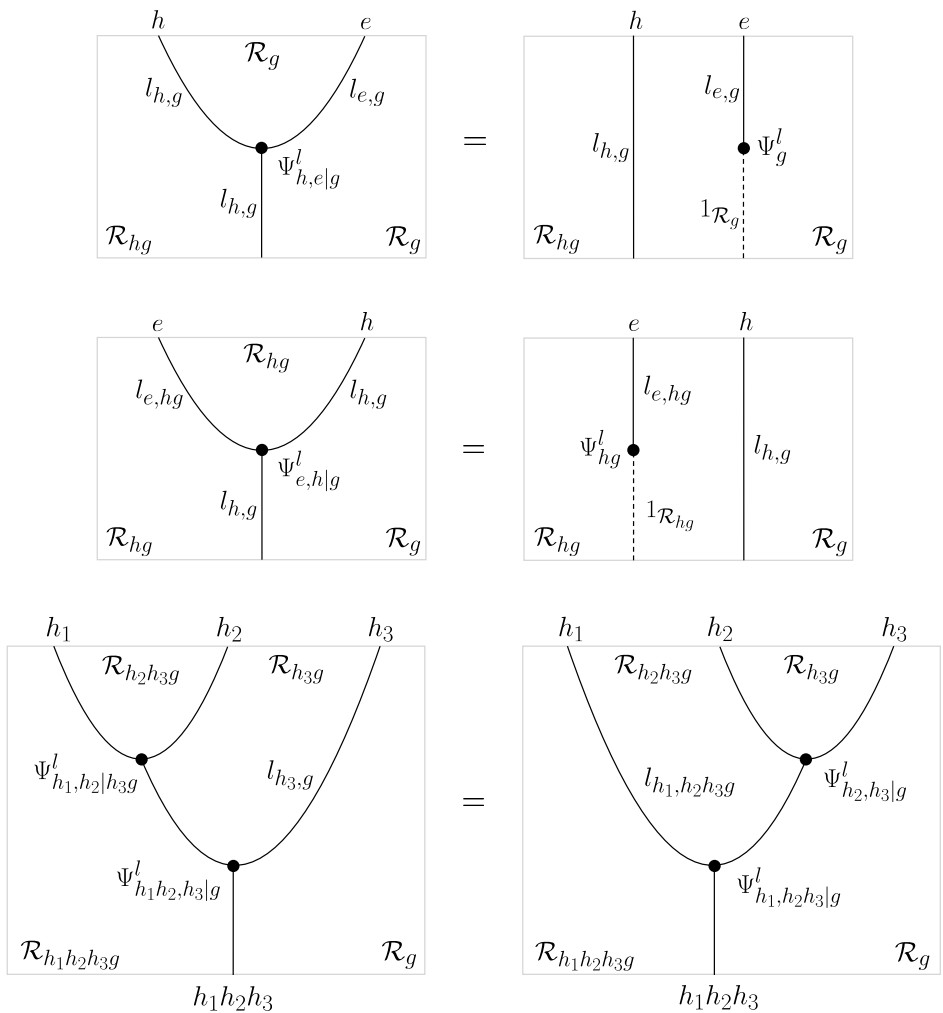

**Figure 16**.

The 2-morphisms must themselves satisfy further compatibility conditions. The first set of conditions may be viewed as a normalisation condition for the 2-isomorphisms in

equation (4.11) and take the form

$$\Psi^l_{h,1|g} = l_{h,g} \otimes \Psi^l_g \qquad \Psi^l_{1,h|g} = \Psi^l_{hg} \otimes l_{h,g}$$
$$\Psi^r_{g|1,h} = r_{g,h} \otimes \Psi^r_g \qquad \Psi^r_{g|h,1} = \Psi^r_{gh} \otimes r_{g,h}\,. \tag{4.13}$$

The second set of conditions ensure compatibility of the 2-isomorphisms with associativity of the fusion of symmetry defects,

$$\Psi^l_{h_1h_2,h_3|g} \circ (\Psi^l_{h_1,h_2|h_3g} \otimes l_{h_3,g}) = \Psi^l_{h_1,h_2h_3|g} \circ (l_{h_1,h_2h_3g} \otimes \Psi^l_{h_2,h_3|g})$$
$$\Psi^r_{g|h_1,h_2h_3} \circ (\Psi^r_{gh_1|h_2,h_3} \otimes r_{g,h_1}) = \Psi^r_{g|h_1h_2,h_3} \circ (r_{gh_1h_2,h_3} \otimes \Psi^r_{g|h_1,h_2}) \tag{4.14}$$

We are using here a shorthand notation where $l_{h,g}$, $r_{g,h}$ denotes the identity 2-isomorphism on the same topological lines. The interpretation of these conditions for the left 1-morphisms is illustrated in figure 16.

The task is now to classify solutions. First, the existence of 2-isomorphisms in (4.10) and (4.11) imply the 1-morphisms $l_{h,g}$, $r_{g,h}$ are weakly invertible and provide explicit inverting 2-isomorphisms. For example

$$\Psi^l_{h^{-1},h|g} \circ \Psi^l_g \quad : \quad (l_{h,g})^{-1} \otimes l_{h,g} \Rightarrow 1_{\mathcal{R}_g}$$
$$\Psi^r_{h,h^{-1}|g} \circ \Psi^r_g \quad : \quad (r_{g,h})^{-1} \otimes r_{g,h} \Rightarrow 1_{\mathcal{R}_g}\,, \tag{4.15}$$

where we define $(l_{h,g})^{-1} := l_{h^{-1},hg}$ and $(r_{g,h})^{-1} := r_{gh,h^{-1}}$. All of the component 1-morphisms may then be constructed from the components $l_{g,e}$, $r_{e,g}$ using combinations of the 2-isomorphisms in equations (4.10) and (4.11). We must then solve these remaining component 1-morphisms and associated 2-isomorphisms.

Let us now use the above 2-isomorphisms to identify $\mathcal{R}_g \cong \mathcal{R}_e =: \mathcal{S}$ for all $g \in G$. We then formulate the remaining conditions on $l_{g,e}$, $r_{e,g}$ using the combination

$$\rho_g := (r_{e,g})^{-1} \circ l_{g,e} \in \mathrm{Hom}(\mathcal{S}, \mathcal{S})\,. \tag{4.16}$$

This represents the topological line arising from the intersection of a symmetry defect $g \in G$ with the topological surface. This is illustrated in figure 17.

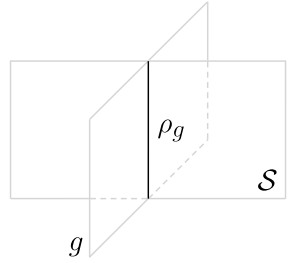

**Figure 17**.

The remaining 2-isomorphisms may be organised into combinations of the form

$$\Psi_e : 1_{\mathcal{S}} \Rightarrow \rho_e \qquad \Psi_{g,h} : \rho_{gh} \Rightarrow \rho_g \circ \rho_h \tag{4.17}$$

and are subject to the conditions

$$\Psi_{e,g} = \Psi_e \otimes \rho_g \qquad \Psi_{g,e} = \rho_g \otimes \Psi_e$$
$$\Psi_{h_1 h_2, h_3} \circ (\Psi_{h_1, h_2} \otimes \rho_{h_3}) = \Psi_{h_1, h_2 h_3} \circ (\rho_{h_1} \otimes \Psi_{h_2, h_3}) \tag{4.18}$$

illustrated in figure 18. We believe this exhausts the remaining 1-morphisms, 2-isomorphisms and the conditions they satisfy.

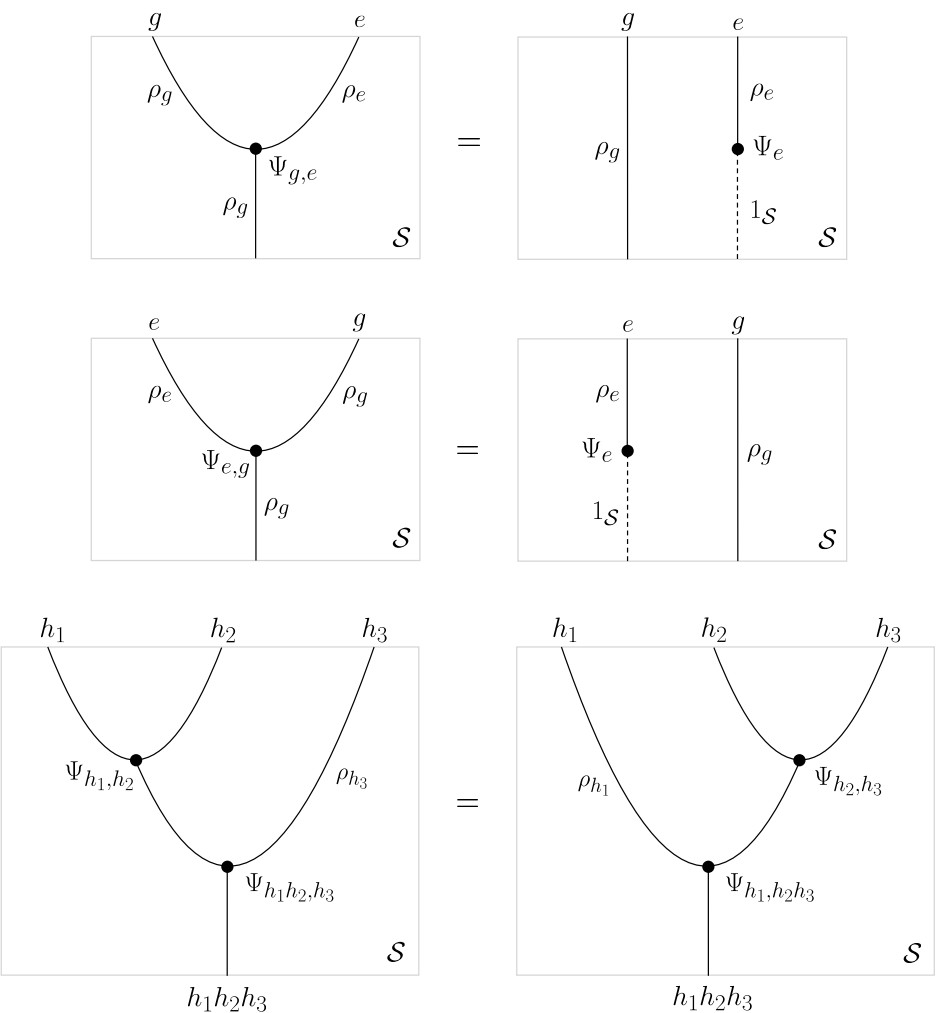

**Figure 18**.

In summary, a topological surface in $\mathcal{T}/G$ is specified by the following data:

1. A set $\mathcal{S} \cong \{1, \ldots, n\} \in 2\text{-Vec}$.

2. A collection of $n \times n$ 2-matrices $\rho_g \in \mathrm{Hom}(\mathcal{S}, \mathcal{S})$.

3. A 2-isomorphism $\Psi_e : 1_{\mathcal{S}} \Rightarrow \rho_e$.

4. 2-isomorphisms $\Psi_{g,h} : \rho_{gh} \Rightarrow \rho_g \circ \rho_h$.

The 2-isomorphisms are subject to the conditions (4.18). This is precisely the data of a 2-representation of the finite group $G$ in 2Vect [18–21].

Let us now summarise the classification of 2-representations following [21]. First, the 2-isomorphisms imply that the 1-morphisms $\rho_g \in \mathrm{Hom}(\mathcal{S}, \mathcal{S})$ are weakly invertible. For example, we have

$$\Psi_{g,g^{-1}} \otimes \Psi_e : 1_{\mathcal{S}} \Rightarrow \rho_g \otimes \rho_{g^{-1}} . \tag{4.19}$$

As a consequence, they endow $\mathcal{S}$ with the structure of a $G$-set

$$\sigma : G \to \mathrm{Aut}(\mathcal{S}) . \tag{4.20}$$

More concretely, using $\mathcal{S} = \{1, \ldots, n\}$, up to isomorphism $\rho_g$ is an $n \times n$ permutation 2-matrix whose non-zero entries are 1-dimensional vector spaces. It is therefore entirely determined by the associated permutation representation $\sigma : G \to S_n$. This is an analogue of topological Wilson lines being labelled by linear representations.

Next, since $\rho_{gh}$ and $\rho_g \circ \rho_h$ are permutation 2-matrices, they have only one non-zero entry per row and column, which is a 1-dimensional vector space. The 2-isomorphisms $\Psi_{g,h}$ are therefore completely determined by a sequence of $n$ phases $\{c_j(g,h) \in U(1)\}$ specifying the isomorphism between the 1-dimensional vector spaces in the $j$-th row. By varying the group elements $g$ and $h$, we can think of this sequence as a 2-cochain

$$c : G \times G \to U(1)^n . \tag{4.21}$$

Condition (4.18) then translates into the 2-cocycle condition

$$c_{\sigma_g^{-1}(j)}(h,k) - c_j(gh,k) + c_j(g,hk) - c_j(g,h) = 0 \tag{4.22}$$

for all group elements $g, h, k \in G$ and $i = 1, \ldots, n$. Thus, $c$ defines a class

$$c \in H^2(G, U(1)^{\mathcal{S}}) , \tag{4.23}$$

where $U(1)^{\mathcal{S}}$ is the abelian group $U(1)^{|\mathcal{S}|}$ supplemented with the structure of a $G$-module via the permutation representation $\sigma$. This is an analogue of Wilson lines in one-dimensional representations of $G$, which are SPT phases $H^1(G, U(1))$.

In summary, topological surfaces in $\mathcal{T}/G$ are 2-representations of the finite group $G$, which can be labelled by pairs $(\mathcal{S}, c)$ consisting of

1. a $G$-set $\mathcal{S}$,

2. a class $c \in H^2(G, U(1)^{\mathcal{S}})$.

The dimension of the 2-representation is $|\mathcal{S}| = n$. This agrees with the classification of 2-representations described in appendix A.3.2. Note that for one-dimensional 2-representations $\mathcal{S} \cong \{1\}$ one specifies a group cohomology class $c \in H^2(G, U(1))$. The associated topological surfaces are constructed by inserting the associated SPT phase supported on a surface in the path integral of $\mathcal{T}/G$.

The 2-representations with $n > 1$ are called condensation defects in the physics literature.[6] A clean physical interpretation of the topological surfaces with $n > 1$ is perhaps not transparent in the current formulation, but this will be remedied momentarily with a more direct construction of simple objects or irreducible 2-representations.

### 4.2.2  Sum, Product, Conjugation

The sum and product of topological surfaces in $\mathcal{T}/G$ are inherited from those of parent topological surfaces in $\mathcal{T}$ and correspond to sum and product in the symmetry category $2\mathsf{Vec}(G)$. They correspond to natural ways in which to combine the data labelling 2-representations fo $G$ and are described in generality below.

First, given two $G$-sets $\mathcal{S}$ and $\mathcal{S}'$, we define their direct sum and tensor product via disjoint union and Cartesian product respectively, i.e.

$$
\begin{aligned}
\mathcal{S} \oplus \mathcal{S}' &= \mathcal{S} \sqcup \mathcal{S}' \\
\mathcal{S} \otimes \mathcal{S}' &= \mathcal{S} \times \mathcal{S}'
\end{aligned}
\tag{4.24}
$$

with the appropriate induced $G$-actions. More concretely, let us write $\mathcal{S} = \{1, \ldots, n\}$ and $\mathcal{S}' = \{1, \ldots, n'\}$ with permutations $\sigma, \sigma' : G \to S_n, S_{n'}$. Then

$$
\begin{aligned}
(\sigma \oplus \sigma')_g(j) &= \begin{cases} \sigma_g(j) & j \in \mathcal{S} \\ \sigma'_g(n-j) + n & j - n \in \mathcal{S}' \end{cases} \\
(\sigma \otimes \sigma')_g(j) &= (\sigma_g(i), \sigma'_g(i')) \qquad j = (i, i') \in \mathcal{S} \times \mathcal{S}'.
\end{aligned}
\tag{4.25}
$$

provide explicit permutation actions on $\mathcal{S} \oplus \mathcal{S}'$ and $\mathcal{S} \otimes \mathcal{S}'$.

Similarly, given two classes $c \in H^2(G, U(1)^{\mathcal{S}})$ and $c' \in H^2(G, U(1)^{\mathcal{S}'})$, we define their direct sum and tensor product

$$
\begin{aligned}
c \oplus c' &\in H^2(G, U(1)^{\mathcal{S} \oplus \mathcal{S}'}) \\
c \otimes c' &\in H^2(G, U(1)^{\mathcal{S} \otimes \mathcal{S}'})
\end{aligned}
\tag{4.26}
$$

by setting for each $g, h \in G$

$$
\begin{aligned}
(c \oplus c')_j(g, h) &= \begin{cases} c_j(g, h) & j \in \mathcal{S} \\ c'_{j-n}(g, h) & j - n \in \mathcal{S}' \end{cases} \\
(c \otimes c')_j(g, h) &= c_i(g, h) + c'_{i'}(g, h) \quad j = (i, i') \in \mathcal{S} \times \mathcal{S}'.
\end{aligned}
\tag{4.27}
$$

It is straightforward to check that these satisfy the appropriate 2-cocycle conditions. Combining these formulae provides a combinatorial definition of the direct sum and fusion of topological surfaces $(\mathcal{S}, c)$ and $(\mathcal{S}', c')$ in $\mathcal{T}/G$.

In addition, the conjugation of a 2-representation $(\mathcal{S}, c)$ may be defined as the 2-representation $(\mathcal{S}, c)^{\#} := (\mathcal{S}, -c)$.

These operations coincide with the corresponding operations in $2\mathsf{Rep}(G)$, as described in appendix A.4.

---

[6]From a mathematical perspective, they are all condensations.

### 4.2.3   1-Morphisms

The 1-morphism categories capture topological lines that sit at junctions between topological surfaces. The 1-morphisms between two topological surfaces in $\mathcal{T}/G$ may be constructed from 1-morphisms between parent topological surfaces together with instructions on how they interact with networks of symmetry defects in $\mathcal{T}$.

Let us first consider the 1-morphism category

$$\mathrm{Hom}_{\mathcal{T}/G}(1, (\mathcal{S}, c)) \,, \tag{4.28}$$

which describes topological lines bounding or screening a topological surface $(\mathcal{S}, c)$. The starting point is then the 1-morphisms of the parent topological surface in $\mathcal{T}$. However, as above, the problem may be reduced to the component 1-morphisms $\mathrm{Hom}_{\mathcal{T}}(1, \mathcal{S})$ with $\mathcal{R}_e \cong \mathcal{S} := \{1, \ldots, n\}$, which are $\mathcal{S}$-graded vector spaces or equivalently collections of vector spaces $\{V_j\}$ index by $j \in \{1, \ldots, n\}$.

The component 1-morphisms must satisfy compatibility conditions involving the topological lines $\rho_g \in \mathrm{Hom}(\mathcal{S}, \mathcal{S})$ arising from the intersection with symmetry defects. In particular, these topological lines may end at the boundary on topological local operators corresponding to 2-isomorphisms in $\mathcal{T}$,

$$\Phi_g: \quad \mathrm{Hom}_{\mathcal{T}}(1, \mathcal{S}) \;\Rightarrow\; \mathrm{Hom}_{\mathcal{T}}(1, \mathcal{S}) \,. \tag{4.29}$$

Concretely, such a 2-isomorphism is a collection of linear maps $\Phi_{g,j}: V_j \to V_{\sigma_g(j)}$ for all $j = 1, \ldots, n$. This is illustrated in figure 19.

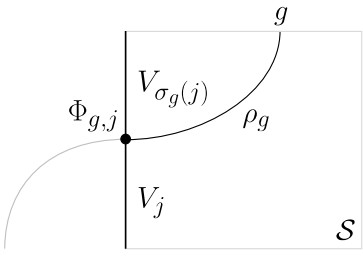

**Figure 19**.

The compatibility with the fusion of symmetry defects intersecting the parent topological surface in $\mathcal{T}$ requires that the 2-morphisms compose as

$$\Phi_{gh,j} \;=\; c_j(g, h) \cdot \Phi_{g,\sigma_h(j)} \circ \Phi_{h,j} \,. \tag{4.30}$$

The additional phase arises due to the same anomaly inflow mechanism described in section 2.3. This condition is illustrated in figure 20.

To summarise, an object in the 1-morphism category $\mathrm{Hom}_{\mathcal{T}/G}(1, (\mathcal{S}, c))$ is determined by the following data:

- A collection of vector spaces $\{V_1, \ldots, V_n\}$ indexed by $\mathcal{S} \cong \{1, \ldots, n\}$,

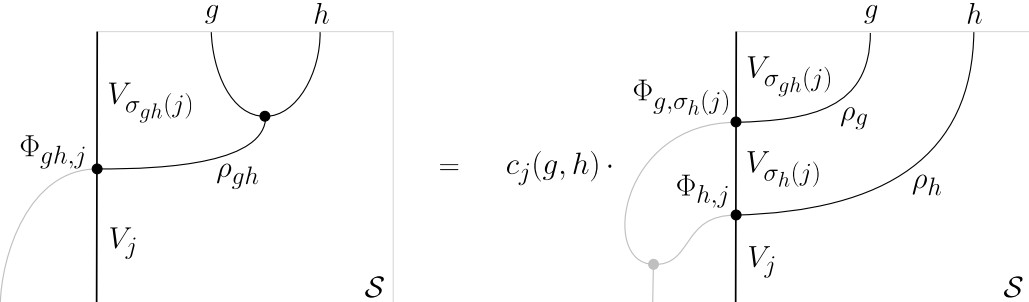

**Figure 20**.

- a collection of linear maps $\Phi_{g,j} : V_j \to V_{\sigma_g(j)}$ satisfying

$$\Phi_{gh,j} = c_j(g,h) \cdot \Phi_{g,j} \circ \Phi_{h,j}.$$

We call this an $\mathcal{S}$-graded projective representation of $G$. Note that this reduces to an ordinary projective representation for a one-dimensional 2-representation or topological surface constructed from an ordinary cohomology class $c \in H^2(G, U(1))$. In this case, the topological line corresponds to a Wilson line whose anomalous transformation is cancelled by anomaly inflow from the topological surface, similar to section 2.3. The general case is a vast generalisation of this picture.

This data of a 1-morphism in $\mathrm{Hom}_{\mathcal{T}/G}(1, (\mathcal{S}, c))$ can be framed more invariantly as follows:

1. A vector bundle $\pi : \mathcal{V} \to \mathcal{S}$.

2. A projective homomorphism $\Phi : G \to \mathrm{Aut}(\mathcal{V})$ satisfying

$$\pi \circ \Phi = \sigma \circ \pi \tag{4.31}$$

    where $\sigma : G \to \mathrm{Aut}(\mathcal{S})$ is the $G$-action on $\mathcal{S}$.

Here, by projective homomorphism we mean $\Phi$ is a group homomorphism up to multiplication by elements $c \in U(1)^{|\mathcal{S}|}$, viewed as bundle automorphisms

$$v \in V_j \mapsto c_j \cdot v. \tag{4.32}$$

This can be seen as a more abstract way to formulate the composition property in (4.30).

Having classified the objects, let us now consider the 2-morphisms in $\mathrm{Hom}_{\mathcal{T}/G}(1, (\mathcal{S}, c))$. They may also be computed from $\mathcal{T}$ and generalise the notion of intertwiners between projective representations to $\mathcal{S}$-graded projective representations. In particular, a 2-morphism between 1-morphisms $(\mathcal{V}, \Phi)$ and $(\mathcal{V}', \Phi)$ is specified by collections of linear maps

$$m_j : V_j \to V_j' \tag{4.33}$$

such that

$$\Phi'_{g,j} \circ m_j = m_{\sigma_g(j)} \circ \Phi_{g,j}. \tag{4.34}$$

They can be regarded as bundles maps $m : \mathcal{V} \to \mathcal{V}'$ commuting with the projective $G$-actions. This clearly reduces to ordinary intertwiners between projective representations for a one-dimensional 2-representation.

In summary, we have found that

$$\operatorname{Hom}_{\mathcal{T}/G}(1, (\mathcal{S}, c)) \; \cong \; \operatorname{Rep}^{(\mathcal{S}, c)}(G) \tag{4.35}$$

is the category of $\mathcal{S}$-graded projective representations of $G$ with cocycle $c$. A more detailed exposition of this category can be found in appendix B.1.

We can now generalise this result to 1-morphisms between arbitrary pairs of topological surfaces, with the result

$$\operatorname{Hom}_{\mathcal{T}/G}((\mathcal{S}, c), (\mathcal{S}, c')) \; = \; \operatorname{Rep}^{(\mathcal{S} \otimes \mathcal{S}', \, c'-c)}(G). \tag{4.36}$$

This may be computed directly by generalising the line of reasoning above, or alternatively using the folding trick to equate the result with 1-morphisms from the trivial topological surface to the tensor product $(\mathcal{S}, c)^{\#} \otimes (\mathcal{S}', c')$. This agrees with the classification of 1-morphisms between 2-representations in $2\operatorname{Rep}(G)$ described in appendix C.1.

Finally, note that for 1-dimensional 2-representations described purely by two group cohomology classes $c, c' \in H^2(G, U(1))$ we have

$$\operatorname{Hom}_{\mathcal{T}/G}(c, c') \; = \; \operatorname{Rep}^{c'-c}(G) \tag{4.37}$$

corresponding to topological Wilson lines in projective representations of $G$, whose anomalous transformations are absorbed by anomaly inflow from the SPT phases $c$, $c'$ on the adjoining surfaces. In particular, the category of endomorphisms of any 1-dimensional 2-representation reproduces the fusion category $\operatorname{Rep}(G)$ of ordinary representations of $G$ and corresponds to genuine topological Wilson lines.

### 4.2.4 Composition of 1-morphisms

The composition of 1-morphisms also has a convenient description in terms of $\mathcal{S}$-graded projective representations. The composition corresponds to functors

$$\operatorname{Rep}^{(\mathcal{S} \otimes \mathcal{S}', \, c'-c)}(G) \; \times \; \operatorname{Rep}^{(\mathcal{S}' \otimes \mathcal{S}'', \, c''-c')}(G) \; \overset{\circ}{\to} \; \operatorname{Rep}^{(\mathcal{S} \otimes \mathcal{S}'', \, c''-c)}(G), \tag{4.38}$$

which is illustrated in figure 21.

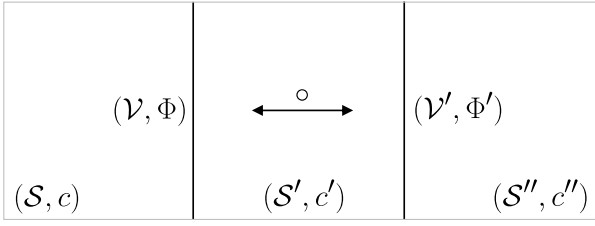

**Figure 21**.

Given an $\mathcal{S} \otimes \mathcal{S}'$-graded representation $(\mathcal{V}, \Phi)$ and an $\mathcal{S}' \otimes \mathcal{S}''$-graded projective representation $(\mathcal{V}', \Phi')$, their composition is an $\mathcal{S} \otimes \mathcal{S}''$-graded projective representation $(\mathcal{V}, \Phi) \circ (\mathcal{V}', \Phi')$ that can be constructed as follows:

- The collection of vector spaces $\mathcal{V} \circ \mathcal{V}'$ is given by

$$(V \circ V')_{(j,j'')} \;=\; \bigoplus_{j' \in \mathcal{S}'} V_{(j,j')} \otimes V'_{(j',j'')} \,. \tag{4.39}$$

- The collection of linear maps $\Phi \circ \Phi'$ is given by

$$(\Phi \circ \Phi')_g \cdot (v \circ v')_{j,j''} \;=\; \bigoplus_{j' \in \mathcal{S}'} (\Phi_g \cdot v_{(j,j')}) \otimes (\Phi'_g \cdot v'_{(j',j'')}) \,. \tag{4.40}$$

It is straightforward to check that this defines an $\mathcal{S} \times \mathcal{S}''$-graded projective representation with 2-cocycle $c'' - c$. Further details on the composition of graded projective representations can be found in appendix B.5.

### 4.2.5 Fusion of 1-morphisms

The fusion of 1-morphisms also has a convenient description in terms of $\mathcal{S}$-graded projective representations. Let us first consider fusion of 1-morphisms of the form

$$\mathrm{Hom}_{\mathcal{T}/G}(1, (\mathcal{S}, c)) \;\times\; \mathrm{Hom}_{\mathcal{T}/G}(1, (\mathcal{S}', c')) \;\xrightarrow{\otimes}\; \mathrm{Hom}_{\mathcal{T}/G}(1, (\mathcal{S}, c) \otimes (\mathcal{S}', c')) \tag{4.41}$$

which are illustrated in figure 22.

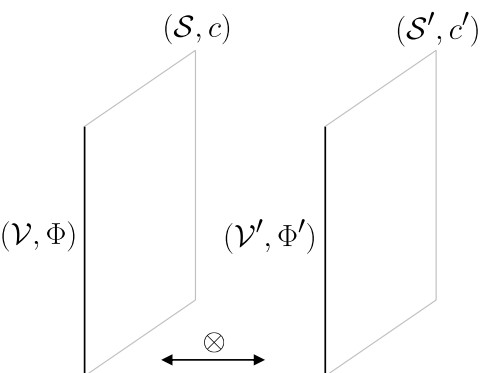

**Figure 22**.

This corresponds to the fusion of graded projective representations,

$$\mathrm{Rep}^{(\mathcal{S},c)}(G) \times \mathrm{Rep}^{(\mathcal{S}',c')}(G) \;\xrightarrow{\otimes}\; \mathrm{Rep}^{(\mathcal{S} \otimes \mathcal{S}', \, c+c')}(G) \,. \tag{4.42}$$

To describe this fusion explicitly, consider an $\mathcal{S}$-graded projective representation $(\mathcal{V}, \Phi)$ and an $\mathcal{S}'$-graded projective representation $(\mathcal{V}', \Phi')$. Their fusion is the $\mathcal{S} \otimes \mathcal{S}'$-graded projective representation $(\mathcal{V}, \Phi) \otimes (\mathcal{V}', \Phi')$ where

- The collection of vector spaces $\mathcal{V} \otimes \mathcal{V}'$ is given by $(V \otimes V)_{(j,j')} = V_j \otimes V'_{j'}$,

- The collection of linear maps $\Phi \otimes \Phi'$ is given by $(\Phi \otimes \Phi')_{g,(j,j')} = \Phi_{g,j} \otimes \Phi'_{g,j'}$.

This may then be generalised to fusion of 1-morphisms between arbitrary pairs of topological surfaces, for example, using the folding trick.

As a consistency check, consider the fusion of 1-endomorphisms of the trivial surface (or any one-dimensional 2-representation), which are ordinary topological Wilson lines. It is clear that this reproduces the tensor product of ordinary representations of $G$, which is the correct fusion of topological Wilson lines in $\mathcal{T}/G$. More details on sums and fusions of graded projective representations can be found in appendix B.2.

### 4.3 Simple Objects

Let us now consider simple objects in the symmetry category $2\mathrm{Rep}(G)$ of $\mathcal{T}/G$, which correspond to irreducible 2-representations of $G$. This uncovers an alternative mathematical structure that sheds light on the physical interpretation of topological surfaces corresponding to 2-representations of dimension greater than one in terms of condensation defects.

#### 4.3.1 Irreducible 2-representations

The decomposition of topological surfaces into simple topological surfaces corresponds to the decomposition of a 2-representation $(\mathcal{S}, c)$ into irreducible 2-representations.

First, we may decompose any $G$-set as a union of disjoint orbits $\mathcal{S} = \sqcup_\alpha \mathcal{O}_\alpha$, which form transitive $G$-sets by definition. Second, there is an associated decomposition $c = \oplus_\alpha c_\alpha$ into classes on each orbit according to the isomorphism

$$H^2(G, U(1)^S) \cong \bigoplus_\alpha H^2(G, U(1)^{\mathcal{O}_\alpha}). \tag{4.43}$$

Consequently, any 2-representation $(\mathcal{S}, c)$ of $G$ can be decomposed as a direct sum

$$(\mathcal{S}, c) \cong \bigoplus_\alpha (\mathcal{O}_\alpha, c_\alpha). \tag{4.44}$$

of simple objects, where the latter are labelled by pairs $(\mathcal{O}, c)$ consisting of

1. a $G$-orbit $\mathcal{O}$,

2. a class $c \in H^2(G, U(1)^{\mathcal{O}})$,

and correspond to irreducible 2-representations of $G$. More details on the latter can be found in appendix A.6.

#### 4.3.2 Induction

The irreducible 2-representations may also be obtained by induction of one-dimensional 2-representations of subgroups $H \subset G$, leading to a more direct physical construction of topological surfaces that clarifies their relationship to condensation defects for the topological Wilson lines.

The first observation is that $G$-orbits $\mathcal{O}$ are in 1-1 correspondence with conjugacy classes of subgroups $H \subset G$. Representative subgroups of the conjugacy class are stabilisers of elements in the orbit. Any irreducible 2-representation $(\mathcal{O}, c')$ is then the induction of a one-dimensional 2-representation of $H \subset G$ labelled by a group cohomology class $c \in H^2(H, U(1))$ [21]. Let us write this correspondence as

$$(\mathcal{O}, c') = \mathrm{Ind}_H^G(c). \tag{4.45}$$

Explicitly, the induction works as follows:

1. Given the subgroup $H \subset G$, we can obtain a $G$-orbit by setting $\mathcal{O} = G/H$. The permutation action of $G$ on $\mathcal{O}$ can be constructed by picking a system $\{r_1, \ldots, r_n\}$ of representatives of left $H$-cosets $r_j H$ and defining $\sigma : G \to S_n$ by

$$g \cdot r_j \in r_{\sigma_g(j)} H. \tag{4.46}$$

In addition, this allows us to define little group elements

$$\ell_{g,j} := r_{\sigma_g(j)}^{-1} \cdot g \cdot r_j \in H. \tag{4.47}$$

2. Utilising Shapiro's isomorphism

$$H^2(H, U(1)) \cong H^2(G, U(1)^{\mathcal{O}}), \tag{4.48}$$

we can construct a class $c' \in H^2(G, U(1)^{\mathcal{O}})$ from $c$ by setting

$$c_j'(g_1, g_2) := c\big(\ell_{g_1, \sigma_{g_1}^{-1}(j)}, \ell_{g_2, \sigma_{g_1 g_2}^{-1}(j)}\big) \tag{4.49}$$

where $\ell_{g_1, \sigma_{g_1}^{-1}(j)}$ and $\ell_{g_2, \sigma_{g_1 g_2}^{-1}(j)}$ are little group elements associated to $g_1$, $g_2$.

One can check that the 2-representation $(\mathcal{O}, c')$ obtained in this way depends on the coset representatives $r_j$ only up to isomorphism. More details on the induction of 2-representations can be found in appendix A.5.

In summary, simple objects may alternatively be labelled by pairs $(H, c)$ consisting of

- a subgroup $H \subset G$,

- a class $c \in H^2(H, U(1))$,

and correspond to irreducible 2-representations of $G$.

In computing fusion and 1-morphisms of simple objects below, we will encounter a version of Mackey's decomposition formula for the restriction of induced one-dimensional 2-representations to other subgroups. Namely,

$$(\mathrm{Res}_H^G \circ \mathrm{Ind}_K^G)(c) = \bigoplus_{[g] \in H \backslash G / K} \mathrm{Ind}_{H \cap K^g}^H(c^g) \tag{4.50}$$

where the summation is over representatives $g$ of double cosets, $K^g \equiv gKg^{-1}$, and the 1-dimensional 2-representation $c^g$ of $K^g$ is defined by

$$(c^g)(x_1, x_2) := c(x_1^g, x_2^g) \tag{4.51}$$

where $x_j^g := g^{-1}x_j g \in K$. On the right-hand side of equation (4.50) we view $c^g$ as a class on $H \cap K^g$ and therefore ought to write $\mathrm{Res}_{H \cap K^g}^{K^g}(c^g)$ instead of simply $c^g$. In order to avoid cumbersome notation, we will leave this implicit in what follows.

Finally, we note that the induction of 2-representations reflects an alternative physical construction of simple topological surfaces. They correspond to topological surfaces in $\mathcal{T}/G$ where the bulk gauge symmetry is broken to $H \subset G$ by a partial Dirichlet boundary condition, supplemented by a two-dimensional SPT phase $c \in H^2(H, U(1))$ for the unbroken gauge symmetry. This interpretation of full condensation defects with $H = 1$ and full Dirichlet boundary conditions appeared in [4].

### 4.3.3  Fusion of Simple Objects

We now consider the fusion ring generated by decomposing products of simple topological surfaces as sums of simple topological surfaces. While the structure follows from the general construction of direct sums and tensor products of 2-representations above, we will also provide an explicit description in terms of induced 2-representations.

Let us then introduce a basis of $G$-orbits $\mathcal{O}_\alpha$ and denote simple objects by $(\mathcal{O}_\alpha, c)$ where $c \in H^2(G, U(1)^{\mathcal{O}_\alpha})$. The fusion rules will take the form

$$(\mathcal{O}_\alpha, c) \otimes (\mathcal{O}_\beta, c') = \bigoplus_\gamma n_{\alpha\beta}^\gamma \cdot (\mathcal{O}_\gamma, c''), \tag{4.52}$$

where the coefficients $n_{\alpha\beta}^\gamma$ can be determined as follows:

- The fusion of underlying orbits corresponds to the Cartesian product. The coefficients $n_{\alpha\beta}^\gamma \in \mathbb{Z}_+$ are therefore determined by decomposing the cartesian product of orbits as a disjoint union

$$\mathcal{O}_\alpha \times \mathcal{O}_\beta = \bigsqcup_\gamma n_{\alpha\beta}^\gamma \cdot \mathcal{O}_\gamma. \tag{4.53}$$

- The associated cohomology classes $c''$ are determined using the sum and product defined in (4.27). Concretely, if $\mathcal{O}_\alpha = \{1, \ldots, n_\alpha\}$ and $\mathcal{O}_\beta = \{1, \ldots, n_\beta\}$, then the class $c''$ on the orbit $\mathcal{O}_\gamma \subset \mathcal{O}_\alpha \times \mathcal{O}_\beta$ is given by the formula

$$c''_{(j,j')} = c_j + c'_{j'} \tag{4.54}$$

for each element $(j, j') \in O_\gamma$.

This provides a full description of the fusion structure of simple topological surfaces.

However, it is also useful to reformulate the fusion structure in terms of induced 2-representations. From this perspective, the simple objects are labelled by conjugacy classes of subgroups $H_\alpha \subset G$ together with classes $c_\alpha \in H^2(H_\alpha, U(1))$.

First, the decomposition of the Cartesian product of $G$-orbits is equivalent to the double coset decomposition formula

$$G/H_\alpha \times G/H_\beta = \bigsqcup_{[g] \in H_\alpha \backslash G / H_\beta} G / (H_\alpha \cap H_\beta^g), \tag{4.55}$$

where the summation is over representatives $g$ of double $H_\alpha$-$H_\beta$-cosets. The cohomology classes decompose such that the class associated to the summand $G/(H_\alpha \cap H_\beta^g)$ is

$$c_\alpha \otimes c_\beta^g \ \in \ H^2(H_\alpha \cap H_\beta^g, U(1)), \tag{4.56}$$

where restriction of arguments to the intersection is understood.

In summary, the fusion of simple topological surfaces is

$$(H_\alpha, c_\alpha) \otimes (H_\beta, c_\beta) \ = \ \bigoplus_{[g] \in H_\alpha \backslash G / H_\beta} (H_\alpha \cap H_\beta^g, \, c_\alpha \otimes c_\beta^g), \tag{4.57}$$

where an appropriate restriction of the classes to $H_\alpha \cap H_\beta^g$ is understood implicitly on the right-hand side. This provides another concrete method to compute the fusion structure and agrees with the fusion of irreducible 2-representations described in appendix A.6.

The fusion formula may also be viewed as an application of Mackey's decomposition formula together with the push-pull formula for induced 2-representations via the following manipulations,

$$
\begin{aligned}
(H_\alpha, c_\alpha) \otimes (H_\beta, c_\beta) \ &\cong \ \mathrm{Ind}_{H_\alpha}^G(c_\alpha) \otimes \mathrm{Ind}_{H_\beta}^G(c_\beta) \\
&= \ \mathrm{Ind}_{H_\alpha}^G(c_\alpha \otimes \mathrm{Res}_{H_\alpha}^G \mathrm{Ind}_{H_\beta}^G(c_\beta)) \\
&= \ \bigoplus_{[g] \in H_\alpha \backslash G / H_\beta} \mathrm{Ind}_{H_\alpha}^G(c_\alpha \otimes \mathrm{Ind}_{H_\alpha \cap H_\beta^g}^{H_\alpha}(c_\beta^g)) \\
&= \ \bigoplus_{[g] \in H_\alpha \backslash G / H_\beta} \mathrm{Ind}_{H_\alpha \cap H_\beta^g}^G(c_\alpha \otimes c_\beta^g) \\
&\cong \ \bigoplus_{[g] \in H_\alpha \backslash G / H_\beta} (H_\alpha \cap H_\beta^g, \, c_\alpha \otimes c_\beta^g)
\end{aligned}
\tag{4.58}
$$

where the appropriate restrictions on the $c$'s are understood implicitly in the final line.

We now consider some special cases. The simplest is perhaps the fusion of one-dimensional 2-representations or pure SPT phase topological surfaces, which is addition of the associated cohomology classes,

$$c \otimes c' = c + c'. \tag{4.59}$$

This is consistent with a direct path integral argument by inserting SPT phases supported on surfaces in $\mathcal{T}/G$. A generalisation is the fusion of a one-dimensional 2-representation (or pure SPT phase) with a general irreducible 2-representation or condensation defect. The result is that

$$(H, c) \otimes c' \ = \ (H, \, c + \mathrm{Res}_H^G(c')). \tag{4.60}$$

This formula reflects the fact that the gauge symmetry is broken to a subgroup $H \subset G$ on the topological surface and therefore fusion with another SPT phase $c'$ only detects the restriction to the subgroup $H \subset G$.

We may also restrict attention to the fusion of simple topological surfaces corresponding to normal subgroups. This produces a summation of the form

$$(H_\alpha, c_\alpha) \otimes (H_\beta, c_\beta) \ = \ \bigoplus_{[g] \in H_\alpha \backslash G / H_\beta} (H_\alpha \cap H_\beta, \, c_\alpha \otimes c_\beta^g) \tag{4.61}$$

with a common subgroup appearing in each summand. This reflects the fact that fusion of topological surfaces breaking the gauge symmetry to normal subgroups $H_\alpha$, $H_\beta$ will break the gauge symmetry to the intersection $H_\alpha \cap H_\beta$. In particular,

$$(H, c) \otimes (H, c') = \bigoplus_{[g] \in G/H} (H, c \otimes (c')^g) \tag{4.62}$$

when fusing 2-representations induced from the same normal subgroup $H \subset G$.

Finally, if $G$ is abelian, the fusion structure simplifies dramatically with a single summand appearing in the fusion of simple objects

$$(H_\alpha, c_\alpha) \otimes (H_\beta, c_\beta) = n_{\alpha\beta} \cdot (H_\alpha \cap H_\beta, c_\alpha \otimes c_\beta), \tag{4.63}$$

where the coefficient on the right-hand side is

$$n_{\alpha\beta} \equiv |H_\alpha \backslash G / H_\beta| = \frac{|H_\alpha \cap H_\beta|}{|H_\alpha| \cdot |H_\beta|} \cdot |G|. \tag{4.64}$$

by the Cauchy-Frobenius lemma. This again reflects the fact that fusion of topological surfaces breaking the gauge symmetry to abelian subgroups $H_\alpha$, $H_\beta$ will break the gauge symmetry to the intersection $H_\alpha \cap H_\beta$.

### 4.3.4 1-morphisms

We start by considering topological lines on which a simple topological surface $(O, c')$ may end. They are captured by the 1-morphism category

$$\mathrm{Hom}_{\mathcal{T}/G}(1, (\mathcal{O}, c')) = \mathrm{Rep}^{(\mathcal{O}, c')}(G), \tag{4.65}$$

which is the category of $\mathcal{O}$-graded projective representations of $G$ with 2-cocycle $c \in H^2(G, U(1)^{\mathcal{O}})$ by specialising the general result (4.35).

It is also useful to reformulate this result in terms of induced 2-representations where we label the simple topological surface by a pair $(H, c)$ with $(\mathcal{O}, c') = \mathrm{Ind}_H^G(c)$. As described in appendix B.3, the 1-morphisms may be obtained by an analogous process of induction. In particular, any $\mathcal{O}$-graded projective representation $(\mathcal{V}, \Phi)$ of $G$ may be obtained as the induction of an ordinary projective representation $\varphi : H \to \mathrm{GL}(W)$ of $H$ with cocycle $c \in H^2(H, U(1))$. Let us write this correspondence as

$$(\mathcal{V}, \Phi) \cong \mathrm{Ind}_H^G(W, \varphi). \tag{4.66}$$

Moreover, 2-morphisms or morphisms in the category of 1-morphisms are obtained by induction of intertwiners between projective representations.

In summary,

$$\mathrm{Hom}_{\mathcal{T}/G}(1, (H, c)) \cong \mathsf{Rep}^c(H) \tag{4.67}$$

is the category of projective representations of $H$ with 2-cocycle $c \in H^2(H, U(1))$. The objects may be regarded as Wilson lines for the unbroken gauge symmetry $H \subset G$,

whose anomalous transformation is cancelled by anomaly inflow from the attached two-dimensional SPT phase surface defect, as described in section 2.3. In particular, endomorphisms of the identity object reproduces ordinary representations

$$\mathrm{End}_{\mathcal{T}/G}(1) = \mathsf{Rep}(G) \tag{4.68}$$

corresponding to genuine topological Wilson lines.

We can now generalise this result to 1-morphisms between pairs of simple topological surfaces by specialising (4.36) or using the folding trick together with the fusion described above. The result is that

$$\mathrm{Hom}_{\mathcal{T}/G}\big((H,c),\,(H',c')\big) \;=\; \bigoplus_{[g] \in H \backslash G / H'} \mathsf{Rep}^{(c')^g - c}\big(H \cap (H')^g\big)\,. \tag{4.69}$$

This agrees with the classification of 1-morphisms between irreducible 2-representations described in appendix C.1.

We are particularly interested in 1-endomorphisms describing topological line operators on a simple topological surface,

$$\mathrm{End}_{\mathcal{T}/G}(H,c) \;=\; \bigoplus_{[g] \in H \backslash G / H} \mathsf{Rep}^{c^g - c}\big(H \cap H^g\big)\,. \tag{4.70}$$

In the special case when $H \subset G$ is normal and $c = 0$, this becomes

$$\mathrm{End}_{\mathcal{T}/G}(H) \;=\; \bigoplus_{[g] \in G / H} \mathsf{Rep}(H)\,. \tag{4.71}$$

Note that this clearly contains $\mathsf{Rep}(H)$ as a sub-category. Furthermore, by considering only trivial representations in each summand, it also contains $\mathsf{Vect}(G/H)$ as a sub-category. The fusion structure on these sub-categories stems from the composition of 1-morphisms, which turns $\mathrm{End}_{\mathcal{T}/G}(H)$ into a fusion category. We will expand on this in more detail below.

Physically, this captures the fact that the simple topological surface breaks the gauge symmetry to $H \subset G$, leaving a global symmetry $G/H$. In other words, the 1-endomorphisms in $\mathcal{T}/G$ contain topological Wilson lines for $H$ and symmetry generators for $G/H$ that arise from symmetry generators in $\mathcal{T}$ ending on the surface.

These symmetries exhibit a mixed 't Hooft anomaly of the type discussed in [17], which is straightforward to describe explicitly when $H$ is abelian and $\mathsf{Rep}(H) = \mathsf{Vec}(\widehat{H})$. If we consider the exact sequence

$$1 \;\to\; H \;\to\; G \;\to\; G/H \;\to\; 1\,, \tag{4.72}$$

with extension class $e \in H^2(G/H, H)$ then the 't Hooft anomaly takes the form

$$\int_{X_3} \widehat{\mathbf{h}} \cup e(\mathbf{a}) \tag{4.73}$$

with background fields $\widehat{\mathbf{h}} \in H^1(X_3, \widehat{H})$ and $\widehat{\mathbf{a}} : X_3 \to B(G/H)$.

### 4.3.5 Composition of 1-morphisms

We restrict our attention to composition of 1-endomorphisms of a simple object associated to a normal subgroup $H \subset G$ and $c = 0$. The general case is described in appendix C.3.

As above, we can think of 1-morphisms as sums of pairs $([g], \varphi)$ consisting of a coset $[g] \in G/H$ and a topological Wilson line in a representation $\varphi : H \to \mathrm{GL}(W)$ of $H$. The composition of these 1-endomorphisms is

$$([g_1], \varphi_1) \circ ([g_2], \varphi_2) = \left( [g_1] \cdot [g_2], \, \varphi_1 \otimes (\varphi_2)^{g_1} \right) \tag{4.74}$$

where the product structure on $G/H$ is understood and $\varphi^g : H \to \mathrm{GL}(W)$ is the conjugated representation defined by

$$\varphi^g(h) = \varphi(g^{-1}hg). \tag{4.75}$$

This endows the sub-categories $\mathsf{Rep}(H)$ and $\mathsf{Vect}(G/H)$ separately with their obvious fusion structure. However, due to the appearance of the twist $(\varphi_2)^{g_1}$ in (4.74), the fusion category $\mathrm{End}_{\mathcal{T}/G}(H)$ is not in general equivalent to the product $\mathsf{Vect}(G/H) \times \mathsf{Rep}(H)$.

### 4.3.6 Fusion of 1-morphisms

The general case of fusions of 1-morphisms between simple topological surfaces is presented in appendix C.2. Here we restrict ourselves to an instructive example that illustrates the condensation nature of the topological surfaces.

Consider the fusion of the identity 1-endomorphism $\mathrm{Id}_{(H,c)} \in \mathrm{End}(H, c)$ of a simple topological surface labelled by $(H, c)$ and a general 1-endomorphism of the identity surface $\varphi \in \mathrm{End}(1) = \mathsf{Rep}(G)$. This is the fusion of a topological surface with a topological Wilson line as illustrated in figure 23. We find that

$$\mathrm{Id}_{(H,c)} \otimes \varphi = \varphi|_H, \tag{4.76}$$

which captures precisely the condensation nature of the topological surfaces. In particular, for the full condensation defect with $H = 1$, all of the topological Wilson lines condense on the surface.

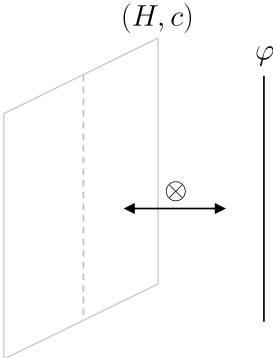

**Figure 23**.

### 4.4 Examples

#### 4.4.1 $G = \mathbb{Z}_2$

Let us consider the simplest example $G = \mathbb{Z}_2$. The theory $\mathcal{T}$ has symmetry category $2\mathsf{Vec}(\mathbb{Z}_2)$ with two simple objects $1$, $s$ with fusion $s \otimes s = 1$ and non-trivial 1-morphism categories $\mathrm{Hom}_{\mathcal{T}}(1,1) = \mathrm{Hom}_{\mathcal{T}}(s,s) = \mathrm{Vect}$.

Upon gauging the symmetry $G$, the resulting theory $\mathcal{T}/G$ has topological Wilson lines generating the Pontryagin dual $\mathbb{Z}_2$ 1-form symmetry. However, there is also condensation surface defect for the topological Wilson lines and the full symmetry category is the fusion 2-category $2\mathsf{Rep}(\mathbb{Z}_2)$.

The simple objects are irreducible 2-representations. There are only two $\mathbb{Z}_2$-orbits: the trivial orbit with stabiliser $\mathbb{Z}_2$, and the maximal orbit with trivial stabiliser. There are no SPT phases because $H^2(\mathbb{Z}_2, U(1)) = 0$. Let us denote the corresponding simple objects by $1$, $X$, respectively. The physical interpretation of these objects is clear: $1$ is the identity surface, while $X$ is the condensation defect for the $\mathbb{Z}_2$ 1-form symmetry.

Their fusion is determined by

$$X \otimes X = 2X \,, \tag{4.77}$$

which follows from the fact that the cartesian product of two maximal orbits decomposes as a sum of two orbits. The 1-morphism categories are

$$\mathrm{End}_{\mathcal{T}/G}(1) = \mathsf{Rep}(\mathbb{Z}_2) \tag{4.78}$$
$$\mathrm{Hom}_{\mathcal{T}/G}(1,X) = \mathrm{Hom}_{\mathcal{T}/G}(X,1) = \mathrm{Vect} \tag{4.79}$$
$$\mathrm{End}_{\mathcal{T}/G}(X) = \mathsf{Vect}(\mathbb{Z}_2) \,. \tag{4.80}$$

A diagrammatic representation of $2\mathsf{Rep}(\mathbb{Z}_2)$ can be found in appendix C.4.1[7].

#### 4.4.2 $G = \mathbb{Z}_2 \times \mathbb{Z}_2$

As a slightly more involved example, let us consider $G = \mathbb{Z}_2 \times \mathbb{Z}_2$. The theory $\mathcal{T}$ has symmetry category $2\mathsf{Vec}(\mathbb{Z}_2 \times \mathbb{Z}_2)$ with four simple objects $1$, $s_+$, $s_0$, $s_-$ with fusion

$$\begin{aligned} s_+^2 = s_0^2 = s_-^2 &= 1 \\ s_\pm \cdot s_0 &= s_\mp \\ s_+ \cdot s_- &= s_0 \end{aligned} \tag{4.81}$$

and non-trivial 1-morphisms $\mathrm{Hom}(1,1) = \mathrm{Hom}(s_\pm, s_\pm) = \mathrm{Hom}(s_0, s_0) = \mathrm{Vect}$.

Upon gauging the symmetry $G$, the symmetry category of the resulting theory $\mathcal{T}/G$ is the fusion 2-category $2\mathsf{Rep}(\mathbb{Z}_2 \times \mathbb{Z}_2)$. There are now five orbits, corresponding to the five subgroups of $\mathbb{Z}_2 \times \mathbb{Z}_2$ acting as stabilizers of the orbits: the group $G = \mathbb{Z}_2 \times \mathbb{Z}_2$ itself, three subgroups of order 2, and the trivial subgroup. In particular, the trivial orbit with stabilizer $G = \mathbb{Z}_2 \times \mathbb{Z}_2$ can be supplemented by an SPT phase

$$\alpha \in H^2(\mathbb{Z}_2 \times \mathbb{Z}_2, U(1)) \cong \mathbb{Z}_2 \,. \tag{4.82}$$

---

[7]Note that in appendix C.4.1 we used the labels **1** and **2** for the simple objects of $2\mathsf{Rep}(\mathbb{Z}_2)$.

Let us denote the corresponding simple objects by $1^\alpha$, $X_i$ and $Y$ respectively (where $i = 1, 2, 3$). Their fusion is determined by

$$1^\alpha \otimes 1^\beta = 1^{\alpha+\beta}, \tag{4.83}$$

$$X_i \otimes X_j = \begin{cases} 2\,X_i & \text{if } i = j, \\ Y & \text{if } i \neq j, \end{cases} \tag{4.84}$$

$$X_i \otimes Y = 2Y, \tag{4.85}$$

$$Y \otimes Y = 4Y. \tag{4.86}$$

A diagrammatic representation of $2\mathrm{Rep}(\mathbb{Z}_2 \times \mathbb{Z}_2)$ can be found in appendix C.4.3[8].

## 5 Three dimensions: split 2-groups

We now generalise section 4 to gauging a finite 2-group symmetry in three dimensions. We emphasise that we consider finite 2-group symmetries with finite 0-form and 1-form components. A common source of such symmetries in dynamical gauge theories is discussed below in section 6. Aspects of 2-group global symmetries (typically with continuous 0-form components) and their 't Hooft anomalies have been investigated in [36–45].

In this paper, we focus on gauging a split finite 2-group with vanishing Postnikov class, returning more general finite 2-groups to a subsequent paper. Such a 2-group is specified by a finite 0-form symmetry group $H$, a finite abelian 1-form symmetry group $A$ and a homomorphism $\varphi : A \to \mathrm{Aut}\,H$. By a natural extension of the notation for a split extension or semi-direct product group, we denote this by

$$G := A[1] \rtimes H. \tag{5.1}$$

As for a semi-direct product group in two dimensions in section 3, the 0-form and 1-form components of an anomaly free split 2-group may be gauged independently. This generates the commuting square of symmetry categories illustrated in figure 24, where the arrows denote gauging the labelled symmetry.

The upshot is that the symmetry category resulting from gauging the split 2-group symmetry is the fusion 2-category $2\mathrm{Rep}(G)$ of 2-representations of $G$. In particular, we will enumerate the topological surfaces and show that they are in 1-1 correspondence with 2-representations of $G$. We will also consider in detail the fusion, 1-morphisms, fusion or 1-morphisms and composition of 1-morphisms. The structure of this category for general finite 2-groups has been elaborated in [19].

In this section, we will actually begin our exploration from the theory $\widehat{\mathcal{T}} = \mathcal{T}/A[1]$ in figure 24 obtained by gauging the 1-form component of the 2-group symmetry. This has a finite 0-form symmetry taking the form of a semi-direct product

$$\widehat{G} = \widehat{A} \rtimes H \tag{5.2}$$

---

[8]Note that in appendix C.4.3 we used the labels $\mathbf{1}^\alpha$, $\mathbf{2}_+$, $\mathbf{2}_0$, $\mathbf{2}_-$ and $\mathbf{4}$ to denote the simple objects of $2\mathrm{Rep}(\mathbb{Z}_2 \times \mathbb{Z}_2)$.

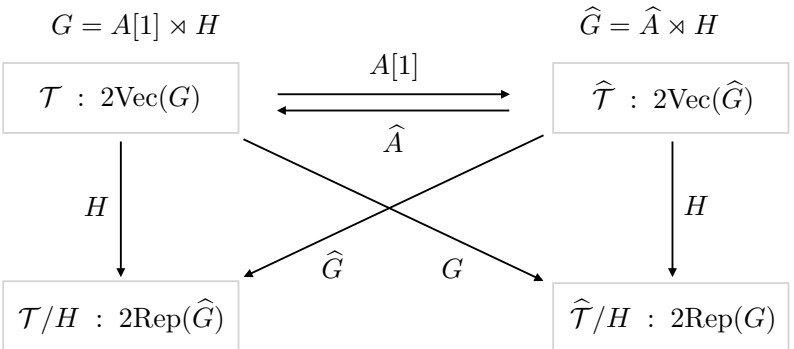

**Figure 24**.

and the associated symmetry category $2\mathsf{Vec}(G)$ has been discussed at the beginning of section 4. This allows us to generalise the results of section 4 in two directions by gauging subgroups of the symmetry $\widehat{G}$:

- First, gauging $\widehat{A} \subset \widehat{G}$ recovers the original theory $\mathcal{T}$ with finite 2-group symmetry $G = A[1] \rtimes H$. This allows us to derive the full structure of the symmetry category $2\mathrm{Vect}(G)$ associated to a split 2-group symmetry.

- Second, gauging $H \subset \widehat{G}$ is equivalent to gauging the entire 2-group symmetry $G$ of $\mathcal{T}$ and results in the symmetry category $2\mathrm{Rep}(G)$ of 2-representations of $G$.

The two generalisations can be treated in a similar way, by means of a combination of arguments in sections 3 and 4.

## 5.1 Split 2-group symmetry

We first consider the symmetry category $2\mathrm{Vec}(G)$ of $\mathcal{T}$. This is essentially a higher analogue of a semi-direct product combining contributions from $2\mathrm{Vect}(H)$ and $2\mathrm{Rep}(\widehat{A})$ obtained by gauging the symmetry $\widehat{A}$. The derivation is therefore a mild generalisation of section 4 taking into account the additional action of $H$ on $\hat{A}$, and therefore we do not perform the derivation in detail. To our knowledge, the presence of condensation defects arising from $2\mathrm{Rep}(\widehat{A})$ has not been emphasised in the literature.

Let us start by enumerating the simple topological surfaces by combining simple objects of $2\mathrm{Vect}(H)$ and $2\mathrm{Rep}(\widehat{A})$. The simple topological surfaces are therefore labelled by pairs

$$((\mathcal{O}, c), h) \tag{5.3}$$

where $(\mathcal{O}, c)$ is an irreducible 2-representation of $\widehat{A}$ and $h \in H$. Here we have chosen to label the irreducible 2-representation as in (4.44), namely:

1. $\mathcal{O}$ is a $\widehat{A}$-orbit,

2. $c \in H^2(\widehat{A}, U(1)^{\mathcal{O}})$.

The fusion of simple topological objects is determined by fusion in $2\text{Vect}(H)$ and $2\text{Rep}(\widehat{A})$, together with the natural action of $H$ on 2-representations of $\widehat{A}$. In particular, let $\sigma : \widehat{A} \to \text{Aut}(\mathcal{O})$ denote the transitive action of $\widehat{A}$ on the orbit $\mathcal{O}$. For any $h \in H$, we then define $\mathcal{O}^h$ as the $\widehat{A}$-set with the same underlying set but shifted action $\sigma^h_\chi := \sigma_{\chi^h}$. The higher analogue of the semi-direct product fusion rule is then

$$((\mathcal{O}, c), h) \otimes ((\mathcal{O}', c'), h') = ((\mathcal{O} \otimes \mathcal{O}'^h, c \otimes c'^h), hh'). \tag{5.4}$$

Notice that the topological surface on the right-hand side is not necessarily simple. It can be decomposed into simple objects as outlined in section 4.3.3, either using (4.52) or the subsequent discussion on induced representations.

We now briefly summarise 1-morphisms. They again combine the result (4.2) for 1-morphisms in $2\text{Vect}(H)$ and (4.36) for 1-morphisms in $2\text{Rep}(\widehat{A})$, taking into account the $H$-action on $\widehat{A}$. We obtain the result

$$\text{Hom}_{\mathcal{T}/\widehat{A}}(((\mathcal{O}, c), h), ((\mathcal{O}', c'), h')) = \begin{cases} \text{Rep}^{(\mathcal{O} \otimes \mathcal{O}'^h, c'^h - c)}(\widehat{A}) & h = h' \\ 0 & h \neq h', \end{cases} \tag{5.5}$$

which can again be understood as Wilson lines in projective representations of $\widehat{A}$, whose anomalous transformation is compensated by inflow from the topological surfaces. Composition and fusions of 1-morphisms are determined by those in $2\text{Vec}(H)$ and $2\text{Rep}(\widehat{A})$ and the $H$-action on $\widehat{A}$.

## 5.2 Gauging a split 2-group

We now consider gauging the finite 0-form symmetry group $H \subset \widehat{G}$ of $\widehat{\mathcal{T}}$. This is tantamount to gauging the entire 2-group $G = A[1] \rtimes H$ of $\mathcal{T}$. Following the line of reasoning and combining arguments from sections 3 and 4, we will show here that this results in a theory with the symmetry category $2\text{Rep}(G)$, the fusion 2-category of 2-representations of the 2-group $G$.

### 5.2.1 Objects

Since the arguments are a combination of those in sections 3 and 4, we will be brief. We start from $\widehat{\mathcal{T}} = \mathcal{T}/A[1]$ with 0-form symmetry group $\widehat{G} = \widehat{A} \rtimes H$ and symmetry category $2\text{Vec}(\widehat{G})$. A general topological surface in $\widehat{\mathcal{T}}$ is labelled by a $\widehat{G}$-graded set $\mathcal{R}$.

In order to gauge $H \subset \widehat{G}$, we introduce the algebra object

$$\mathcal{A}_H = \bigoplus_{h \in H} h \tag{5.6}$$

which is the $\widehat{G}$-graded set with

$$(\mathcal{A}_H)_g = \begin{cases} \{1\} & \text{if } g \in H \\ \emptyset & \text{if } g \notin H \end{cases}. \tag{5.7}$$

This is analogous to the algebra object in (4.7) but now restricted to the subgroup $H$ analogously to the two-dimensional case (3.9).

Topological surfaces in $\widehat{\mathcal{T}}/H$ can be identified with topological surfaces in $\widehat{\mathcal{T}}$ together with instructions for how networks of the algebra object may consistently consistently end on them. The instructions are implemented by 1-morphisms

$$
\begin{aligned}
l &\in \mathrm{Hom}_{\mathcal{T}}(\mathcal{A}_H \otimes \mathcal{R}, \mathcal{R})\,, \\
r &\in \mathrm{Hom}_{\mathcal{T}}(\mathcal{R} \otimes \mathcal{A}_H, \mathcal{R})\,.
\end{aligned}
\tag{5.8}
$$

To formulate additional data and constraints, we consider the components

$$
\begin{aligned}
l_{h,g} &\in \mathrm{Hom}_{\mathcal{T}}(h \otimes \mathcal{R}_g, \mathcal{R}_{hg})\,, \quad h \in H\,, \\
r_{g,h} &\in \mathrm{Hom}_{\mathcal{T}}(\mathcal{R}_g \otimes h, \mathcal{R}_{gh})\,, \quad h \in H\,,
\end{aligned}
\tag{5.9}
$$

which are topological lines specifying how individual symmetry defects end on the surface. Consistency with topological manipulations require the existence of various 2-isomorphisms between these 1-morphisms. These are the same as the ones (4.10), (4.11) (4.12) that we encountered in section 4, but now with appropriate restrictions to $H$.

To make further progress, we write elements of $\widehat{G} = \widehat{A} \rtimes H$ as $g = (\chi, h)$ with $\chi \in \widehat{A}$ and $h \in H$, and by a slight abuse of notation $\mathcal{R}_g = \mathcal{R}_{\chi,h}$. Similarly to section 4, the existence of 2-isomorphisms allows us to construct all component 1-morphisms from the components

$$
\begin{aligned}
l_{h,\chi} &: \quad h \otimes \mathcal{R}_{\chi,e} \to \mathcal{R}_{\chi^h,h}\,, \\
r_{h,\chi} &: \quad \mathcal{R}_{\chi,e} \otimes h \to \mathcal{R}_{\chi,h}\,,
\end{aligned}
\tag{5.10}
$$

where the shift $\chi^h$ appears in the left action due to the ordering in the semi-direct product. The 2-isomorphisms imply that these 1-morphisms are weakly invertible and we can identify $\mathcal{R}_{\chi,h} \cong \mathcal{R}_{\chi,e} =: \mathcal{S}_\chi$ using the right action.

We now form the combination

$$
\rho_{h,\chi} := (r_{\chi^h,h})^{-1} \circ l_{h,\chi} : \quad \mathcal{S}_\chi \to \mathcal{S}_{\chi^h}\,,
\tag{5.11}
$$

which represents the topological line arising from the intersection of a symmetry defect $h \in H$ with the topological surface. The various remaining 2-isomorphisms may be organised into combinations

$$
\Psi_{h,h'|\chi}: \quad \rho_{hh',\chi} \Rightarrow \rho_{h,\chi^{h'}} \circ \rho_{h',\chi} \qquad \Psi_{e|\chi}: \quad 1_{\mathcal{S}_\chi} \Rightarrow \rho_{e,\chi}
\tag{5.12}
$$

subject to the compatibility conditions

$$
\begin{aligned}
\Psi_{h,e|\chi} &= \rho_{h,\chi} \otimes \Psi_{e|\chi}\,, \qquad \Psi_{e,h|\chi} = \Psi_{e|\chi^h} \otimes \rho_{h,\chi}\,, \\
\Psi_{h_1h_2,h_3|\chi} \circ (\Psi_{h_1,h_2|\chi^{h_3}} \otimes \rho_{h_3,\chi}) &= \Psi_{h_1,h_2h_3|\chi} \circ (\rho_{h_1,\chi^{h_2h_3}} \otimes \Psi_{h_2,h_3|\chi})\,,
\end{aligned}
\tag{5.13}
$$

which are illustrated in figure 25.

In summary a topological surface in $\mathcal{T}/G$ is labelled by the following data

1. A collection of sets $\mathcal{S}_\chi \in 2\mathrm{Vect}$ indexed by $\chi \in \widehat{A}$.

2. A collection of 2-matrices $\rho_{h,\chi} \in \mathrm{Hom}(\mathcal{S}_\chi, \mathcal{S}_{\chi^h})$ for all $h \in H$.

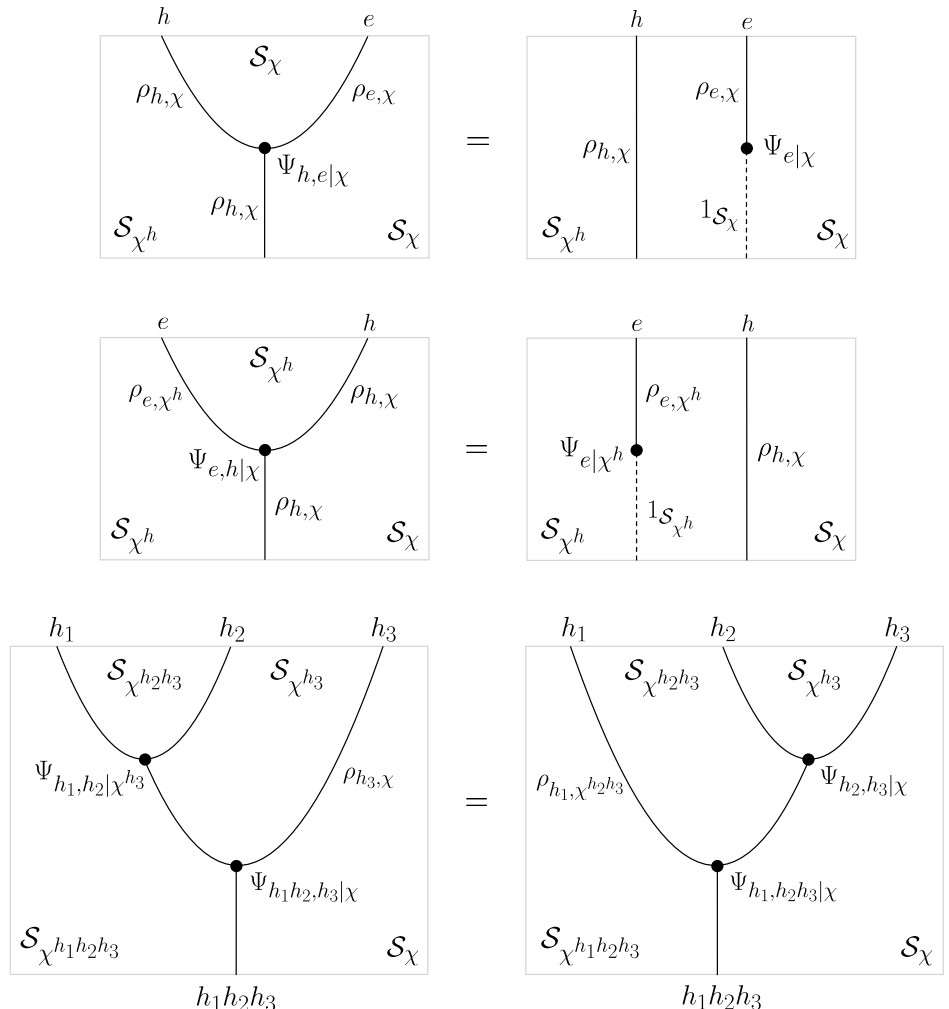

**Figure 25**.

3. 2-isomorphisms $\Psi_{e|\chi} : 1_{\mathcal{S}_\chi} \Rightarrow \rho_{e,\chi}$

4. 2-isomorphisms $\Psi_{h,h'|\chi} : \rho_{hh',\chi} \Rightarrow \rho_{h,\chi^{h'}} \circ \rho_{h',\chi}$

where the 2-isomorphisms are subject to the compatibility relations above. This is precisely the data of a 2-representation of the finite 2-group $G = A[1] \rtimes H$ in 2Vect.

Let us now classify 2-representations up to isomorphism. We introduce the total set

$$\mathcal{S} := \bigsqcup_{\chi \in \widehat{A}} \mathcal{S}_\chi \tag{5.14}$$

with

$$\rho_h := \bigoplus_{\chi \in \widehat{A}} \rho_{h,\chi} : \ \mathcal{S} \to \mathcal{S} . \tag{5.15}$$

Since individual 1-morphisms are weakly invertible, $\rho_h : \mathcal{S} \to \mathcal{S}$ is weakly invertible and acts by a permutation 2-matrix. However, there are restrictions on the form of this permutation. To understand the restrictions, note that for each element $j \in \mathcal{S}$, we may associate a character $\chi_j \in \widehat{A}$ such that $j \in \mathcal{S}_{\chi_j}$. The form of the individual 1-morphisms $\rho_{h,\chi} : \mathcal{S}_\chi \to \mathcal{S}_{\chi^h}$ implies that

$$\chi_j^h \ = \ \chi_{\sigma_h(j)} \tag{5.16}$$

where $\sigma_h$ is the underlying permutation at the level of sets. This says that the $H$ action on the collection $\chi_j$ from the semi-direct product $\widehat{A} \rtimes H$ coincides with that coming from the permutation representation on the set $\mathcal{S}$.

Next, since $\rho_{hh',\chi}$ and $\rho_{h,\chi^h} \circ \rho_{h',\chi}$ are permutation 2-matrices, the 2-isomorphisms $\Psi_{h,h'|\chi}$ are completely determined by a sequence of $|S_\chi|$ phases $c_j^\chi(h,h') \in U(1)$ specifying the isomorphism between the 1-dimensional vector spaces in the $j$-th row. We can combine these phases into an overall sequence $c(h,h') \in U(1)^{|\mathcal{S}|}$ by setting

$$c_j(h,h') \ := \ c_j^{\chi_j}(h,h') \quad \text{for} \quad j \in \mathcal{S}. \tag{5.17}$$

This defines a 2-cochain

$$c : \ H \times H \ \to \ U(1)^{\mathcal{S}}, \tag{5.18}$$

which as a consequence of the compatibility condition (5.13) satisfies the 2-cocycle condition

$$c_{\sigma_h^{-1}(j)}(h',h'') - c_j(hh',h'') + c_j(h,h'h'') - c_j(h,h') \ = \ 0. \tag{5.19}$$

This defines a class

$$c \ \in \ H^2(H, U(1)^{\mathcal{S}}), \tag{5.20}$$

where $U(1)^{\mathcal{S}}$ denotes the abelian group $U(1)^{|\mathcal{S}|}$ supplemented with the structure of a $H$-module via the permutation representation $\sigma$. The 2-representation depends up to isomorphism only on the class.

To summarise, topological surfaces in $\mathcal{T}/G$ are in 1-1 correspondence with isomorphism classes of 2-representations of the split 2-group $G = A[1] \rtimes H$ labelled by triples $(\mathcal{S}, c, \chi)$ consisting of:

- A $H$-set $\mathcal{S}$.

- A class $c \in H^2(H, U(1)^{\mathcal{S}})$.

- A collection of $\chi_j \in \widehat{A}$ indexed by $j \in \mathcal{S}$ such that

$$\chi_j^h \ = \ \chi_{\sigma_h(j)}. \tag{5.21}$$

Here $\sigma : H \to S_n$ denotes the permutation action of $H$ on $\mathcal{S}$.

The above data coincides with the classifying data of a 2-representation of $G = A[1] \rtimes H$. The size $|\mathcal{S}| = n$ is again the dimension of the 2-representation. If $A$ is the trivial group, $\chi = (1, \ldots, 1)$ and the above data reduces to the classifying data of a 2-representation of the ordinary group $H$ as in section 4.2.1[9]. More details on the classification of 2-representations of split 2-groups can be found in appendix A.3.2.

---

[9]In section 4 we denoted finite groups by $G$ instead of $H$.

### 5.2.2 Sum, Product, Conjugation

The sum and product of topological surfaces are determined from those of the parent objects in $\widehat{\mathcal{T}}$ and follow from the sum and product of the underlying $H$-sets and cohomology classes as in section 4, together with the abelian group structure on characters $\widehat{A}$.

Consider two topological surfaces labelled by triples $(\mathcal{S}, c, \chi)$ and $(\mathcal{S}', c', \chi')$. The direct sum is the topological surface labelled by the triple

$$(\mathcal{S}, c, \chi) \oplus (\mathcal{S}', c', \chi') \ = \ (\mathcal{S} \oplus \mathcal{S}', \, c \oplus c', \, \chi \oplus \chi') , \tag{5.22}$$

using the definitions in (4.25) and (4.27) and $\chi \oplus \chi' \equiv \chi \cup \chi'$.

Similarly, the fusion is the topological surface labelled by

$$(\mathcal{S}, c, \chi) \otimes (\mathcal{S}', c', \chi') \ = \ (\mathcal{S} \otimes \mathcal{S}', \, c \otimes c', \, \chi \otimes \chi') , \tag{5.23}$$

using the definitions in (4.25) and (4.27) and $\chi \otimes \chi'$ is defined by

$$(\chi \otimes \chi')_{(i,j)} \ \equiv \ \chi_i + \chi'_j \ \in \ \widehat{A} \tag{5.24}$$

for all $(i, j) \in \mathcal{S} \otimes \mathcal{S}'$. Here, we used an additive notation to denote the group structure on the abelian group $\widehat{A}$.

In addition, the conjugation of a 2-representation $(\mathcal{S}, c, \chi)$ may be defined as the 2-representation $(\mathcal{S}, c, \chi)^{\#} := (\mathcal{S}, -c, -\chi)$ where we have again used additive notation. These operations agree with the corresponding operations in $2\mathrm{Rep}(G)$ as described in appendix A.4.

### 5.2.3 1-Morphisms

Consider the 1-morphism category

$$\mathrm{Hom}_{\mathcal{T}/G}(1, (\mathcal{S}, c, \chi)) \tag{5.25}$$

which describes topological lines bounding or screening the topological surface $(\mathcal{S}, c, \chi)$. This is determined by 1-morphisms of the parent topological surface in $\widehat{\mathcal{T}}$. However, the computation is again determined by the component 1-morphisms involving

$$\mathcal{S} \ \cong \ \bigoplus_{j \in \mathcal{S}} \mathcal{R}_{\chi_j, e} . \tag{5.26}$$

In particular,

$$\mathrm{Hom}_{\widehat{\mathcal{T}}}(1, \mathcal{S}) \ \cong \ \bigoplus_{j \in \mathcal{S}} \mathrm{Hom}_{\widehat{\mathcal{T}}}(1, \mathcal{R}_{\chi_j, e}) \ \cong \ \mathsf{Vec}(\mathcal{S}(\chi)) \tag{5.27}$$

where

$$\mathcal{S}(\chi) \ := \ \{ j \in \mathcal{S} \mid \chi_j = 1 \} \subset \mathcal{S} . \tag{5.28}$$

This happens because the only 1-morphisms in $\widehat{\mathcal{T}}$ are 1-endomorphisms.

These 1-morphisms are supplemented, as discussed in section 4.2.3, by a collection of linear maps ensuring compatibility with intersections of symmetry defects. The result is

that the objects are again graded projective representations of $H$. However, the support of the graded projective representation is now restricted to $S(\chi) \subset S$. We therefore write

$$\mathrm{Hom}_{\mathcal{T}/G}(1, (\mathcal{S}, c, \chi)) \; \cong \; \mathrm{Rep}_{\mathcal{S}(\chi)}^{(\mathcal{S},c)}(H).\tag{5.29}$$

for the category of graded projective representations of $H$ with support $\mathcal{S}(c) \subset \mathcal{S}$.

More generally, the 1-morphism space between arbitrary topological surfaces is

$$\mathrm{Hom}_{\mathcal{T}/G}((\mathcal{S}, c, \chi), (\mathcal{S}', c', \chi')) \; \cong \; \mathrm{Rep}_{\mathcal{S}(\overline{\chi} \otimes \chi')}^{(\mathcal{S} \otimes \mathcal{S}', c'-c)}(H),\tag{5.30}$$

where $\overline{\chi}$ denotes the complex conjugation of $\chi$ with respect to the coefficients in $U(1)$, or simply the inverse in $\widehat{A}$. When $A = 1$, this reproduces the 1-morphism between 2-representations of the ordinary group $H$ in section 4.2.3. More details on the classification of 1-morphisms in $2\mathrm{Rep}(G)$ can be found in appendix C.1.

The composition and fusion of 1-morphisms is given by the composition and tensor product of graded projective representations as described in sections 4.2.4 and 4.2.5. One can check that these operations respect the corresponding restrictions of the support imposed by the characters $\chi$.

## 5.3 Simple Objects

The simple topological surfaces are those that cannot be written as a direct sum of other topological surfaces and correspond to irreducible 2-representations of $G$.

### 5.3.1 Irreducible 2-Representations

Generalising the discussion in section 4.3, a general topological surface $(\mathcal{S}, c, \chi)$ can be decomposed into simple objects by decomposing the underlying $H$-set $\mathcal{S}$ into $H$-orbits. A simple topological surface is then labelled by triples $(\mathcal{O}, c, \chi)$ consisting of

- A $H$-orbit $\mathcal{O}$.

- A class $c \in H^2(H, U(1)^{\mathcal{O}})$.

- A collection of $\chi_j \in \widehat{A}$ indexed by $j \in \mathcal{O}$ such that

$$\chi_j^h \; = \; \chi_{\sigma_h(j)}.\tag{5.31}$$

We emphasise that the collection $\{\chi_j\}$ does not necessarily form a $H$-orbit in $\widehat{A}$. This fact, together with the group cohomology class $c \in H^2(H, U(1)^{\mathcal{O}})$, means we arrive at a larger class of simple objects than the gauging procedure in [5].

However, the fact that $\mathcal{O}$ is a transitive $H$-set implies that the collection $\{\chi_j\}$ of a simple object is constructed from at most one $H$-orbit in $\widehat{A}$, albeit as a union of multiple copies of that orbit. This multiplicity is the origin of condensation defects in our construction. An extreme example is the collection

$$\chi = \{1, \ldots, 1\}\tag{5.32}$$

with $|\mathcal{O}|$ entries, which is a pure simple condensation defect. At the other extreme, simple objects where the collection $\{\chi_j\}$ form a single $H$-orbit are topological surfaces that do not involve condensation at all. The general case is a mixture.

In summary, the set of simple objects may be partitioned into subsets labelled by $H$-orbits in $\widehat{A}$. Each subset contains a minimal object where the collection $\{\chi_j\}$ consists of a single $H$-orbit in $\widehat{A}$ and $c = 0$. This is joined by associated condensation surfaces involving multiple copies of the same orbit and nontrivial classes $c$. To put it another way, the simple topological surfaces, modulo condensations and SPT phases, are labelled by $H$-orbits in $\widehat{A}$. After analysing 1-morphisms below, we will see that these subsets can be regarded as the connected components of the symmetry category $2\mathsf{Rep}(G)$.

### 5.3.2 Induction

An alternative description of simple topological surfaces can be obtained by utilising the fact that every irreducible $n$-dimensional 2-representation $(\mathcal{O}, c', \chi')$ of $G$ can be seen as being induced by a 1-dimensional 2-representation $(c, \chi)$ of a sub-2-group $(A[1] \rtimes K) \subset G$ with $K \subset H$ a subgroup of $H$ of index $|K : H| = n$. Let us write this correspondence as

$$(\mathcal{O}, c', \chi') \;\cong\; \mathrm{Ind}_K^G(c, \chi). \tag{5.33}$$

Thus, we can alternatively label the irreducible 2-representations of $G$ or simple topological surfaces by triples $(K, c, \chi)$ consisting of

- a subgroup $K \subset H$,

- a class $c \in H^2(K, U(1))$,

- a $K$-invariant character $\chi \in \widehat{A}$.

When $A = 1$, this reproduces the labelling of irreducible 2-representations of an ordinary group $H$ as in section 4.3.2. More details on the induction of 2-representations of split 2-groups can be found in appendix A.5.

### 5.3.3 Fusion of Simple Objects

The fusion of simple topological surfaces may again be determined by introducing a basis $\mathcal{O}_\alpha$ of $H$-orbits and decomposing Cartesian products into disjoint unions of orbits.

From the point of view of induced 2-representations, we label the simple topological surfaces by subgroups $K_\alpha \subset H$ (corresponding to the stabilisers of the orbits $\mathcal{O}_\alpha$) together with a class $c_\alpha \in H^2(K_\alpha, U(1))$ and a $K_\alpha$-invariant character $\chi_\alpha \in \widehat{A}$. The fusion of two such simple topological surfaces is then given by

$$(K_\alpha, c_\alpha, \chi_\alpha) \otimes (K_\beta, c_\beta, \chi_\beta) \;\cong\; \bigoplus_{[h] \in K_\alpha \backslash H / K_\beta} \left( K_\alpha \cap K_\beta^h, \, c_\alpha \otimes c_\beta^h, \, \chi_\alpha \otimes \chi_\beta^h \right). \tag{5.34}$$

When $A = 1$, this reduces to the fusion rule for simple 2-representations of the ordinary group $H$ as in section 4.3.3. More details on the fusion of simple 2-representations of $G$ can be found in appendix A.6.

We consider a few special cases. First, the fusion of 1-dimensional irreducible 2-representations $(c, \chi)$ and $(c', \chi')$ labelled by SPT phases $c, c' \in H^2(H, U(1))$ and $H$-invariant characters $\chi, \chi' \in \widehat{A}$ corresponds to addition of the associated cohomology classes and characters,

$$(c, \chi) \otimes (c', \chi') \cong (c + c', \chi + \chi'). \tag{5.35}$$

If $H$ is abelian, fusion simplifies to

$$(K_\alpha, c_\alpha, \chi_\alpha) \otimes (K_\beta, c_\beta, \chi_\beta) \cong \bigoplus_{[h] \in K_\alpha \backslash H / K_\beta} \left( K_\alpha \cap K_\beta, \, c_\alpha \otimes c_\beta, \, \chi_\alpha \otimes \chi_\beta^h \right). \tag{5.36}$$

Note that, unlike in 4.3.3, the right-hand side is not necessarily a multiple of a single summand, but a priori consists of a sum of several different simple topological surfaces due to the appearance of the character $\chi_\alpha \otimes \chi_\beta^h$.

### 5.3.4  1-Morphisms

The category of 1-morphisms between simple topological surfaces can again be simplified using the notion induced graded projective representations (see appendix B.3).

The category of topological lines screening a simple topological surface labelled by $(K, c, \chi)$ is given by

$$\mathrm{Hom}_{\mathcal{T}/G}(1, (K, c, \chi)) \cong \delta_{\chi, 1} \cdot \mathrm{Rep}^c(K). \tag{5.37}$$

More generally,

$$\mathrm{Hom}_{\mathcal{T}/G}\big((K, c, \chi), (K', c', \chi')\big) \cong \bigoplus_{\substack{[h] \in K \backslash H / K': \\ \chi = (\chi')^h}} \mathrm{Rep}^{(c')^h - c}(K \cap (K')^h). \tag{5.38}$$

When $A = 1$, this reduces to 1-morphism categories between irreducible 2-representations of the group $H$, as in section 4.3.4. Further details on the classification of 1-morphisms between irreducible 2-representations can be found in appendix C.1.

For 1-endomorphisms of a simple topological surface we obtain

$$\mathrm{End}_{\mathcal{T}/G}(K, c, \chi) \cong \bigoplus_{\substack{[h] \in H/K: \\ \chi = \chi^h}} \mathrm{Rep}^{c^h - c}(K \cap K^h). \tag{5.39}$$

If $H$ is abelian, this further simplifies to

$$\mathrm{End}_{\mathcal{T}/G}(K, c, \chi) \cong \bigoplus_{[h] \in (H/K)_\chi} \mathrm{Rep}(K), \tag{5.40}$$

where we denoted by $(H/K)_\chi$ the subgroup of $H/K$ consisting of all $[h] \in H/K$ such that $\chi^h = \chi$. The fusion structure on the right-hand side is induced by the composition of 1-morphisms, which is identical to the composition of 1-morphisms described in (4.74).

### 5.3.5 Connected components

Finally, let us comment on the connected components of the symmetry category. A consequence of (5.38) is that 1-morphisms between distinct simple objects exist if and only if $\chi$, $\chi'$ lie in the same $H$-orbit in $\widehat{A}$. In other words, the connected components of the symmetry category are labelled by $H$-orbits in $\widehat{A}$.

## 5.4 Example

Let us consider a theory $\mathcal{T}$ with 2-group symmetry $G = (\mathbb{Z}_2 \times \mathbb{Z}_2)[1] \rtimes \mathbb{Z}_2$, where $\mathbb{Z}_2$ acts on $(\mathbb{Z}_2 \times \mathbb{Z}_2)[1]$ by exchanging the two cyclic factors. In other words, we identify $H = \mathbb{Z}_2$ and $A = \mathbb{Z}_2 \times \mathbb{Z}_2$. This is the 2-group analogue of the ordinary symmetry group $D_8 = (\mathbb{Z}_2 \times \mathbb{Z}_2) \rtimes \mathbb{Z}_2$.

Let us determine the symmetry category $2\mathsf{Rep}(G)$ of $\mathcal{T}/G$ explicitly following the procedure described above. The simple objects are irreducible 2-representations constructed from the trivial orbit $\{1\}$ with stabiliser $K = \mathbb{Z}_2$ and the maximal orbit $\{1, 2\}$ with trivial stabiliser $K = 1$. There are no additional SPT-phases because

$$H^2(\mathbb{Z}_2, U(1)) = 0. \tag{5.41}$$

However, each orbit $\mathcal{O}$ can be supplemented by a collection $\chi$ of $H$-equivariant characters of $\mathbb{Z}_2 \times \mathbb{Z}_2$. Let us denote the characters of $\mathbb{Z}_2 \times \mathbb{Z}_2$ by $\{1, \chi_1, \chi_2, \chi_1\chi_2\}$. Labelling each simple object by a pair $(\mathcal{O}, \chi)$, the simple objects are then given by

- the trivial 2-representation $1 = \big(\{1\}, (1)\big)$,

- a 1-dimensional 2-representation $V = \big(\{1\}, (\chi_1\chi_2)\big)$,

- a 2-dimensional 2-representation $D = \big(\{1, 2\}, (\chi_1, \chi_2)\big)$,

- condensation defect $X = \big(\{1, 2\}, (1, 1)\big)$,

- condensation defect $X' = \big(\{1, 2\}, (\chi_1\chi_2, \chi_1\chi_2)\big)$.

It is straightforward to compute the fusion rules following the general procedure described in section section 5.2.2. In particular, we find

$$
\begin{aligned}
V \otimes V &= 1 \\
V \otimes D &= D \otimes V = D \\
V \otimes X &= X' \\
D \otimes D &= X \otimes (1 \oplus V) \\
X \otimes D &= D \otimes X = D \oplus D \\
X \otimes X &= 2X.
\end{aligned} \tag{5.42}
$$

Note that the only invertible simple objects are $1$ and $V$.

It is worth focussing in particular on the fusion

$$D \otimes D = X \otimes (1 \oplus V) \tag{5.43}$$

and specifically on the appearance of the condensation defect $X$ as a factor on the right-hand side. In [5], the condensation defect arose from the global fusion of topological surfaces on a compact 2-surface $\Sigma_2$ and is expressed in the form

$$D(\Sigma_2) \otimes D(\Sigma_2) \;=\; \frac{1}{\mathbb{Z}_2}(\Sigma_2) \oplus \frac{V}{\mathbb{Z}_2}(\Sigma)\,, \tag{5.44}$$

where the denominators express the fact that from the perspective of $\mathcal{T}/G$ the $\mathbb{Z}_2$ 1-form symmetry is gauged along the surface $\Sigma_2$ as described in [4]. In contrast, we include here the condensation defects $X$, $X'$ from the beginning as simple objects in the symmetry category.

The 1-morphism categories are computed using the procedure described in section 5.3.4 with the result

$$\operatorname{End}(1) = \operatorname{End}(V) = \operatorname{Rep}(\mathbb{Z}_2) \tag{5.45}$$

$$\operatorname{End}(X) = \operatorname{End}(X') = \operatorname{Vect}(\mathbb{Z}_2) \tag{5.46}$$

$$\operatorname{End}(D) = \operatorname{Vect} \tag{5.47}$$

$$\operatorname{Hom}(1, X) = \operatorname{Hom}(V, X') = \operatorname{Vect} \tag{5.48}$$

with all other 1-morphism spaces vanishing.

Note that the only 1-morphisms between distinct objects are between the condensation defects. Therefore the connected components of the symmetry category are labelled by the three $\mathbb{Z}_2$-orbits in $\mathbb{Z}_2 \times \mathbb{Z}_2$: $\{1\}$, $\{\chi_1, \chi_2\}$ and $\{\chi_1\chi_2\}$ with representative objects $1$, $D$, $V$. A diagrammatic representation of the 2-category $2\operatorname{Rep}((\mathbb{Z}_2 \times \mathbb{Z}_2)[1] \rtimes \mathbb{Z}_2)$ can be found in appendix C.4.2[10].

## 6 Applications to Gauge Theory

A common source of finite split 2-groups of the type discussed in section 5 are the automorphism 2-groups of compact connected simple Lie groups, whose 0-form component consists of outer automorphisms and 1-form component consists of the centre. The automorphism 2-group is then realised as a symmetry of associated dynamical gauge theories based on this simple Lie group.

Gauging outer automorphisms leads to dynamical gauge theories with disconnected gauge theories and non-invertible symmetries [5]. In this section, we apply the results of section 5 to compute and identify the symmetry category of certain disconnected gauge theories in three dimensions with 2-representations of a group or 2-group.

### 6.1 Automorphism 2-group

Consider a simple Lie algebra $\mathfrak{g}$ and denote the associated compact, connected, simply connected Lie group $\mathbf{G}$. The associated automorphism 2-group takes the form

$$Z(\mathbf{G}) \rtimes \operatorname{Out}(\mathbf{G}) \tag{6.1}$$

---

[10]Note that in appendix C.4.2, we labelled the simple objects by $\mathbf{1}_\pm$, $\mathbf{2}_\pm$ and $\mathbf{2}_0$, which correspond to the simple topological surfaces $\mathbf{1}_+ \leftrightarrow 1$, $\mathbf{1}_- \leftrightarrow V$, $\mathbf{2}_+ \leftrightarrow X$, $\mathbf{2}_- \leftrightarrow X'$ and $\mathbf{2}_0 \leftrightarrow D$ in our current notation.

where

- Out($\mathbf{G}$) is the group of outer automorphisms of $\mathbf{G}$

- $Z(\mathbf{G})$ is the center of $\mathbf{G}$.

and the Out($\mathbf{G}$)-action on $Z(\mathbf{G})$ is induced from the action of outer automorphisms on $\mathbf{G}$.

Let us now consider a pure gauge theory $\mathcal{T}$ in $D = 3$ dimensions with gauge group $G$. [11] This displays an automorphism 2-group symmetry where

- A 0-form charge conjugation symmetry Out($\mathbf{G}$).

- A 1-form symmetry $Z(\mathbf{G})$ generated by topological Gukov-Witten defects.

and charge conjugation acts on topological Gukov Witten defects by the action of outer automorphisms on representative of conjugacy classes. The symmetry is free from 't Hooft anomalies.

## 6.2  Gauging

Following the discussion in section 5, we may independently gauge the 0-form and 1-form components of the symmetry leading to a commuting square of theories shown in figure 26. In particular, gauging the 1-form centre symmetry $Z(G)$ results in a theory $\widehat{\mathcal{T}}$ corresponding to a dynamical gauge theory with the Langlands dual gauge group $^L G$. This has a magnetic 0-form symmetry

$$\pi_1\left(^L\mathbf{G}\right) = \widehat{Z(\mathbf{G})}, \tag{6.2}$$

forming part of a semi-direct product 0-form symmetry

$$\pi_1(^L\mathbf{G}) \rtimes \text{Out}(\mathbf{G}). \tag{6.3}$$

Now gauging outer automorphisms on either side leads to a dynamical gauge theory with a disconnected gauge group and non-invertible categorical symmetry given by a fusion 2-category of 2-representations.

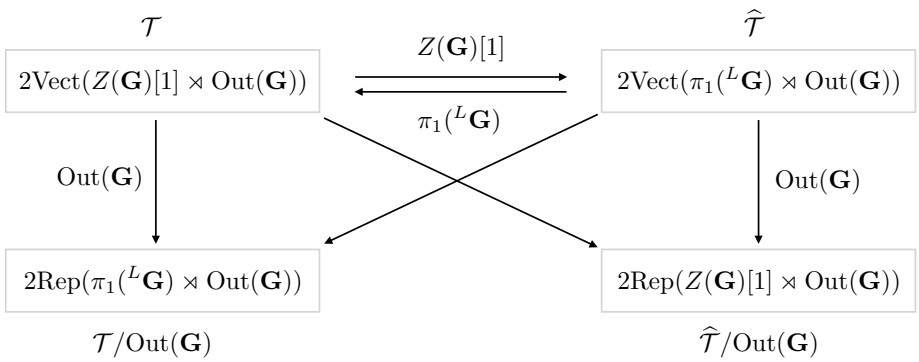

**Figure 26**.

---

[11] Higher dimensional versions are discussed in the following section.

The symmetry categories are summarised in figure 26. This reproduces and extends examples considered in [5] with a systematic inclusion of condensation defects and full description of the symmetry category.

## 6.3 Example

Let us start from the three dimensional gauge theory $\mathcal{T}$ with gauge group

$$\mathbf{G} = \mathrm{Spin}(4N)\,. \tag{6.4}$$

This has automorphism 2-group symmetry

$$(\mathbb{Z}_2 \times \mathbb{Z}_2)[1] \rtimes \mathbb{Z}_2 \tag{6.5}$$

where $\mathbb{Z}_2$ acts by permuting the two factors in $\mathbb{Z}_2 \times \mathbb{Z}_2$. Gauging generates the commuting square of gauge theories and symmetry categories in figure 27. In particular, the symmetry category of the PO(4N) gauge theory was considered in detail in section 5.4.

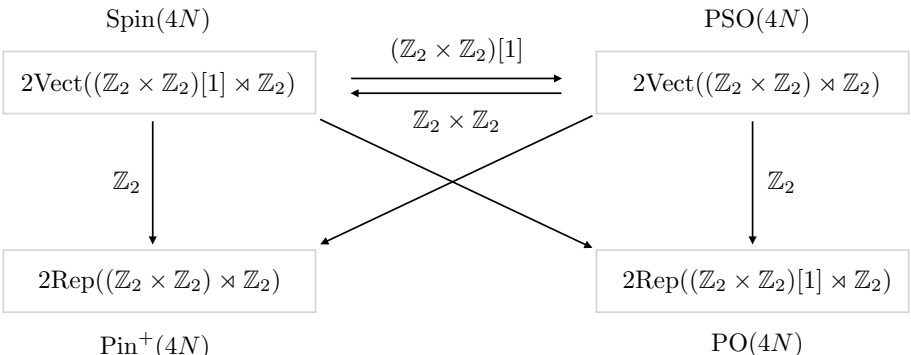

Figure 27.

# 7 Generalisations

## 7.1 Higher groups and higher dimensions

The results of this paper have a natural generalisation to gauging a split $n$-group $G$ in dimension $D > 3$ with $n = 1, \ldots, D-1$ with vanishing Postnikov data. They are determined by a finite 0-form symmetry group $H$, a collection of finite abelian $q$-form symmetry groups $A_q$, and homomorphisms $\varphi_q : H \to \mathrm{Aut}\, A_q$, with $q = 1, \ldots, n-1$. Following our previous notation, this would be denoted

$$(\prod_{q=1}^{D-2} A_q[q]) \rtimes H\,. \tag{7.1}$$

This entire symmetry may be gauged following the methods in this paper by gauging the subgroups $A_q, \ldots, A_1, H$ in sequence.

The symmetry category of $\mathcal{T}/G$ is expected to be a the fusion $(D-1)$-category $(D-1)\mathsf{Rep}(G)$. To the authors' best knowledge, a rigorous definition and construction of such

higher categories is not yet available in the mathematical literature. There is a recent explicit description of $3\mathsf{Rep}(G)$ for an ordinary group $G$ in and more generally we refer the reader to for the state-of-the-art.

Nevertheless, some features may be safely determined. Indeed, the construction of higher representations of higher groups is to some extent inductive in nature. For example, the descriptions topological surfaces as 2-representations in this paper is enough to compute the 2-category of $(D-1)$-fold iterations of endomorphisms of the identity in $\mathcal{T}/G$. For $G$ and ordinary finite group, there will certainly exist simple codimension-1 topological surfaces labelled SPT phases

$$c \in H^{D-1}(G, U(1)) \tag{7.2}$$

with 1-morphisms

$$\mathrm{Hom}_{\mathcal{T}/G}(c, c') = (D-1)\mathsf{Rep}^{c'-c}(G) \tag{7.3}$$

given by projective $(D-2)$-representations of $G$ with cocycle $c' - c$. This structure can already be seen for 3-representations in $D = 4$ in. However, as $D = 3$ already demonstrates there will certainly exist many more simple objects.

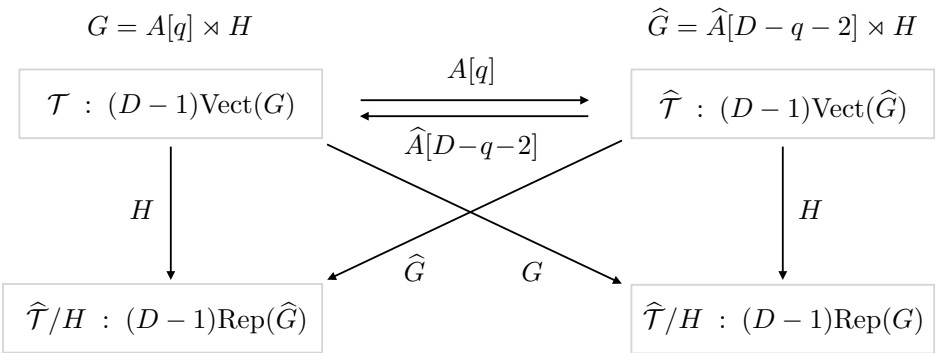

**Figure 28**.

Particularly simple examples are split higher groups involving a single $q$-form symmetry with $q > 0$, which may be written

$$G = A[q] \rtimes H\,. \tag{7.4}$$

for some $q = 1, \ldots D-2$. Gauging symmetries will produce a commuting square of theories illustrated in figure 28. In particular, the example $q = 1$ arises uniformly in pure dynamical gauge theories in $D$-dimensions where $G$ is the automorphism 2-group of a simple, compact, connected Lie group **G**. The situation becomes rather symmetric in $D = 4$ where gauging a 1-form symmetry results in another 1-form symmetry and is illustrated in figure 29. This encompasses the full symmetry categories of anomaly free examples in four dimensions in [5].

## 7.2  Postnikov data, subgroups, 't Hooft anomalies

An important extension is to consider gauging more general higher groups with non-trivial Postnikov data, which is closely connected to gauging more general subgroups and mixed 't

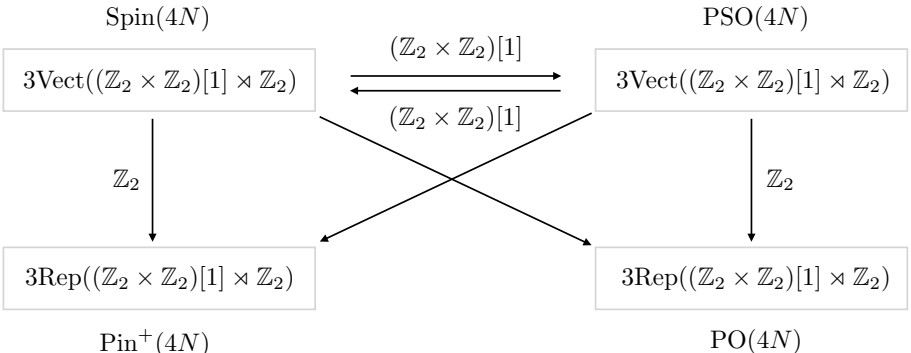

**Figure 29**.

Hooft anomalies. This will bring into the fold more examples of non-invertible symmetries in higher dimensions considered in [2, 3, 7]

Let us consider the situation in dimension $D = 3$ and consider a theory with a finite 2-group $G$ constructed from a 0-form symmetry $H$, abelian 1-form symmetry $A$ and non-trivial Postnikov class $e \in H^3(H, A)$. This may be regarded as a higher analogue of a finite group extension and for this reason it is sometime written

$$1 \to A[1] \to G \to H \to 1. \tag{7.5}$$

The non-trivial Postnikov data means that the commutative diagram of theories considered throughout this paper is restricted to the form illustrated in figure.

Starting from a theory $\mathcal{T}$ with 2-group symmetry $G$ of this type, one cannot now gauge the 0-form symmetry $H$. On the other hand, gauging the 1-form symmetry $A[1]$ leads to a theory $\widehat{\mathcal{T}} = \mathcal{T}/A[1]$ with a direct product 0-form symmetry

$$\widehat{A} \times H \tag{7.6}$$

with a mixed 't Hooft anomaly

$$\int_{X_4} \widehat{\mathbf{a}} \cup e(\mathbf{h}) \tag{7.7}$$

with background fields $\widehat{\mathbf{a}} \in H^1(X_4, \widehat{A})$ and $\mathbf{h} : X_4 \to BH$ [46].

Subsequently gauging the 0-form symmetry $\widehat{A}$ in the theory $\widehat{\mathcal{T}}$ is equivalent to gauging the 2-group symmetry $G$ in $\mathcal{T}$ and the diagram in figure is commutative. By this sequence of gauging, and generalising the methods in this paper, one compute the resulting symmetry category explicitly and demonstrate that it is equivalent to 2Rep$(G)$. This and higher dimensional analogues will be considered in a subsequent paper.

## Acknowledgments

The authors would like to thank Lakshya Bhardwaj and Sakura Schäfer-Nameki for discussions. The work of MB is supported by the EPSRC Early Career Fellowship EP/T004746/1 "Supersymmetric Gauge Theory and Enumerative Geometry", the STFC Research Grant ST/T000708/1 "Particles, Fields and Spacetime", and the Simons Collaboration on Global Categorical Symmetry.

# A  Higher Representation Theory

Ordinary symmetries in quantum field theories can often be described by the structure of ordinary groups. These can be discrete or continuous, abelian or non-abelian. A useful way to study ordinary groups is via their representations, which form an ordinary category.

On the other hand, higher form symmetries in quantum field theories can often be described by higher categorical analogues of groups. The aim of this appendix is to define the first instance of such a higher categorical generalization, and to study a corresponding higher categorical analogue of the notion of representations. Our description follows [47].

## A.1  2-Groups

The first higher categorical generalization of the notion of a group is called a *2-group*. In the following, we will define 2-groups abstractly in analogy to the categorical description of ordinary groups.

### A.1.1  Groups as Categories

Let us begin our discussion with the well-known notion of an ordinary group:

**Definition A.1.** A *group* is a category $C$ with a single object, all of whose morphisms are invertible.

Pictorially, we can visualize this by representing the single object of $C$ by $\bullet$ , and all of its morphisms by loops starting and ending at $\bullet$ :

$$\text{(A.1)}$$

The composition of morphisms then provides a binary operation

$$\circ : \ \mathrm{End}_C(\bullet) \ \times \ \mathrm{End}_C(\bullet) \ \to \ \mathrm{End}_C(\bullet) , \tag{A.2}$$

which we can visualize pictorially by

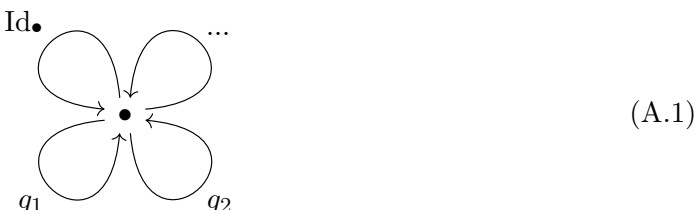

$$\text{(A.3)}$$

As a consequence of the axioms of a category and our assumption of invertibility of morphisms, the binary operation $\circ$ has the following properties:

- *Associativity:* For all $g_1, g_2, g_3 \in \mathrm{End}_C(\bullet)$ it holds that

$$(g_1 \circ g_2) \circ g_3 \ = \ g_1 \circ (g_2 \circ g_3) . \tag{A.4}$$

- *Identity element:* There exists a distinguished morphism $e := \mathrm{Id}_\bullet \in \mathrm{End}_C(\bullet)$ such that for all $g \in \mathrm{End}_C(\bullet)$

$$g \circ e = e \circ g = g. \tag{A.5}$$

- *Inverse element:* For each morphism $g \in \mathrm{End}_C(\bullet)$, there exist a distinguished morphism $g^{-1} \in \mathrm{End}_C(\bullet)$ such that

$$g \circ g^{-1} = g^{-1} \circ g = e. \tag{A.6}$$

The binary operation $\circ$ is also called *group multiplication*. Since the whole structure of the category $C$ is contained in the endomorphism space of $\bullet$ and the group multiplication on it, we will often denote $C$ by $(\mathrm{End}_C(\bullet), \circ)$ in what follows.

### A.1.2   2-Groups as 2-Categories

We can now readily generalize the notion of a group by adding an additional layer of structure to the picture in (A.1). This leads to a higher categorical analogue of a group:

**Definition A.2.** A *2-group* is a 2-category $\mathcal{G}$ with a single object, all of whose 1-morphisms and 2-morphisms are invertible.

Pictorially, we can visualize this by adding 2-morphisms as "morphisms between morphisms" to the picture in (A.1):

$$\tag{A.7}$$

The vertical composition of 2-morphisms then provides a map

$$* : \ 2\mathrm{Hom}_{\mathcal{G}}(g_1, g_2) \ \times \ 2\mathrm{Hom}_{\mathcal{G}}(g_2, g_3) \ \to \ 2\mathrm{Hom}_{\mathcal{G}}(g_1, g_3), \tag{A.8}$$

which we can visualize pictorially by

$$\tag{A.9}$$

Furthermore, the horizontal compostion of 2-morphisms provides a map

$$\star : \ 2\mathrm{Hom}_{\mathcal{G}}(g_1, g_2) \ \times \ 2\mathrm{Hom}_{\mathcal{G}}(g_3, g_4) \ \to \ 2\mathrm{Hom}_{\mathcal{G}}(g_1 \circ g_2, \, g_3 \circ g_4) \,, \tag{A.10}$$

which we can visualize pictorially by

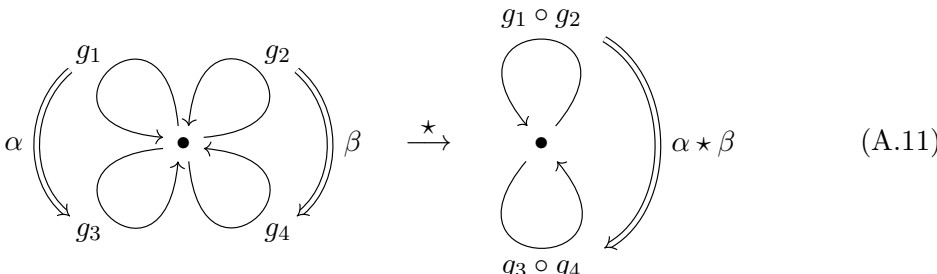

$$\tag{A.11}$$

As a consequence of the axioms of a 2-category and our assumption of invertibility of 1-morphisms and 2-morphisms, the vertical and horizontal compositions $*$ and $\star$ have the following properties:

- *Associativity of* $*$: Given 2-morphisms $g_1 \xRightarrow{\alpha} g_2 \xRightarrow{\beta} g_3 \xRightarrow{\gamma} g_4$, it holds that

$$(\alpha * \beta) * \gamma \ = \ \alpha * (\beta * \gamma) \,. \tag{A.12}$$

- *Associativity of* $\star$: Given 2-morphisms $g_1 \xRightarrow{\alpha} g_2$, $g_3 \xRightarrow{\beta} g_4$ and $g_5 \xRightarrow{\gamma} g_6$, it holds that

$$(\alpha \star \beta) \star \gamma \ = \ \alpha \star (\beta \star \gamma) \,. \tag{A.13}$$

- *Identity elements:* For each 1-morphism $g$ in $\mathcal{G}$ there exists a distinguished 2-morphism $\mathrm{Id}_g \in 2\mathrm{End}_{\mathcal{G}}(g)$ such that for all $\alpha \in 2\mathrm{Hom}(g, g')$ it holds that

$$\mathrm{Id}_g * \alpha \ = \ \alpha * \mathrm{Id}_{g'} \qquad \text{and} \qquad \mathrm{Id}_e \star \alpha \ = \ \alpha \star \mathrm{Id}_e \,. \tag{A.14}$$

- *Inverses for* $*$: For each 2-morphism $\alpha \in 2\mathrm{Hom}_{\mathcal{G}}(g, g')$ there exists a distinguished 2-morphism $\alpha^{-1} \in 2\mathrm{Hom}_{\mathcal{G}}(g', g)$ such that

$$\alpha * \alpha^{-1} = \mathrm{Id}_g \qquad \text{and} \qquad \alpha^{-1} * \alpha = \mathrm{Id}_{g'} \,. \tag{A.15}$$

- *Exchange law:* Given 2-morphisms $g_1 \xRightarrow{\alpha} g_2 \xRightarrow{\alpha'} g_3$ and $h_1 \xRightarrow{\beta} h_2 \xRightarrow{\beta'} h_3$, it holds that

$$(\alpha * \alpha') \star (\beta * \beta') \ = \ (\alpha \star \beta) * (\alpha' \star \beta') \,. \tag{A.16}$$

Note that the exchange law guarantees that the different possibilities to compose 2-morphisms in the flower diagram

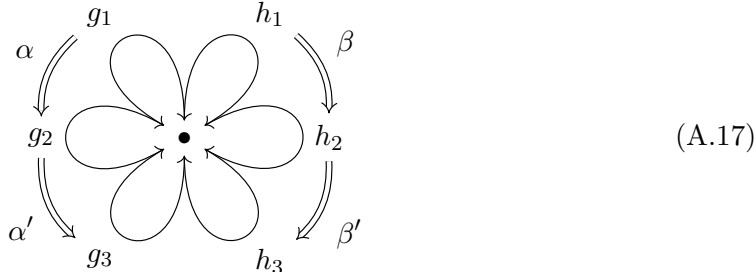

$$\text{(A.17)}$$

using $*$ and $\star$ are compatible with one another. Furthermore, given any $\alpha \in 2\text{Hom}_{\mathcal{G}}(g_1, g_2)$, the exchange law implies that the 2-morphism

$$\overline{\alpha} \; := \; \left(\text{Id}_{g_2^{-1}}\right) \star (\alpha^{-1}) \star \left(\text{Id}_{g_1^{-1}}\right) \; \in \; 2\text{Hom}_{\mathcal{G}}(g_1^{-1}, g_2^{-1}) \tag{A.18}$$

is an inverse of $\alpha$ w.r.t. $\star$ in the sense that $\alpha \star \overline{\alpha} = \overline{\alpha} \star \alpha = \text{Id}_e$.

**Example A.3.** Given a group $C$, we can always construct a trivial 2-group containing $C$ by attaching to each 1-morphism $g$ in $C$ its identity 2-morphism $\text{Id}_g$. Thus, as expected, the notion of a 2-group contains the notion of an ordinary group.

## A.2   2-Representations

Having generalized the notion of a group to a higher categorical analogue, we would like to do the same for the notion of representations of groups. Here and in the following, what we mean by "representations" are *linear* representations on complex vector spaces, all of which we take to be finite-dimensional.

### A.2.1   Representations as Functors

Let us recall the notion of ordinary representations of an ordinary group $C$:

**Definition A.4.** A *representation* of $C$ is a functor $F : C \to \text{Vect}$ from $C$ into the category Vect of vector spaces.

More concretely, what this means is that $F$ assigns

- a vector space $V := F(\bullet)$ to the single object $\bullet$ of $C$,

- a linear map $F_g : V \to V$ to each morphism $g \in C$.

Pictorially, we can visualize this as follows:

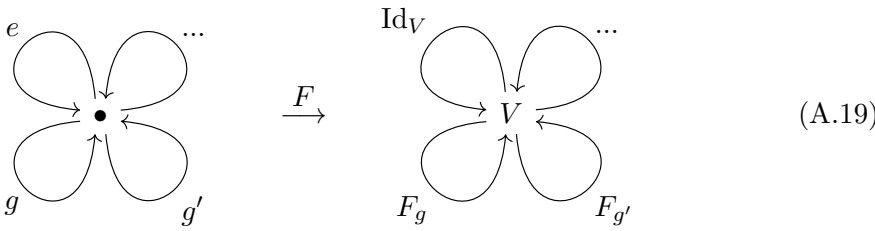

$$\text{(A.19)}$$

Compatibility of $F$ with composition then ensures that for all $g, g' \in C$ it holds that

$$F_{g \circ g'} \;=\; F_g \circ F_{g'} \,. \tag{A.20}$$

As a usful by-product, the description of representations as functors allows us to readily define what we mean by a morphism between representations $F$ and $F'$ of $C$:

**Definition A.5.** An *intertwiner* between $F$ and $F'$ is a natural transformation $\epsilon : F \Rightarrow F'$.

More concretely, what this means is that $\epsilon$ assigns to the single object $\bullet$ of $C$ a linear map $\varphi := \epsilon_\bullet$ between the vector spaces $V := F(\bullet)$ and $V' := F'(\bullet)$, such that for each morphism $g \in C$ there is a commutative diagram

$$\begin{array}{ccc} V & \xrightarrow{\;\;F_g\;\;} & V \\ \varphi \downarrow & & \downarrow \varphi \\ V' & \xrightarrow{\;\;F'_g\;\;} & V' \end{array} \tag{A.21}$$

### A.2.2  2-Representations as 2-Functors

In order to lift up the notion of representations to the level of 2-groups, we need to introduce the 2-categorical analogue of the category Vect of vector spaces:

**Definition A.6.** The 2-category 2Vect of *2-vector spaces* consists of the following data:

- It's objects are natural numbers $n \in \mathbb{N}$.

- Given two objects $n, m \in \mathbb{N}$, the 1-morphism space $\mathrm{Hom}_{2\mathrm{Vect}}(n, m)$ between them is given by the space of $(n \times m)$-matrices $A$ whose entries are vector spaces. Pictorially, we write this as

$$n \xrightarrow{\;A\;} m \;. \tag{A.22}$$

  Composition of 1-morphisms is given by "matrix multiplication" using direct sums and tensor products of vector spaces, i.e.

$$(A \circ B)_{ik} \;:=\; \bigoplus_{j=1}^{m} A_{ij} \otimes B_{jk} \,. \tag{A.23}$$

  Pictorially, we write this as

$$n \xrightarrow{\;A\;} m \xrightarrow{\;B\;} l \qquad \xrightarrow{\;\circ\;} \qquad n \xrightarrow{\;A \circ B\;} l \;. \tag{A.24}$$

- Given two 1-morphisms $A, B \in \mathrm{Hom}_{2\mathrm{Vect}}(n, m)$ between $n$ and $m$, the 2-morphism space $2\mathrm{Hom}_{2\mathrm{Vect}}(A, B)$ between them is given by the space of $(n \times m)$-matrices $\varphi$ whose $(i, j)^{\mathrm{th}}$ entry is a linear map $\varphi_{ij} : A_{ij} \to B_{ij}$. Pictorially, we write this as

$$\begin{array}{c} A \\ n \;\; \Downarrow \varphi \;\; m \\ B \end{array} \tag{A.25}$$

Vertical composition of 2-morphisms is given by composition of linear maps, i.e.

$$(\varphi * \psi)_{ij} := \varphi_{ij} \circ \psi_{ij}, \qquad (A.26)$$

which we write pictorially as

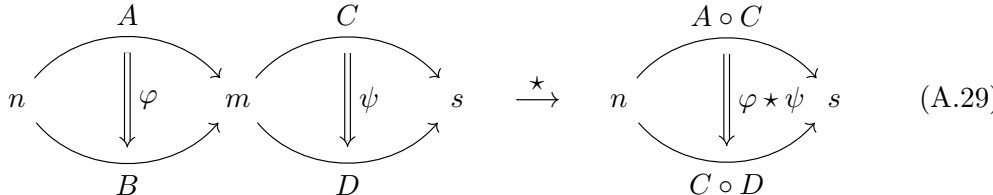

Horizontal composition of 2-morphsims is given by "matrix multiplication" using direct sums and tensor products of linear maps, i.e.

$$(\varphi \star \psi)_{ik} := \bigoplus_{j=1} \varphi_{ij} \otimes \psi_{jk}, \qquad (A.28)$$

which we write pictorially as

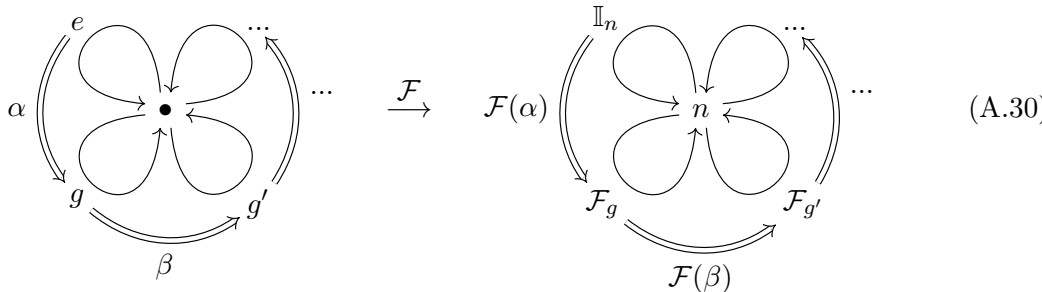

In analogy to before, we can now readily define what we mean by a 2-representation of a 2-group $\mathcal{G}$:

**Definition A.7.** A *2-representation* of $\mathcal{G}$ is a 2-pseudofunctor $\mathcal{F} : \mathcal{G} \to 2\text{Vect}$ from $\mathcal{G}$ into the 2-category 2Vect of 2-vector spaces.

More concretely, what this means is that $\mathcal{F}$ assigns

- a natural number $n := \mathcal{F}(\bullet)$ to the single object $\bullet$ of $\mathcal{G}$,

- a $(n \times n)$-matrix $\mathcal{F}_g$ with vector spaces as entries to each 1-morphism $g$ in $\mathcal{G}$,

- a $(n \times n)$-matrix $\mathcal{F}(\alpha)$ with linear maps as entries to each 2-morphism $\alpha$ in $\mathcal{G}$.

We call $n \in \mathbb{N}$ the *dimension* of $\mathcal{F}$. Pictorially, we can visualize this as follows:

$$ (A.30) $$

Compatibility of $\mathcal{F}$ with composition of 2-morphisms then ensures that

- for all vertically composable 2-morphisms $\alpha$ and $\beta$ in $\mathcal{G}$ it holds that

$$\mathcal{F}(\alpha * \beta) \; = \; \mathcal{F}(\alpha) * \mathcal{F}(\beta), \tag{A.31}$$

- for all 2-morphisms $\alpha$ and $\beta$ in $\mathcal{G}$ it holds that

$$\mathcal{F}(\alpha \star \beta) \; = \; \mathcal{F}(\alpha) \star \mathcal{F}(\beta). \tag{A.32}$$

In addition, since $\mathcal{F}$ is a 2-*pseudo*functor, there exist specified 2-isomorphimsms

$$\phi_e : \; \mathcal{F}_e \Rightarrow \mathbb{I}_n \qquad \text{and} \qquad \phi_{g,g'} : \; \mathcal{F}_g \circ \mathcal{F}_{g'} \Rightarrow \mathcal{F}_{g \circ g'} \tag{A.33}$$

called the *identifier* and the *compositor*, respectively, which are subject to the following compatibility conditions:

- For all 1-morphisms $g$, $h$ and $k$ in $\mathcal{G}$ the diagram

$$
\begin{array}{ccc}
\mathcal{F}_g \circ \mathcal{F}_h \circ \mathcal{F}_k & \xrightarrow{\;\mathrm{Id}_{\mathcal{F}_g} \star \phi_{h,k}\;} & \mathcal{F}_g \circ \mathcal{F}_{h \circ k} \\
{\scriptstyle \phi_{g,h} \star \mathrm{Id}_{\mathcal{F}_k}} \Big\Downarrow & & \Big\Downarrow {\scriptstyle \phi_{g \circ h,k}} \\
\mathcal{F}_{g \circ h} \circ \mathcal{F}_k & \xrightarrow[\;\phi_{g,h \circ k}\;]{} & \mathcal{F}_{g \circ h \circ k}
\end{array}
\tag{A.34}
$$

commutes w.r.t. vertical composition of 2-morphisms.

- For all 1-morphisms $g$ in $\mathcal{G}$ the following diagrams "commute":

$$
\begin{array}{ccccc}
& \overset{\phi_e \star \mathrm{Id}_{\mathcal{F}_g}}{\rightrightarrows} & & \overset{\mathrm{Id}_{\mathcal{F}_g} \star \phi_e}{\leftleftarrows} & \\
\mathcal{F}_e \circ \mathcal{F}_g & & \mathcal{F}_g & & \mathcal{F}_g \circ \mathcal{F}_e \\
& \underset{\phi_{e,g}}{\rightrightarrows} & & \underset{\phi_{g,e}}{\leftleftarrows} &
\end{array}
\tag{A.35}
$$

In analogy to before, it is now straightforward to generalize the notion of intertwiners between representations to the notion of intertwiners between 2-representations:

**Definition A.8.** A *1-intertwiner* between two 2-representations $\mathcal{F}$ and $\mathcal{F}'$ of $\mathcal{G}$ is a pseudo-natural transformation $\varepsilon : \mathcal{F} \Rightarrow \mathcal{F}'$.

More concretely, what this means is that $\varepsilon$ assigns

- a 1-morphism $\varphi := \varepsilon_\bullet : n \to m$ to the single element $\bullet$ of $\mathcal{G}$,

- a 2-morphism $\varepsilon_g : \varphi \circ \mathcal{F}'_g \Rightarrow \mathcal{F}_g \circ \varphi$ to each 1-morphism $g$ in $\mathcal{G}$,

so that the following compatibility conditions are satisfied:

- $\varphi$ is compatible with the identifiers of $\mathcal{F}$ and $\mathcal{F}'$ in the sense that the following diagram commutes:

$$\begin{array}{ccc} \varphi \circ \mathcal{F}'_e & \xrightarrow{\ \varepsilon_g\ } & \mathcal{F}_e \circ \varphi \\ & & \\ \mathrm{Id}_\varphi \star \phi'_e \searrow & & \swarrow \phi_e \star \mathrm{Id}_\varphi \\ & \varphi & \end{array} \tag{A.36}$$

- $\varepsilon$ is compatible with the compositors of $\mathcal{F}$ and $\mathcal{F}'$ in the sense that for all 1-morphisms $g_1$ and $g_2$ in $\mathcal{G}$ the following diagram commutes:

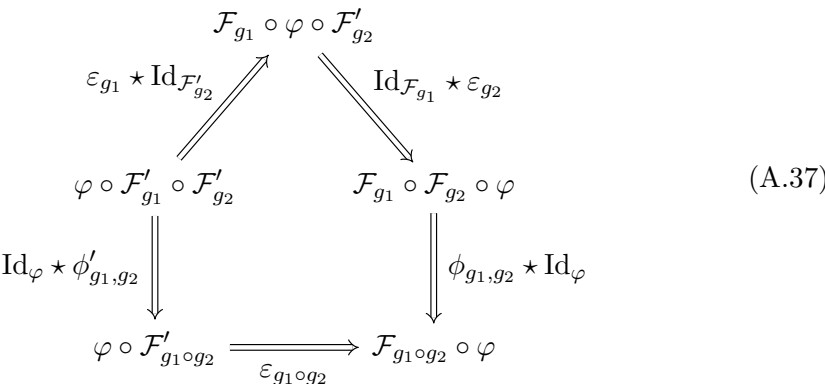

$$\tag{A.37}$$

- $\varepsilon$ is compatible with the 2-morphisms in $\mathcal{G}$ in the sense that for any 2-morphism $\alpha : g \to g'$ in $\mathcal{G}$ the following diagram commutes:

$$\begin{array}{ccc} \varphi \circ \mathcal{F}'_g & \xrightarrow{\ \varepsilon_g\ } & \mathcal{F}_g \circ \varphi \\ \mathrm{Id}_\varphi \star \mathcal{F}'(\alpha) \Big\Downarrow & & \Big\Downarrow \mathcal{F}(\alpha) \star \mathrm{Id}_\varphi \\ \varphi \circ \mathcal{F}'_{g'} & \xrightarrow[\ \varepsilon_{g'}\ ]{} & \mathcal{F}_{g'} \circ \varphi \end{array} \tag{A.38}$$

As the denomination suggests, we can regard 1-intertwiners as 1-morphisms between 2-representations. This viewpoint is supported by the fact that there is a natural way to compose them:

**Definition A.9.** Let $\varepsilon : \mathcal{F} \Rightarrow \mathcal{F}'$ and $\varepsilon' : \mathcal{F}' \to \mathcal{F}''$ be two 1-intertwiners between 2-representations $\mathcal{F}$, $\mathcal{F}'$ and $\mathcal{F}'$, $\mathcal{F}''$ of $\mathcal{G}$, respectively. Then, their *composition* is the 1-intertwiner $\varepsilon \circ \varepsilon' : \mathcal{F} \Rightarrow \mathcal{F}''$ defined by

$$(\varepsilon \circ \varepsilon')_\bullet \ := \ \varphi \circ \varphi' \tag{A.39}$$

(where $\varphi \equiv \varepsilon_\bullet$ and $\varphi' \equiv \varepsilon'_\bullet$) and

$$(\varepsilon \circ \varepsilon')_g \ := \ \left( \mathrm{Id}_\varphi \star \varepsilon'_g \right) \ast \left( \varepsilon_g \star \mathrm{Id}_{\varphi'} \right) \tag{A.40}$$

for all 1-morphisms $g$ in $\mathcal{G}$.

The composition of 1-intertwiners then allows us to define what we mean by saying that two given 2-representations of $\mathcal{G}$ are "essentially the same":

**Definition A.10.** Two 2-representations $\mathcal{F}$ and $\mathcal{F}'$ of $\mathcal{G}$ are said to be *equivalent* if there exists an invertible 1-intertwiner $\varepsilon : \mathcal{F} \Rightarrow \mathcal{F}'$ between them.

In addition to 1-intertwiners, there also exists a notion of 2-intertwiners that can be seen as 2-morphisms between two 1-intertwiners:

**Definition A.11.** Let $\varepsilon, \varepsilon' : \mathcal{F} \Rightarrow \mathcal{F}'$ be two 1-intertwiners between two 2-representations $\mathcal{F}$ and $\mathcal{F}'$ of $\mathcal{G}$. Then, a *2-intertwiner* between $\mathcal{F}$ and $\mathcal{F}'$ is a modification $\eta : \varepsilon \Rrightarrow \varepsilon'$.

More concretely, what this means is that $\eta$ is a 2-morphism $\eta : \varphi \Rightarrow \varphi'$ in 2Vect such that the following diagram commutes for all 1-morphisms $g$ in $\mathcal{G}$:

$$
\begin{array}{ccc}
\varphi \circ \mathcal{F}'_g & \overset{\varepsilon_g}{\Longrightarrow} & \mathcal{F}_g \circ \varphi \\
\eta \star \mathrm{Id}_{\mathcal{F}'_g} \Big\Downarrow & & \Big\Downarrow \mathrm{Id}_{\mathcal{F}_g} \star \eta \\
\varphi' \circ \mathcal{F}'_g & \underset{\varepsilon'_g}{\Longrightarrow} & \mathcal{F}_g \circ \varphi'
\end{array}
\tag{A.41}
$$

In summary, for each 2-group $\mathcal{G}$ we obtain a 2-category $2\mathrm{Rep}(\mathcal{G})$ of 2-representations of $\mathcal{G}$, which can be described as follows:

**Definition A.12.** The 2-category $2\mathrm{Rep}(\mathcal{G})$ consists of the following data:

- Its objects are 2-representations $\mathcal{F} : \mathcal{G} \to 2\mathrm{Vect}$.

- Given two objects $\mathcal{F}$ and $\mathcal{F}'$, the 1-morphism space between them is given by the space of 1-intertwiners $\varepsilon : \mathcal{F} \Rightarrow \mathcal{F}'$. Composition of 1-morphisms is given by composition of 1-intertwiners as in Definition A.9.

- Given two 1-morpshims $\varepsilon$ and $\varepsilon'$ between two objects $\mathcal{F}$ and $\mathcal{F}'$, the space of 2-morphisms between them is given by the space of 2-intertwiners $\eta : \varepsilon \Rrightarrow \varepsilon'$. Vertical and horizontal composition are given by vertical and horizontal composition in 2Vect, respectively.

For the remainder of the appendix, we will be interested in studying the 2-category $2\mathrm{Rep}(\mathcal{G})$ for certain special types of 2-groups $\mathcal{G}$ that we will define below.

## A.3 Classification

In the theory of ordinary groups and their representations, we often only care about families of groups and representations that are "essentially the same". Similarly, in the case of 2-groups and their 2-representations, we often only care about families of 2-groups and 2-representations that are "equivalent" in an appropriate sense. A natural question to ask is if there is a simple way to classify equivalence classes of 2-groups and 2-representations. In the following, we will try to answer this question for special types of 2-groups and 2-representations thereof.

### A.3.1  Classification of 2-groups

The definition of 2-groups as a special kind of 2-categories is elegant but rather abstract. It can be related to the more familiar concept of ordinary groups using the notion of crossed modules. Recall that a crossed module is a quadruple $(C, D, \rhd, \partial)$, where

- $C$ is a group with group multiplication $\odot$,

- $D$ is a group with group multiplication $\otimes$,

- $\rhd : C \to \mathrm{Aut}(D)$ is an action of $C$ on $D$ via automorphisms,

- $\partial : D \to C$ is a group homomorphism compatible with $\rhd$ in the sense that

$$\partial(g \rhd \alpha) \;=\; g \odot \partial(\alpha) \odot g^{-1} \tag{A.42}$$
$$\partial(\alpha) \rhd \beta \;=\; \alpha \otimes \beta \otimes \alpha^{-1} \tag{A.43}$$

for all $g \in C$ and $\alpha, \beta \in D$.

**Remark A.13.** Note that that relation (A.42) implies that

$$g \odot h \odot g^{-1} \;\in\; \mathrm{im}(\partial) \tag{A.44}$$

for all $h \in \mathrm{im}(\partial)$ and all $g \in C$, so that $\mathrm{im}(\partial) \subset C$ is a normal subgroup of $C$. Consequently, the quotient $\mathrm{coker}(\partial) \equiv C \,/\, \mathrm{im}(\partial)$ is itself a group. Similarly, relation (A.43) implies that

$$[\alpha, \beta] \;=\; 1 \tag{A.45}$$

for all $\alpha \in \ker(\partial)$ and all $\beta \in D$, so that $\ker(\partial) \subset D$ is a subgroup of the centre of $D$ (and is hence abelian). One can then check that $\rhd$ descends to a well-defined action of $\mathrm{coker}(\partial)$ on $\ker(\partial)$.

Using the above, we can now state the following:

**Proposition A.14.** There is a 1:1-correspondence between 2-groups and crossed modules.

*Proof.* We will only sketch one side of the correspondence. That is, given 2-group $\mathcal{G}$, we can define a crossed module $(C, D, \rhd, \partial)$ as follows:

- We set the group $C$ to be the 1-endomorphism space of the single object $\bullet$ in $\mathcal{G}$ together with composition of 1-morphisms, i.e.

$$C \;:=\; (\mathrm{End}_{\mathcal{G}}(\bullet), \circ) \,. \tag{A.46}$$

- We set the group $D$ to be space of 2-morphisms with source $e \in \mathrm{End}_{\mathcal{G}}(\bullet)$ together with horizontal composition of 2-morphisms, i.e.

$$D \;:=\; (2\mathrm{Hom}_{\mathcal{G}}(e, \,.\,), \star) \,. \tag{A.47}$$

- We define the group homomorphism $\vartriangleright : C \to \mathrm{Aut}(D)$ by

$$g \vartriangleright \alpha \ := \ \big(\mathrm{Id}_{g^{-1}}\big) \star \alpha \star \big(\mathrm{Id}_g\big) \tag{A.48}$$

for each $g \in C$ and $\alpha \in D$.

- We define the group homomorphism $\partial : D \to C$ by

$$\alpha \ \in \ 2\mathrm{Hom}_{\mathcal{G}}(e,g) \ \mapsto \ g \ \in \ \mathrm{End}_{\mathcal{G}}(\bullet) \,. \tag{A.49}$$

One can check that $\vartriangleright$ and $\partial$ as defined above satisfy the relations (A.42) and (A.43), so that $(C, D, \vartriangleright, \partial)$ forms a crossed module as claimed. $\qquad\square$

**Example A.15.** Given a group $D$, we can use it to construct a crossed module $(C, D, \vartriangleright, \partial)$ by setting $C := \mathrm{Aut}(D)$, $\vartriangleright := \mathrm{Id} : C \to \mathrm{Aut}(D)$ and $\partial := \mathrm{conj} : D \to C$. We call this the *automorphism crossed module* of $D$.

As a by-product, the description of 2-groups as crossed modules allows us to readily define what we mean by a morphism between two 2-groups:

**Definition A.16.** Let $\mathcal{G} = (C, D, \vartriangleright, \partial)$ and $\mathcal{G}' = (C', D', \vartriangleright', \partial')$ be two crossed modules. Then, a *morphism* between $\mathcal{G}$ and $\mathcal{G}'$ is a pair $(\varphi, \psi)$ of group homomorphisms sitting in the commutative diagram

$$
\begin{array}{ccc}
D & \xrightarrow{\ \partial\ } & C \\
{\scriptstyle\psi}\big\downarrow & & \big\downarrow{\scriptstyle\varphi} \\
D' & \xrightarrow{\ \partial'\ } & C'
\end{array}
\tag{A.50}
$$

that respect the group actions $\vartriangleright$ and $\vartriangleright'$ in the sense that

$$\psi(g \vartriangleright \alpha) \ = \ \varphi(g) \vartriangleright' \psi(\alpha) \tag{A.51}$$

for all $g \in C$ and $\alpha \in D$. The morphism $(\varphi, \psi)$ is called an *equivalence* between $\mathcal{G}$ and $\mathcal{G}'$ if it induces group isomorphisms between $\ker(\partial) \cong \ker(\partial')$ and $\mathrm{coker}(\partial) \cong \mathrm{coker}(\partial')$.

Furthermore, the notion of crossed modules allows us to define what we mean by a sub-2-group (or equivalently a crossed submodule) of a 2-group:

**Definition A.17.** A crossed module $(C, D, \vartriangleright, \partial)$ is called a *crossed submodule* of a crossed module $(C', D', \vartriangleright', \partial')$ if

- $C$ is a subgroup of $C'$,

- $D$ is a subgroup of $D'$,

- the action $\vartriangleright$ of $C$ on $D$ is induced by the action $\vartriangleright'$ of $C'$ on $D'$,

- $\partial = \partial'|_D$ is the restriction of $\partial'$ to $D$.

Finally, we are able to classify equivalence classes of 2-groups as follows:

**Theorem A.18.** There is a 1:1-correspondence between equivalence classes of 2-groups $\mathcal{G}$ and equivalence classes of quadruples $(G, A, \rho, \theta)$, where

- $G$ is a group,

- $A$ is an abelian group,

- $\rho : G \to \mathrm{Aut}(A)$ is an action of $G$ on $A$ via automorphisms,

- $\theta \in Z_\rho^3(G, A)$ is a twisted 3-cocycle on $G$ with values in $A$.

Two such quadruples $(G, A, \rho, \theta)$ and $(G', A', \rho', \theta')$ are said to be *equivalent* if there exist group isomorphisms $\varphi : G \xrightarrow{\sim} G'$ and $\psi : A \xrightarrow{\sim} A'$ such that

$$\psi(g \rhd_\rho a) \;=\; \varphi(g) \rhd_{\rho'} \psi(a) \qquad \text{and} \qquad [\varphi^*(\theta')] \;=\; [\psi \circ \theta] \tag{A.52}$$

for all $g \in G$ and $a \in A$, where $[.] : Z_{\rho' \circ \varphi}^3(G, A') \to H_{\rho' \circ \varphi}^3(G, A')$ denotes the projection into group cohomology. The class $[\theta] \in H_\rho^3(G, A)$ is often called the *Postnikov class* of the (equivalence class of the) 2-group $\mathcal{G}$.

*Proof.* We will only sketch one side of the correspondence. That is, given a 2-group $\mathcal{G}$, thought of as a crossed module $(C, D, \rhd, \partial)$, we can construct a quadruple $(G, A, \rho, \theta)$ as follows: According to Remark A.13, the action $\rhd$ descends to a well-defined action $\rho$ of the group $G := \mathrm{coker}(\partial)$ on the abelian group $A := \ker(\partial)$. By definition, the isomorphism classes of $G$ and $A$ do not depend on the equivalence class of $\mathcal{G}$. In order to construct the twisted 3-cocycle $\theta$, we embed $G$ and $A$ into the four-term exact sequence

$$1 \;\longrightarrow\; A \;\overset{\imath}{\hookrightarrow}\; D \;\overset{\partial}{\longrightarrow}\; C \;\overset{\pi}{\twoheadrightarrow}\; G \;\longrightarrow\; 1 \,, \tag{A.53}$$

where $\imath$ denotes inclusion of $A$ into $D$ and $\pi$ denotes projection from $C$ onto $G$. We then choose a section $s : G \to C$ such that $\pi \circ s = \mathrm{Id}_G$ and define $c : G \times G \to C$ by

$$c(g_1, g_2) \;:=\; s(g_1) \odot s(g_2) \odot s(g_1, g_2)^{-1} \tag{A.54}$$

for all $g_1, g_2 \in G$. Since $\pi \circ c = 1$, there exists a $d : G \times G \to D$ with $\partial \circ d = c$, which allows us to define a 3-cochain $\theta : G \times G \times G \to D$ by

$$\theta(g_1, g_2, g_3) \;:=\; \big[\, s(g_1) \rhd d(g_2, g_3) \,\big] \otimes d(g_1, g_2 \cdot g_3) \otimes d(g_1 \cdot g_2, g_3)^{-1} \otimes d(g_1, g_2)^{-1} \tag{A.55}$$

for all $g_1, g_2, g_3 \in G$. One can then check that

$$\partial \circ \theta \;=\; 1 \qquad \text{and} \qquad \delta_\rho \theta \;=\; 1 \,, \tag{A.56}$$

so that $\theta$ defines a twisted 3-cocycle $\theta \in Z_\rho^3(G, A)$ on $G$ with values in $A$. One can check that its class $[\theta] \in H_\rho^3(G, A)$ in group cohomology does not depend on the choices of $s$ and $d$, and that equivalent 2-groups $\mathcal{G}$ and $\mathcal{G}'$ lead to equivalent cohomology classes $[\theta]$ and $[\theta']$ in the sense of (A.52). $\qquad\square$

**Example A.19.** Let $D$ be a group and let $\mathcal{G}$ be its automorphism crossed module as in Example A.15. Then, $\mathcal{G}$ is classified by the quadruple $(\mathrm{Out}(D), Z(D), \mathrm{Id}, \theta)$, where $\mathrm{Out}(D)$ denotes the group of outer automorphisms of $D$, $Z(D)$ denotes the centre of $D$, and $\theta$ classifies the trivial double extension of $\mathrm{Out}(D)$ by $Z(D)$ through the four-term exact sequence

$$1 \longrightarrow Z(D) \overset{\imath}{\hookrightarrow} D \xrightarrow{\text{conj}} \mathrm{Aut}(D) \xrightarrow{\pi} \mathrm{Out}(D) \longrightarrow 1 \,. \qquad (A.57)$$

**Remark A.20.** In the following, we will abuse notation and speak of quadruples $(G, A, \rho, \theta)$ as 2-groups. According to Theorem A.18, this abuse of notation is justified if we only care about 2-groups up to equivalence.

The classification of 2-groups by quadruples $(G, A, \rho, \theta)$ allows us to define a special type of 2-group that will be our main point of interest in the following:

**Definition A.21.** A 2-group $\mathcal{G}$ is said to be *split* if its Postnikov class vanishes, i.e. $[\theta] = 0$.

In other words, a split 2-group $\mathcal{G}$ is such that $\theta$ is cohomologous to an exact piece. Using equivalences of 2-groups, we can always remove this exact piece and set $\theta = 0$, so that, up to equivalence, we can label split 2-groups $\mathcal{G}$ by triples $(G, A, \rho)$ with $G$, $A$ and $\rho$ as above. It is then straightforward to describe sub-2-groups of split 2-groups:

**Lemma A.22.** Let $\mathcal{H} = (H, B, \eta)$ be a sub-2-group of a split 2-group $\mathcal{G} = (G, A, \rho)$. Then,

- $H \subset G$ is a subgroup of $G$,

- $B \subset A$ is a subgroup of $A$,

- the action $\eta$ of $H$ on $B$ is induced by the action $\rho$ of $G$ on $A$.

**Remark A.23.** Note that, given a subgroup $H \subset G$, we can always construct a split sub-2-group $(H, A, \rho|_H)$ of $\mathcal{G}$. Any other sub-2-group of the form $\mathcal{H} = (H, B, \eta)$ can be obtained from $(H, A, \rho|_H)$ by restricting $A$ to $H$-orbits $B \subset A$ w.r.t. $\rho|_H$. In the following, we will therefore only consider sub-2-groups of the form $(H, A, \rho|_H)$ for some subgroup $H \subset G$, which for simplicity we also just denote by $H \subset \mathcal{G}$.

### A.3.2 Classification of 2-Representations

We now turn to the classification of equivalence classes of 2-representations. For the remainder of the appendix, we will fix $\mathcal{G} = (G, A, \rho)$ to be a finite split 2-group. As shown in [19, 21], the 2-representations of $\mathcal{G}$ can then be classified as follows:

**Theorem A.24.** There is a 1:1-correspondence between equivalence classes of $n$-dimensional 2-representations of $\mathcal{G}$ and equivalence classes of triples $(\sigma, c, \chi)$, where

- $\sigma : G \to S_n$ is a permutation representation of $G$ on $\langle n \rangle \equiv \{1, ..., n\}$,

- $c \in Z^2_\sigma(G, U(1)^n)$ is a twisted 2-cocycle on $G$ with values in $U(1)^n$,

- $\chi \in (A^\vee)^n$ is a collection of $n$ characters of $A$,

such that for all $g \in G$ and $a \in A$ it holds that

$$g \rhd_\sigma \chi(a) \;=\; \chi(g \rhd_\rho a)\,. \tag{A.58}$$

Two such triples $(\sigma, c, \chi)$ and $(\sigma', c', \chi')$ are said to be *equivalent* if there exists a permutation $\tau \in S_n$ such that

$$\sigma' \;=\; \tau \circ \sigma \circ \tau^{-1}\,, \qquad [c'] \;=\; [\tau \rhd c]\,, \qquad \chi' \;=\; \tau \rhd \chi\,, \tag{A.59}$$

where $[.] : Z^2_{\sigma'}(G, U(1)^n) \to H^2_{\sigma'}(G, U(1)^n)$ denotes the projection into group cohomology.

*Proof.* We will only sketch one side of the correspondence. That is, given a 2-representation $\mathcal{F}$ of $\mathcal{G}$, we want to construct a triple $(\sigma, c, \chi)$ as above. To do this, we again think of the 2-group $\mathcal{G}$ as a 2-category and the 2-representation $\mathcal{F}$ as a 2-pseudofunctor from $\mathcal{G}$ to 2Vect, as illustrated in (A.30).

Then, $\mathcal{F}$ assigns to each 1-morphism $g$ in $\mathcal{G}$ a $(n \times n)$-matrix $\mathcal{F}_g$, whose entries are vector spaces. Due to the existence of 2-isomorphisms

$$\phi_{g,g^{-1}} * \phi_e : \;\; \mathcal{F}_g \circ \mathcal{F}_{g^{-1}} \;\Rightarrow\; \mathbb{I}_n \tag{A.60}$$

constructed from the compositors and the identifier of $\mathcal{F}$, each $\mathcal{F}_g$ contains only one non-vanishing vector space per row and column, all of which are 1-dimensional. Thus, we can think of $\mathcal{F}_g$ as a $(n \times n)$-permutation matrix for each 1-morphism $g$ in $\mathcal{G}$, which induces a permutation representation

$$\sigma : \; (\mathrm{End}_{\mathcal{G}}(\bullet), \circ) \;\to\; S_n\,. \tag{A.61}$$

In particular, if there exists a 2-isomorphism $\alpha : e \Rightarrow g$ in $\mathcal{G}$, we can construct the 2-isomorphism

$$\mathcal{F}(\alpha)^{-1} * \phi_e : \;\; \mathcal{F}_g \;\Rightarrow\; \mathbb{I}_n \tag{A.62}$$

in 2Vect, which implies that the associated permutation matrix $\sigma_g$ is trivial. Thus, the permutation representation $\sigma$ descends to a well-defined permutation representation of

$$G \;\equiv\; \mathrm{coker}(\partial)\,, \tag{A.63}$$

where we denoted by $\partial : (2\mathrm{Hom}_{\mathcal{G}}(e, \,.\,), \star) \to (\mathrm{End}_{\mathcal{G}}(\bullet), \circ)$ the group homomorphism

$$\alpha \,\in\, 2\mathrm{Hom}_{\mathcal{G}}(e, g) \;\mapsto\; g \,\in\, \mathrm{End}_{\mathcal{G}}(\bullet)\,. \tag{A.64}$$

Next, we note that for two given 1-morphisms $g$ and $h$ in $\mathcal{G}$, the compositor

$$\phi_{g,h} : \; \mathcal{F}_g \circ \mathcal{F}_h \;\Rightarrow\; \mathcal{F}_{g \circ h} \tag{A.65}$$

is an $(n \times n)$-matrix of invertible linear maps between the 1-dimensional vector spaces sitting in $\mathcal{F}_g \circ \mathcal{F}_h$ and $\mathcal{F}_{g \circ h}$. Since the latter only have one non-vanishing entry per row and column, $\phi_{g,h}$ is entirely determined by a collection of $n$ phases $c_i(g, h) \in U(1)$ specifying

the isomorphism between the 1-dimensional vector spaces sitting in the $i$-th row of $\mathcal{F}_g \circ \mathcal{F}_h$ and $\mathcal{F}_{g \circ h}$, respectively. This induces a map

$$c : \ \text{End}_{\mathcal{G}}(\bullet) \times \text{End}_{\mathcal{G}}(\bullet) \ \to \ U(1)^n \,, \tag{A.66}$$

which as a consequence of (A.34) satisfies the twisted cocycle condition

$$c_{\sigma_g^{-1}(i)}(h,k) - c_i(g \circ h, k) + c_i(g, h \circ k) - c_i(g,h) \ = \ 0 \,. \tag{A.67}$$

Thus, we obtain a twisted 2-cocycle on $(\text{End}_{\mathcal{G}}(\bullet), \circ)$ with values in $U(1)^n$, which can be checked to descend to a well-defined 2-cocycle on $G \equiv \text{coker}(\partial)$.

Lastly, we note that for each 2-isomoprhism $\alpha : e \Rightarrow g$ we have a 2-isomorphism

$$\mathcal{F}(\alpha) : \ \mathcal{F}_e \ \Rightarrow \ \mathcal{F}_g \,, \tag{A.68}$$

which is a $(n \times n)$-matrix of invertible linear maps between the 1-dimensional vector spaces sitting in $\mathcal{F}_e$ and $\mathcal{F}_g$. Since the latter only have non-vanishing entries on the diagonal, $F(\alpha)$ is again entirely determined by a collection of $n$ phases $\chi_i(\alpha) \in U(1)$ specifying the isomorphism between the 1-dimensional vector spaces in the $i$-th diagonal entry of $\mathcal{F}_e$ and $\mathcal{F}_g$, respectively. This induces a homomorphism

$$\chi : \ (2\text{Hom}_{\mathcal{G}}(e, \,.\,), \star) \ \to \ U(1)^n \,, \tag{A.69}$$

which can be checked to satisfy

$$\chi(g \triangleright \alpha) \ = \ g \triangleright_\sigma \chi(\alpha) \,, \tag{A.70}$$

where we denoted by $\triangleright$ the action of $(\text{End}_{\mathcal{G}}(\bullet), \circ)$ on $(2\text{Hom}_{\mathcal{G}}(e, \,.\,), \star)$ as in (A.48). The homomorphism $\chi$ then descends to a well-defined collection of characters of $A \equiv \ker(\partial)$. $\square$

**Remark A.25.** In the following, we will abuse notation and speak of triples $(\sigma, c, \chi)$ as 2-representations of $\mathcal{G}$. According to Theorem A.24, this abuse of notation is justified if we only care about 2-representations up to equivalence.

**Example A.26.** For any split 2-group $\mathcal{G}$, setting $(\sigma, c, \chi) = (1, 1, 1)$ gives a 1-dimensional 2-representation of $\mathcal{G}$, which is called the *trivial* 2-representation and denoted by **1**.

Just as for ordinary complex representations of a group $G$, there exists a notion of the conjugate of a 2-representation of the 2-group $\mathcal{G}$:

**Definition A.27.** Let $(\sigma, c, \chi)$ be a 2-representation of $\mathcal{G}$. Then, its *conjugate* is the 2-representation

$$(\sigma, c, \chi)^\# \ := \ (\sigma, \bar{c}, \bar{\chi}) \,, \tag{A.71}$$

where $\bar{c}$ and $\bar{\chi}$ denote complex conjugation of $c$ and $\chi$ w.r.t their coefficients in $U(1)$.

## A.4 Direct Sum and Tensor Product

From the theory of ordinary representations of finite groups $G$ we are used to being able to combine two given representations of $G$ into new representations by taking direct sums and tensor products. The description of 2-representations of $\mathcal{G}$ as triples $(\sigma, c, \chi)$ as in Theorem A.24 allows us to introduce analogous constructions for 2-representations:

**Definition A.28.** Let $(\sigma, c, \chi)$ and $(\sigma', c', \chi')$ be two 2-representations of $\mathcal{G}$ of dimensions $n$ and $n'$. Then, we can combine them as follows:

- Their *direct sum* is the $(n + n')$-dimensional 2-representation

$$(\sigma, c, \chi) \oplus (\sigma', c', \chi') := (\sigma \oplus \sigma', c \oplus c', \chi \oplus \chi'), \tag{A.72}$$

  where the permutation representation $\sigma \oplus \sigma' : G \to S_{n+n'}$ is defined by

$$(\sigma \oplus \sigma')_g(i) := \begin{cases} \sigma_g(i) & \text{if } 1 \leq i \leq n \\ \sigma'_g(i - n) + n & \text{if } n + 1 \leq i \leq n + n' \end{cases}, \tag{A.73}$$

  the twisted 2-cocycle $c \oplus c' \in Z^2_{\sigma \oplus \sigma'}(G, U(1)^{n+n'})$ is given by

$$(c \oplus c')_i(g, h) := \begin{cases} c_i(g, h) & \text{if } 1 \leq i \leq n \\ c'_{i-n}(g, h) & \text{if } n + 1 \leq i \leq n + n' \end{cases}, \tag{A.74}$$

  and the collection of characters $\chi \oplus \chi' \in (A^\vee)^{n+n'}$ is taken to be

$$(\chi \oplus \chi')_i := \begin{cases} \chi_i & \text{if } 1 \leq i \leq n \\ \chi'_{i-n} & \text{if } n + 1 \leq i \leq n + n' \end{cases} \tag{A.75}$$

  for all $g, h \in G$ and $i \in \langle n + n' \rangle$,

- Their *tensor product* is the $(n \cdot n')$-dimensional 2-representation

$$(\sigma, c, \chi) \otimes (\sigma', c', \chi') := (\sigma \otimes \sigma', c \otimes c', \chi \otimes \chi'), \tag{A.76}$$

  where the permutation representation $\sigma \otimes \sigma' : G \to S_{n \cdot n'}$ is defined by

$$(\sigma \otimes \sigma')_g(i, j) := (\sigma_g(i), \sigma'_g(j)), \tag{A.77}$$

  the twisted 2-cocycle $c \otimes c' \in Z^2_{\sigma \otimes \sigma'}(G, U(1)^{n \cdot n'})$ is given by

$$(c \otimes c')_{(i,j)}(g, h) := c_i(g, h) \cdot c_j(g, h)', \tag{A.78}$$

  and the collection of characters $\chi \otimes \chi' \in (A^\vee)^{n \cdot n'}$ is taken to be

$$(\chi \otimes \chi')_{(i,j)} := \chi_i \cdot \chi_j \tag{A.79}$$

  for all $g, h \in G$ and $(i, j) \in \langle n \rangle \times \langle n' \rangle \simeq \langle n \cdot n' \rangle$.

### A.5 Induction and Restriction

From the theory of ordinary representations of finite groups $G$ we are used to being able to construct representations of $G$ from representations of subgroups $H \subset G$ by induction. We would like to obtain analogous constructions in the case of 2-representations of finite split 2-groups.

Let therefore $H \subset \mathcal{G}$ be a sub-2-group of $\mathcal{G}$ in the sense of Remark A.23. We denote by $n := |G : H|$ the index of $H$ in $G$. Let $(\sigma, c, \chi)$ be a $m$-dimensional 2-representation of $H$. We would like to construct a $(n \cdot m)$-dimensional 2-representation $(\sigma', c', \chi')$ of $\mathcal{G}$ out of $(\sigma, c, \chi)$. To do this, we consider the space

$$G/H \;=\; \{[R_1], ..., [R_n]\} \tag{A.80}$$

of left $H$-cosets $[R_i] \equiv R_i \cdot H$ in $G$ with fixed representatives $R_i \in G$ such that $[R_1] = H$. Then, each $g \in G$ acts on the left cosets as

$$g \cdot [R_i] \;\overset{!}{=}\; [R_{\eta_g(i)}] \tag{A.81}$$

for some $\eta_g(i) \in \langle n \rangle$, which induces a permutation representation $\eta : G \to S_n$ of $G$ on $\langle n \rangle$. More concretely, Eq. (A.81) means that

$$g \cdot R_i \;=\; R_{\eta_g(i)} \cdot h_i(g) \tag{A.82}$$

with $h_i(g) \in H$ for each $i \in \langle n \rangle$ and $g \in G$. Using this, we define an induced permutation representation $\sigma' : G \to S_{n \cdot m}$ of $G$ on $\langle n \cdot m \rangle$ by

$$\sigma'_g(i, j) \;:=\; \big(\eta_g(i),\, \sigma_{h_i(g)}(j)\big). \tag{A.83}$$

Furthermore, we can construct an induced twisted 2-cocycle $c' \in Z^2_{\sigma'}(G, U(1)^{n \cdot m})$ on $G$ as

$$c'_{(i,j)}(g, g') \;:=\; c_j\big(h_{\eta_g^{-1}(i)}(g),\, h_{\eta_{g \cdot g'}^{-1}(i)}(g')\big). \tag{A.84}$$

Lastly, we obtain an induced collection $\chi' \in (A^\vee)^{n \cdot m}$ of $(n \cdot m)$ characters of $A$ by

$$\chi'_{(i,j)}(a) \;:=\; \chi_j(R_i^{-1} \triangleright_\rho a). \tag{A.85}$$

One can then check that the triple $(\sigma', c', \chi')$ forms a well-defined 2-representation of $\mathcal{G}$, whose equivalence class is independent of the choice of representatives $R_i$ of left $H$-cosets in $G$. We name it as follows:

**Definition A.29.** The 2-representation $(\sigma', c', \chi')$ is called the *induction* of the 2-representation $(\sigma, c, \chi)$ from $H$ to $\mathcal{G}$ and is denoted by

$$(\sigma', c', \chi') \;=:\; \mathrm{Ind}_H^{\mathcal{G}}(\sigma, c, \chi). \tag{A.86}$$

A natural question to ask is whether two 2-representations of two different sub-2-groups of $\mathcal{G}$ give rise to equivalent 2-representations of $\mathcal{G}$ after induction. To answer this question,

we note that, given a 2-representation $(\sigma, c, \chi)$ of $H \subset \mathcal{G}$ and a group element $g \in G$, we can define a 2-representation $({}^g\sigma, {}^gc, {}^g\chi)$ of ${}^gH := gHg^{-1} \subset \mathcal{G}$ by setting

$$^g\sigma \; := \; \sigma \circ \mathrm{conj}_{g^{-1}}\,, \tag{A.87}$$

$$^gc \; := \; c \circ \mathrm{conj}_{g^{-1}}\,, \tag{A.88}$$

$$^g\chi(.) \; := \; \chi(g^{-1} \triangleright_\rho (.))\,. \tag{A.89}$$

**Definition A.30.** The 2-representation $^g(\sigma, c, \chi) := ({}^g\sigma, {}^gc, {}^g\chi)$ of $^gH \subset \mathcal{G}$ is called the *conjugation* of the 2-representation $(\sigma, c, \chi)$ of $H \subset \mathcal{G}$ by $g \in G$.

One can then check by an explicit calculation that conjugating 2-representations of sub-2-groups leads to equivalent 2-representations of $\mathcal{G}$ after induction:

**Lemma A.31.** Let $H \subset \mathcal{G}$ a sub-2-group and let $(\sigma, c, \chi)$ be a 2-representations of $H$. Then, for any $g \in G$ it holds that

$$\mathrm{Ind}_H^{\mathcal{G}}(\sigma, c, \chi) \; \cong \; \mathrm{Ind}_{^gH}^{\mathcal{G}}\big({}^g(\sigma, c, \chi)\big)\,. \tag{A.90}$$

On the other hand, we know that, given a representation of a finite group $G$, we can restrict it to obtain a representation of a subgroup $H \subset G$. Analogously, given a 2-representation $(\sigma, c, \chi)$ of $\mathcal{G}$ and a sub-2-group $H \subset \mathcal{G}$, we can construct a triple

$$\sigma' \; := \; \sigma|_H\,, \qquad c' \; := \; \imath^*(c)\,, \qquad \chi' \; := \; \chi\,, \tag{A.91}$$

where $\imath : H \hookrightarrow G$ denotes the inclusion of $H$ into $G$. It is then clear that $(\sigma', c', \chi')$ forms a well-defined 2-representation of $H \subset \mathcal{G}$, which we name as follows:

**Definition A.32.** The 2-representation $(\sigma', c', \chi')$ is called the *restriction* of the 2-representation $(\sigma, c, \chi)$ from $\mathcal{G}$ to $H$ and is denoted by

$$(\sigma', c', \chi') \; =: \; \mathrm{Res}_H^{\mathcal{G}}(\sigma, c, \chi)\,. \tag{A.92}$$

A natural question to ask is how induction and restriction of 2-representations interplay with one another. This is answered by *Mackey's decomposition theorem*:

**Theorem A.33.** Let $K$ and $H$ be two sub-2-groups of $\mathcal{G}$ and let $(\sigma, c, \chi)$ be a $m$-dimensional 2-representation of $K$. Then, it holds that

$$\big(\mathrm{Res}_H^{\mathcal{G}} \circ \mathrm{Ind}_K^{\mathcal{G}}\big)(\sigma, c, \chi) \; \cong \; \bigoplus_{[g] \in H\backslash G/K} \mathrm{Ind}_{H \cap {}^gK}^H\big({}^g(\sigma, c, \chi)\big)\,, \tag{A.93}$$

where $g \in G$ labels (arbitrary) representatives of double $H$-$K$-cosets in $G$.

*Proof.* Recall that, by definition, the induction $\mathrm{Ind}_K^{\mathcal{G}}(\sigma, c, \chi)$ of $(\sigma, c, \chi)$ from $K$ to $\mathcal{G}$ can be constructed by considering the space

$$G/K \; = \; \{[R_1], ..., [R_n]\} \tag{A.94}$$

of left $K$-cosets $[R_i] \equiv R_i \cdot K$ with fixed representatives $R_i \in G$ and defining an induced permutation representation $\eta : G \to S_n$ by

$$g \cdot R_i \overset{!}{=} R_{\eta_g(i)} \cdot k_i(g) \tag{A.95}$$

with $k_i(g) \in K$ for each $g \in G$. In order to understand the restriction of $\mathrm{Ind}_K^{\mathcal{G}}(\sigma, c, \chi)$ to $H \subset \mathcal{G}$, we start by decomposing

$$\langle n \rangle = \bigsqcup_{i \in I} O(i) \tag{A.96}$$

into orbits $O(i) \equiv \{\eta_h(i) \,|\, h \in H\}$ of the restricted action $\eta|_H$ of $H$ on $\langle n \rangle$ with fixed representatives $i \in I \subset \langle n \rangle$, whose elements we label as

$$O(i) =: \{i_1, ..., i_{n_i}\}, \tag{A.97}$$

where $i_1 \equiv i$ and $n_i \equiv |O(i)|$. Analogously, we can then decompose

$$\left(\mathrm{Res}_H^{\mathcal{G}} \circ \mathrm{Ind}_K^{\mathcal{G}}\right)(\sigma, c, \chi) \cong \bigoplus_{i \in I} (\sigma^i, c^i, \chi^i), \tag{A.98}$$

where for fixed $i \in I$ we denoted by $(\sigma^i, c^i, \chi^i)$ the $(n_i \cdot m)$-dimensional 2-representation of $H$ defined as

$$\sigma_h^i(j, l) := \left(\theta_h(j), \, \sigma_{k_{i_j}(h)}(l)\right), \tag{A.99}$$

$$c_{(j,l)}^i(h, h') := c_l\left(k_{\eta_h^{-1}(i_j)}(h), \, k_{\eta_{h \cdot h'}^{-1}(i_j)}(h')\right), \tag{A.100}$$

$$\chi_{(j,l)}^i(a) := \chi_l(R_{i_j}^{-1} \rhd_\rho a), \tag{A.101}$$

with $\theta : H \to S_{n_i}$ the permutation action of $H$ on $\langle n_i \rangle$ induced by $\eta$ through

$$\eta_h(i_j) \overset{!}{=} i_{\theta_h(j)}. \tag{A.102}$$

It is then straightforward to check that

$$H_i := \mathrm{Stab}_{\eta|_H}(i) \equiv H \cap (R_i K R_i^{-1}), \tag{A.103}$$

so that, as sets, we have a correspondence

$$O(i) \cong H/H_i =: \{[S_1], ..., [S_{n_i}]\}, \tag{A.104}$$

where we fixed representatives $S_j \in H$ of left $H_i$-cosets $[S_j] \equiv S_j \cdot H_i$ in $H$ such that

$$h \cdot S_j \overset{!}{=} S_{\theta_h(j)} \cdot h_j(h). \tag{A.105}$$

One can check that the elements $h_j(h) \in H_i$ are given by

$$R_i^{-1} \cdot h_j(h) \cdot R_i \equiv k_i(S_{\theta_h(j)})^{-1} \cdot k_{i_j}(h) \cdot k_i(S_j). \tag{A.106}$$

Using the above, we then define a permutation $\tau \in S_{n_i \cdot m}$ by

$$\tau : (j, l) \mapsto \left( j, \sigma^{-1}_{k_i(S_j)}(l) \right), \tag{A.107}$$

which, using (A.106), can be checked to give an equivalence

$$(\sigma^i, c^i, \chi^i) \cong \operatorname{Ind}^H_{H_i} \left( {}^{R_i}(\sigma, c, \chi) \right) \tag{A.108}$$

of 2-representations. Together with (A.98), the claim then follows from the fact that the map $I \to H\backslash G/K$ sending $i \mapsto [R_i]$ is a bijection. $\qquad\square$

**Remark A.34.** Note that, up to equivalence, the choice of representatives $g \in G$ of double $H$-$K$-cosets $[g] \in H\backslash G/K$ in (A.93) does not matter, since choosing different representatives $g' = h \cdot g \cdot k$ for some $h \in H$ and $k \in K$ leads to

$$H \cap {}^{g'}K = {}^h(H \cap {}^gK) \qquad \text{and} \qquad {}^{g'}(\sigma, c, \chi) \cong {}^h({}^g(\sigma, c, \chi)), \tag{A.109}$$

which according to Lemma A.31 gives equivalent 2-representations after induction to $H$.

**Remark A.35.** Note that, on the right-hand side of Mackey's decomposition formula (A.93), the 2-representation ${}^g(\sigma, c, \chi)$ of the intersection $H \cap {}^gK$ should really be seen as the restriction $\operatorname{Res}^{{}^gK}_{H \cap {}^gK}({}^g(\sigma, c, \chi))$. In the following, in order to avoid cumbersome notation, we will leave this restriction understood implicitly.

A special case of the above considerations is when $H = K$ is normal in $G$ (usually denoted by $H \triangleleft G$), which means that ${}^gH = H$ for all $g \in G$. In this case, left $H$-cosets equal right $H$-cosets in $G$, so that Mackey's decomposition formula simplifies as follows:

**Corollary A.36.** Let $H \triangleleft G$ be a normal subgroup and let $(\sigma, c, \chi)$ be a $m$-dimensional 2-representation of $H \subset \mathcal{G}$. Then, it holds that

$$\left( \operatorname{Res}^{\mathcal{G}}_H \circ \operatorname{Ind}^{\mathcal{G}}_H \right)(\sigma, c, \chi) \cong \bigoplus_{[g] \in G/H} {}^g(\sigma, c, \chi), \tag{A.110}$$

where $g \in G$ labels (arbitrary) representatives of left $H$-cosets in $G$.

## A.6 Simplicity

A useful way to study 2-representations of $\mathcal{G}$ is to study a particular subset of 2-representations that form "building blocks" for all other 2-representations of $\mathcal{G}$:

**Definition A.37.** A 2-representation of $\mathcal{G}$ is said to be *simple* if, up to equivalence, it cannot be written as a direct sum of other 2-representations of $\mathcal{G}$.

More concretely, a $n$-dimensional 2-representation $(\sigma, c, \chi)$ of $\mathcal{G}$ is simple if the permutation action $\sigma : G \to S_n$ is transitive on $\langle n \rangle$. The classification of simple 2-representations as "building blocks" is then due to the following:

**Lemma A.38.** Up to equivalence, every 2-representation of $\mathcal{G}$ can be written as a direct sum of simple 2-representations of $\mathcal{G}$.

*Proof.* Let $(\sigma, c, \chi)$ be a $n$-dimensional 2-representation of $\mathcal{G}$. Then, we can decompose

$$\langle n \rangle \;=\; \bigsqcup_{i \in I} O(i) \tag{A.111}$$

into orbits $O(i) \equiv \{\sigma_g(i) \,|\, g \in G\}$ of the permutation action $\sigma$ of $G$ on $\langle n \rangle$ with fixed representatives $i \in I \subset \langle n \rangle$, whose elements we label as

$$O(i) \;=:\; \{i_1, \dots, i_{n_i}\}, \tag{A.112}$$

where $i_1 \equiv i$ and $n_i \equiv |O(i)|$. On each orbit, we then obtain an induced permutation action $\sigma^i : G \to S_{n_i}$ coming from

$$\sigma_g(i_j) \;\overset{!}{=}\; i_{\sigma^i_g(j)}, \tag{A.113}$$

which we can use to decompose

$$(\sigma, c, \chi) \;\cong\; \bigoplus_{i \in I} (\sigma^i, c^i, \chi^i), \tag{A.114}$$

where the 2-cocycles $c^i \in Z^2_{\sigma^i}(G, U(1)^{n_i})$ and characters $\chi^i \in (A^\vee)^{n_i}$ are given by

$$c^i_j \;:=\; c_{i_j} \qquad \text{and} \qquad \chi^i_j \;:=\; \chi_{i_j}. \tag{A.115}$$

Since $G$ acts transitively on each orbit by construction, the $(\sigma^i, c^i, \chi^i)$ then form well-defined simple 2-representation of $\mathcal{G}$ for all $i \in I$. $\qquad \square$

Apart from their building-block nature, simple 2-representations are special since they stem from 1-dimensional 2-representations of sub-2-groups $H \subset \mathcal{G}$. Note that since there are no non-trivial permutation actions on the single-element set $\langle 1 \rangle$, any 1-dimensional 2-representation of $H \subset \mathcal{G}$ is simply labelled by a pair $(u, \alpha)$, where

- $u \in Z^2(H, U(1))$ is an ordinary 2-cocycle on $H$,

- $\alpha \in A^\vee$ is a $H$-invariant character of $A$.

We now state the following:

**Proposition A.39.** Any simple $n$-dimensional 2-representation $(\sigma, c, \chi)$ of $\mathcal{G}$ is equivalent to the induction of a 1-dimensional 2-representation $(u, \alpha)$ of a sub-2-group $H \subset \mathcal{G}$ of index $|G : H| = n$.

*Proof.* Let $(\sigma, c, \chi)$ be a simple $n$-dimensional 2-representation of $\mathcal{G}$. We want to construct a subgroup $H \subset G$ as well as a pair $(u, \alpha)$ as above. To do this, we set

$$H \;:=\; \mathrm{Stab}_\sigma(1) \;\subset\; G \tag{A.116}$$

and define a 2-cochain on $H$ with values in $U(1)$ by

$$u(h, h') \;:=\; c_1(h, h') \tag{A.117}$$

for all $h, h' \in H$, which can be checked to give a well-defined 2-cocycle $u \in Z^2(H, U(1))$ on $H$ (i.e. $\delta u = 1$). Then, we obtain a $H$-invariant character $\alpha \in A^\vee$ by setting

$$\alpha(.) \;:=\; \chi_1(.). \tag{A.118}$$

One can check that $(\sigma, c, \chi) \cong \mathrm{Ind}^{\mathcal{G}}_H(u, \alpha)$ as claimed. $\qquad \square$

**Corollary A.40.** There exists a $n$-dimensional simple 2-representations of $\mathcal{G}$ if and only if $G$ has a subgroup of order $n$. In particular, there exist no simple 2-representations of $\mathcal{G}$ of dimension greater than $|G|$.

**Remark A.41.** Note that the sub-2-group $H \subset \mathcal{G}$ and the 1-dimensional 2-representation $(u, \alpha)$ of $H$ in Proposition A.39 are not unique, since inducing the conjugation ${}^g(u, \alpha)$ of $(u, \alpha)$ up to $\mathcal{G}$ for any $g \in G$ will lead to a simple 2-representation of $\mathcal{G}$ equivalent to $(\sigma, c, \chi)$. Thus, we can label the equivalence class of the simple 2-representation $(\sigma, c, \chi)$ of $\mathcal{G}$ by the equivalence class of a triple $(H, u, \alpha)$ (with $H$, $u$ and $\alpha$ as above), where two triples $(H, u, \alpha)$ and $(H', u', \alpha')$ are considered equivalent if there exists a $g \in G$ such that

$$ H' = {}^g H, \qquad [u'] = [{}^g u], \qquad \alpha' = {}^g \alpha. \tag{A.119} $$

Having classified the simple 2-representations of $\mathcal{G}$, a natural question to ask is how simple 2-representations fuse when taking tensor products. That is, given two simple 2-representations of $\mathcal{G}$, their tensor product must again decompose into simple 2-representations of $\mathcal{G}$, whose form can be determined as follows:

**Proposition A.42.** Let $(u, \alpha)$ and $(v, \beta)$ be two 1-dimensional 2-representations of sub-2-groups $H$ and $K$ of $\mathcal{G}$. Then, the tensor product of their inductions to $\mathcal{G}$ is given by

$$ \mathrm{Ind}_H^{\mathcal{G}}(u, \alpha) \otimes \mathrm{Ind}_K^{\mathcal{G}}(v, \beta) \;\cong\; \bigoplus_{[g] \in H \backslash G / K} \mathrm{Ind}_{H \cap {}^g K}^{\mathcal{G}}\big((u, \alpha) \otimes {}^g(v, \beta)\big), \tag{A.120} $$

where $g \in G$ labels (arbitrary) representatives of double $H$-$K$-cosets in $G$.

*Proof.* Using the *push-pull-formula* for the tensor product of inductions, we see that

$$
\begin{aligned}
\mathrm{Ind}_H^{\mathcal{G}}(u, \alpha) \otimes \mathrm{Ind}_K^{\mathcal{G}}(v, \beta) \;&\cong\; \mathrm{Ind}_H^{\mathcal{G}}\big[(u, \alpha) \otimes (\mathrm{Res}_H^{\mathcal{G}} \circ \mathrm{Ind}_K^{\mathcal{G}})(v, \beta)\big] \\
&\cong\; \bigoplus_{[g] \in H \backslash G / K} \mathrm{Ind}_H^{\mathcal{G}}\big[(u, \alpha) \otimes \mathrm{Ind}_{H \cap {}^g K}^{H}\big({}^g(v, \beta)\big)\big] \\
&\cong\; \bigoplus_{[g] \in H \backslash G / K} \mathrm{Ind}_{H \cap {}^g K}^{\mathcal{G}}\big[(u, \alpha) \otimes {}^g(v, \beta)\big], \tag{A.121}
\end{aligned}
$$

where we used Mackey's decomposition formula from Theorem A.33 in the second line. $\qquad\square$

**Remark A.43.** Note that, as before, the choice of representatives $g \in G$ of double $H$-$K$-cosets $[g] \in H \backslash G / K$ in (A.120) matters only up to equivalence. Furthermore, we again implicitly understand an appropriate restriction of 2-representations on the right-hand side of (A.120).

# B   Graded Projective Representations

In order to understand the 2-category $2\mathrm{Rep}(\mathcal{G})$ of 2-representations of the split 2-group $\mathcal{G}$, it turns out to be useful to study a generalization of the notion of projective representations of an ordinary group $G$, called *graded projective representations*. We will fix $G$ to be a finite group in what follows.

## B.1 The Category

We begin by generalizing the notion of the projective automorphism group $\mathrm{PGL}(V)$ of a vector space $V$ to the notion of the projective automorphism group $\mathrm{PAut}(\mathcal{V})$ of a vector bundle $\mathcal{V} \xrightarrow{\pi} M$. This can be done as follows:

**Definition B.1.** Let $\mathcal{V} \xrightarrow{\pi} M$ be a complex vector bundle over some base space $M$ and let $\mathrm{Aut}(\mathcal{V})$ denote the automorphism group of $\mathcal{V}$. We denote by $U(1)^M$ the abelian normal subgroup of $\mathrm{Aut}(\mathcal{V})$ consisting of maps $f : M \to U(1)$, seen as automorphisms

$$x \in \mathcal{V}_m \quad \mapsto \quad f(m) \cdot x \in \mathcal{V}_m \,. \tag{B.1}$$

Then, the *projective automorphism group* of $\mathcal{V}$ is defined to be

$$\mathrm{PAut}(\mathcal{V}) := \mathrm{Aut}(\mathcal{V}) \,/\, U(1)^M \,. \tag{B.2}$$

In the following, we will be interested in the case where the base space $M$ is finite, i.e. $M = \langle n \rangle$, where $\langle n \rangle \equiv \{1, ..., n\}$ denotes the finite set of $n$ elements. In this case, we have that $U(1)^{\langle n \rangle} \cong U(1)^n$ as groups. We then make the following definition:

**Definition B.2.** A *graded projective representation* of $G$ is a pair $(\mathcal{V}, \Phi)$, where $\mathcal{V}$ is a complex vector bundle $\mathcal{V} \xrightarrow{\pi} \langle n \rangle$ and $\Phi$ is a representative of a group homomorphism $[\Phi] : G \to \mathrm{PAut}(\mathcal{V})$ from $G$ into the projective automorphism group of $\mathcal{V}$. We call $n \in \mathbb{N}$ the *grading* of $(\mathcal{V}, \Phi)$.

More concretely, this means that for each $g \in G$ there is an associated fibre-preserving bundle automorphism $\Phi_g \in \mathrm{Aut}(\mathcal{V})$, which sits in a commutative diagram

$$
\begin{array}{ccc}
\mathcal{V} & \xrightarrow{\ \Phi_g\ } & \mathcal{V} \\
{\scriptstyle \pi}\big\downarrow & & \big\downarrow{\scriptstyle \pi} \\
\langle n \rangle & \xrightarrow{\ \sigma_g\ } & \langle n \rangle
\end{array}
$$

where $\sigma_g \in S_n$ is the corresponding induced bijection on the base space $\langle n \rangle$. This induces a well-defined permutation representation $\sigma : G \to S_n$. The bundle automorphisms $\Phi_g$ themselves however only satisfy the homomorphism property projectively, meaning that

$$\Phi_{g \cdot g'} \;=\; c(g, g') \circ \Phi_g \circ \Phi_{g'} \tag{B.3}$$

for some $c(g, g') \in U(1)^n$, seen as a bundle automorphism as in Definition B.1. This defines a 2-cochain $c : G \times G \to U(1)^n$ in $C^2(G, U(1)^n)$, which, as a consequence of the associativity of the group multiplication in $G$, obeys the twisted cocycle-condition

$$\delta_\sigma(c) \;=\; 0 \,, \tag{B.4}$$

where $\delta_\sigma$ denotes the nilpotent differential on $C^2(G, U(1)^n)$ twisted by $\sigma$. Thus, we obtain a well-defined 2-cocycle $c \in Z_\sigma^2(G, U(1)^n)$, which together with $\sigma$ classifies the graded projective representation $(\mathcal{V}, \Phi)$ in the following sense:

**Definition B.3.** The pair $(\sigma, c)$ is called the *obstruction pair* of the graded projective representation $(\mathcal{V}, \Phi)$ of $G$. To see what it obstructs, we consider the following cases:

- $\sigma = 1$: In this case, $(\mathcal{V}, \Phi)$ consists of $n$ decoupled vector spaces $\mathcal{V}_i$ equipped with projective $G$-actions $\Phi_i := \Phi|_{\mathcal{V}_i} \in \mathrm{PGL}(\mathcal{V}_i)$ of cocycles $c_i \in Z^2(G, U(1))$. We say that that the graded projective representations $(\mathcal{V}, \Phi)$ *splits*.

- $c = \delta_\sigma(b)$: In this case, we can redefine $\Phi_g \to \widehat{\Phi}_g := b_g \circ \Phi_g$ for each $g \in G$, which turns $\widehat{\Phi} : G \to \mathrm{Aut}(\mathcal{V})$ into a group homomorphism. We say that $(\mathcal{V}, \Phi)$ can be *lifted*.

**Notation:** Given a $n$-graded projective representation $(\mathcal{V}, \Phi)$, we will speak of its *support* as the subset of $\langle n \rangle$ whose fibres are non-trivial, i.e.

$$\mathrm{Sup}(\mathcal{V}) := \{i \in \langle n \rangle \mid \mathcal{V}_i \neq 0\}. \tag{B.5}$$

In the following, we would like to study the "category" of graded projective representations of $G$. In order to make this category well-defined, we need to define what we mean by a morphism between two graded projective representations:

**Definition B.4.** A *morphism* between graded projective representations $(\mathcal{V}, \Phi)$ and $(\mathcal{V}', \Phi')$ of $G$ of gradings $n$ and $n'$ is a vector bundle morphism $\varphi : \mathcal{V} \to \mathcal{V}'$ such that

$$[\varphi \circ \Phi] = [\Phi' \circ \varphi]. \tag{B.6}$$

We call $\varphi$ an *isomorphism* if it is a vector bundle isomorphism. In this case, we say that the two graded projective representations are *isomorphic* and write $(\mathcal{V}, \Phi) \cong (\mathcal{V}', \Phi')$.

Note that a necessary condition for an isomorphism between $(\mathcal{V}, \Phi)$ and $(\mathcal{V}', \Phi')$ to exist is that their gradings $n$ and $n'$ coincide, i.e. $n = n'$. In this case, it is straightforward to establish a relationship between the corresponding obstruction pairs:

**Proposition B.5.** Let $\varphi$ be an isomorphism between graded projective representations $(\mathcal{V}, \Phi)$ and $(\mathcal{V}', \Phi')$ of $G$. Then, their obstruction pairs $(\sigma, c)$ and $(\sigma', c')$ are related by

$$\sigma' = \tau \circ \sigma \circ \tau^{-1} \quad \text{and} \quad [c'] = [\tau \triangleright c], \tag{B.7}$$

where $\tau : \langle n \rangle \to \langle n \rangle$ is the bijection on the base spaces induced by $\varphi$.

*Proof.* Since $\varphi$ is an isomorphism between $(\mathcal{V}, \Phi)$ and $(\mathcal{V}', \Phi')$, we know that there exists a 1-cochain $b \in C^1(G, U(1)^n)$ such that

$$\varphi \circ \Phi_g = b_g \circ \Phi'_g \circ \varphi \tag{B.8}$$

for all $g \in G$. We can embed this relation into a cuboid of maps

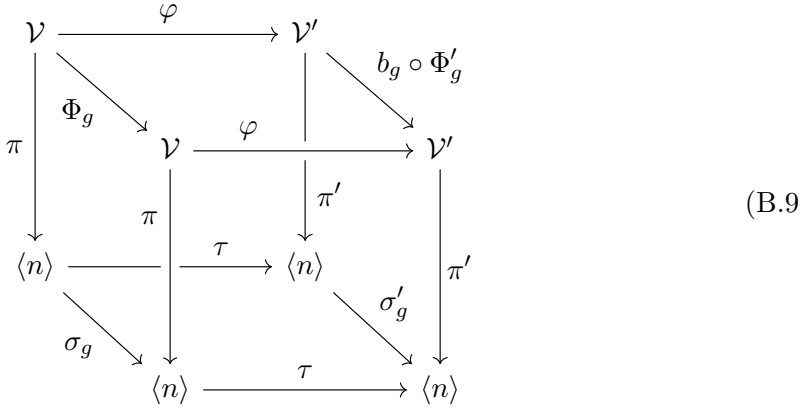

$$\text{(B.9)}$$

where the upper square commutes by construction. One can then check that projecting down on the lower square via $\pi$ and $\pi'$ implies that $\sigma'_g \circ \tau = \tau \circ \sigma_g$ for all $g \in G$. Secondly, we note that for any bundle isomorphism $\psi : \mathcal{V} \to \mathcal{V}'$ and any $u \in U(1)^n$ it holds that

$$\psi \circ u \; = \; (\kappa \triangleright u) \circ \psi \,, \tag{B.10}$$

where $\kappa \in S_n$ is the bijection on the bases spaces induced by $\psi$. Using this, we see that

$$
\begin{aligned}
\varphi \circ \Phi_{g_1 \cdot g_2} \; &= \; \varphi \circ c(g_1, g_2) \circ \Phi_{g_1} \circ \Phi_{g_2} \\
&= \; (\tau \triangleright c(g_1, g_2)) \circ b_{g_1} \circ (\sigma'_{g_1} \triangleright b_{g_2}) \circ \Phi'_{g_1} \circ \Phi'_{g_2} \circ \varphi \,, \\
\varphi \circ \Phi_{g_1 \cdot g_2} \; &= \; b_{g_1 \cdot g_2} \circ \Phi'_{g_1 \cdot g_2} \circ \varphi \\
&= \; b_{g_1 \cdot g_2} \circ c'(g_1, g_2) \circ \Phi'_{g_1} \circ \Phi'_{g_2} \circ \varphi \,.
\end{aligned}
\tag{B.11}
$$
$$\tag{B.12}$$

Comparing (B.11) and (B.12) and using that $\Phi_{g_1}$, $\Phi_{g_2}$ and $\varphi$ are invertible shows that

$$
\begin{aligned}
c'(g_1, g_2) \; &= \; [(\sigma'_{g_1} \triangleright b_{g_2}) \circ b_{g_1 \cdot g_2}^{-1} \circ b_{g_1}] \circ (\tau \triangleright c(g_1, g_2)) \\
&\equiv \; (\delta_{\sigma'} b)(g_1, g_2) \circ (\tau \triangleright c(g_1, g_2)) \,,
\end{aligned}
\tag{B.13}
$$

which implies $[c'] \equiv [\tau \triangleright c]$ as group cohomology classes in $H^2_{\sigma'}(G, U(1)^n)$. $\qquad \square$

**Notation:** In the following, we will denote the category of $n$-graded projective representations of $G$ with fixed obstruction pair $(\sigma, c)$ and support $S \subset \langle n \rangle$ by

$$\mathrm{Rep}_S^{(\sigma, c)}(G) \,. \tag{B.14}$$

## B.2  Direct Sum and Tensor Product

Just as ordinary representations of a group $G$ can be added and multiplied, there exists a notion of direct sums and tensor products for graded projective representations:

**Definition B.6.** Two graded projective representations $(\mathcal{V}, \Phi)$ and $(\mathcal{V}', \Phi')$ of $G$ with gradings $n$ and $n'$ can be combined to form new graded projective representations as follows:

- Their *direct sum* is the $(n + n')$-graded projective representation

$$(\mathcal{V}, \Phi) \oplus (\mathcal{V}', \Phi') = (\mathcal{V} \oplus \mathcal{V}', \Phi \oplus \Phi'), \tag{B.15}$$

where the vector bundle $\mathcal{V} \oplus \mathcal{V}' \xrightarrow{\pi} \langle n + n' \rangle$ is defined by

$$(\mathcal{V} \oplus \mathcal{V}')_i := \begin{cases} \mathcal{V}_i & \text{if } 1 \leq i \leq n \\ \mathcal{V}'_{i-n} & \text{if } n + 1 \leq i \leq n + n' \end{cases} \tag{B.16}$$

and $(\Phi \oplus \Phi') : G \to \mathrm{Aut}(\mathcal{V} \oplus \mathcal{V}')$ is given by

$$(\Phi \oplus \Phi')_g|_i := \begin{cases} \Phi_g|_{\mathcal{V}_i} & \text{if } 1 \leq i \leq n \\ \Phi'_g|_{\mathcal{V}'_{i-n}} & \text{if } n + 1 \leq i \leq n + n' \end{cases} \tag{B.17}$$

for all $g \in G$ and $i \in \langle n + n' \rangle$.

- Their *tensor product* is the $(n \cdot n')$-graded projective representation

$$(\mathcal{V}, \Phi) \otimes (\mathcal{V}', \Phi') = (\mathcal{V} \otimes \mathcal{V}', \Phi \otimes \Phi'), \tag{B.18}$$

where the vector bundle $\mathcal{V} \otimes \mathcal{V}' \xrightarrow{\pi} \langle n \cdot n' \rangle$ is defined by

$$(\mathcal{V} \otimes \mathcal{V}')_{(i,j)} := \mathcal{V}_i \otimes \mathcal{V}'_j \tag{B.19}$$

and $(\Phi \otimes \Phi') : G \to \mathrm{Aut}(E \otimes E')$ is given by

$$(\Phi \otimes \Phi')_g|_{(i,j)} := (\Phi_g|_{\mathcal{V}_i}) \otimes (\Phi_g|_{\mathcal{V}'_j}) \tag{B.20}$$

for all $g \in G$ and $(i, j) \in \langle n \rangle \times \langle n' \rangle \simeq \langle n \cdot n' \rangle$.

It is natural to ask how the obstruction pairs of $(\mathcal{V}, \Phi)$ and $(\mathcal{V}', \Phi')$ behave when taking direct sums and tensor products. One can check that this is answered as follows:

**Proposition B.7.** Let $(\mathcal{V}, \Phi)$ and $(\mathcal{V}', \Phi')$ be two graded projective representations of $G$ with corresponding obstruction pairs $(\sigma, c)$ and $(\sigma', c')$. Then,

- the direct sum $(\mathcal{V}, \Phi) \oplus (\mathcal{V}', \Phi')$ has obstruction pair $(\sigma \oplus \sigma', c \oplus c')$, where $\sigma \oplus \sigma'$ and $c \oplus c'$ are as in (A.73) and (A.74),

- the tensor product $(\mathcal{V}, \Phi) \otimes (\mathcal{V}', \Phi')$ has obstruction pair $(\sigma \otimes \sigma', c \otimes c')$, where $\sigma \otimes \sigma'$ and $c \otimes c'$ are as in (A.77) and (A.78).

For later purposes, it is useful to understand how the support of graded projective representations behaves under direct sums and tensor products. To see this, first note that given two subsets $A \subset \langle n \rangle$ and $B \subset \langle n' \rangle$, we can define their direct sum $A \oplus B \subset \langle n + n' \rangle$ and tensor product $A \otimes B \subset \langle n \cdot n' \rangle$ by

$$A \oplus B := \{ i \in \langle n + n' \rangle \mid i \in A \text{ or } i - n' \in B \}, \tag{B.21}$$

$$A \otimes B := \{ (i, j) \in \langle n \cdot n' \rangle \mid i \in A \text{ and } j \in B \}, \tag{B.22}$$

It is then straightforward to check the following:

**Lemma B.8.** Let $(\mathcal{V}, \Phi)$ and $(\mathcal{V}', \Phi')$ be two graded projective representations of $G$. Then,

$$\mathrm{Sup}(\mathcal{V} \oplus \mathcal{V}') = \mathrm{Sup}(\mathcal{V}) \oplus \mathrm{Sup}(\mathcal{V}'), \tag{B.23}$$

$$\mathrm{Sup}(\mathcal{V} \otimes \mathcal{V}') = \mathrm{Sup}(\mathcal{V}) \otimes \mathrm{Sup}(\mathcal{V}'). \tag{B.24}$$

### B.3 Induction and Restriction

Just as ordinary representations can be induced from and restricted to subgroups of $G$, there exists a notion of induction and restriction for graded projective representations: Let $H \subset G$ be a subgroup of $G$ of index $|G : H| = n$, and let $(\mathcal{V}, \Phi)$ be a $m$-graded projective representation of $H$. We would like to construct a $(n \cdot m)$-graded projective representation $(\mathcal{V}', \Phi')$ of $G$ out of $(\mathcal{V}, \Phi)$. As before, we consider the space

$$G/H \ = \ \{[R_1], ..., [R_n]\} \tag{B.25}$$

of left cosets $[R_i] \equiv R_i \cdot H$ in $G$ with fixed representatives $R_i \in G$ such that $[R_1] = H$. We denote by $\eta : G \to S_n$ the induced permutation action coming from

$$g \cdot R_i \ \overset{!}{=} \ R_{\eta_g(i)} \cdot h_i(g) \tag{B.26}$$

with $h_i(g) \in H$, which allows us to define the vector bundle $\mathcal{V}' \to \langle n \cdot m \rangle$ by

$$\mathcal{V}'_{(i,j)} \ := \ R_i \otimes \mathcal{V}_j \,. \tag{B.27}$$

Furthermore, we can define the projective homomorphism $\Phi' : G \to \mathrm{Aut}(\mathcal{V}')$ by

$$\Phi'_g|_{(i,j)} \ := \ \eta_g \otimes \Phi_{h_i(g)}|_j \tag{B.28}$$

for all $g \in G$ and $(i, j) \in \langle m \rangle \times \langle n \rangle \simeq \langle m \cdot n \rangle$. One can then check that $(\mathcal{V}', \Phi')$ forms a well-defined graded projective representation of $G$, whose isomorphism class is independent of the choice of representatives $R_i$ of left $H$-cosets in $G$. We name it as follows:

**Definition B.9.** The graded projective representation $(\mathcal{V}', \Phi')$ is called the *induction* of $(\mathcal{V}, \Phi)$ from $H$ to $G$ and is denoted by

$$(\mathcal{V}', \Phi') \ =: \ \mathrm{Ind}_H^G(\mathcal{V}, \Phi) \,. \tag{B.29}$$

A natural question to ask is how the obstruction pair of the induction $\mathrm{Ind}_H^G(\mathcal{V}, \Phi)$ is related to the obstruction pair of $(\mathcal{V}, \Phi)$:

**Proposition B.10.** Let $(\mathcal{V}, \Phi)$ be a graded projective representation of $H \subset G$ with obstruction pair $(\sigma, c)$. Then, its induction $\mathrm{Ind}_H^G(\mathcal{V}, \Phi)$ to $G$ has obstruction pair $(\sigma', c')$, where $\sigma'$ and $c'$ are as in (A.83) and (A.84).

Similarly to the case of 2-representations, we can ask whether two different sub-groups of $G$ give rise to isomorphic graded projective representations of $G$ after induction. To answer this question, we note that, given a graded projective representation $(\mathcal{V}, \Phi)$ of $H \subset G$ and a group element $g \in G$, we can define a graded projective representation $({}^g\mathcal{V}, {}^g\Phi)$ of ${}^gH \equiv gHg^{-1} \subset G$ by setting

$${}^g\mathcal{V} := \mathcal{V} \qquad \text{and} \qquad {}^g\Phi := \Phi \circ \mathrm{conj}_{g^{-1}} \,. \tag{B.30}$$

**Definition B.11.** The graded projective representation ${}^g(\mathcal{V}, \Phi) := ({}^g\mathcal{V}, {}^g\Phi)$ of ${}^gH \subset G$ is called the *conjugation* of the graded projective representation $(\mathcal{V}, \Phi)$ of $H \subset G$ by $g \in G$.

One can then check that conjugating graded projective representations of sub-groups leads to isomorphic graded projective representations after induction to $G$:

**Lemma B.12.** Let $H \subset G$ a sub-group and let $(\mathcal{V}, \Phi)$ be a graded projective representation of $H$. Then, for any $g \in G$ it holds that

$$\operatorname{Ind}_H^G(\mathcal{V}, \Phi) \cong \operatorname{Ind}_{gH}^G\left({}^g(\mathcal{V}, \Phi)\right). \tag{B.31}$$

On the other hand, given an $n$-graded projective representation $(\mathcal{V}, \Phi)$ of $G$, we can restrict it to obtain a $n$-graded projective representation $(\mathcal{V}', \Phi')$ of $H \subset G$ by setting

$$\mathcal{V}' := \mathcal{V} \qquad \text{and} \qquad \Phi' := \Phi|_H. \tag{B.32}$$

**Definition B.13.** The graded projective representation $(\mathcal{V}', \Phi')$ is called the *restriction* of $(\mathcal{V}, \Phi)$ from $G$ to $H$ and is denoted by

$$(\mathcal{V}', \Phi') =: \operatorname{Res}_H^G(\mathcal{V}, \Phi). \tag{B.33}$$

Again, we can ask how the obstruction pair of the restriction $\operatorname{Res}_H^G(\mathcal{V}, \Phi)$ is related to the obstruction pair of $(\mathcal{V}, \Phi)$:

**Proposition B.14.** Let $(\mathcal{V}, \Phi)$ be a graded projective representation of $G$ with obstruction pair $(\sigma, c)$. Then, its restriction $\operatorname{Res}_H^G(\mathcal{V}, \Phi)$ to $H \subset G$ has obstruction pair $(\sigma', c')$, where $\sigma'$ and $c'$ are as in (A.91).

A natural question to ask is how induction and restriction interplay with one another. This is again answered by Mackey's decomposition theorem:

**Theorem B.15.** Let $H, K \subset G$ be two subgroups of $G$ and let $(\mathcal{V}, \Phi)$ be a $m$-graded projective representation of $K$. Then:

$$\left(\operatorname{Res}_H^G \circ \operatorname{Ind}_K^G\right)(\mathcal{V}, \Phi) \cong \bigoplus_{[g] \in H \backslash G / K} \operatorname{Ind}_{H \cap {}^gK}^H\left({}^g(\mathcal{V}, \Phi)\right), \tag{B.34}$$

where $g \in G$ labels (arbitrary) representatives of double $H$-$K$-cosets in $G$.

The proof is analogous to the proof of Theorem A.33 describing Mackey's decomposition for 2-representations. Note that, again, the choice of representatives $g \in G$ of double $H$-$K$-cosets $[g] \in H \backslash G / K$ in (B.34) matters only up to isomorphism. Furthermore, we again understand an implicit restriction of the graded projective representation ${}^g(\mathcal{V}, \Phi)$ to the intersection $H \cap {}^gK$ on the right-hand side of (B.34).

Similarly to before, Mackey's decomposition formula simplifies in special cases:

**Corollary B.16.** Let $H \lhd G$ be a normal subgroup and let $(\mathcal{V}, \Phi)$ be a graded projective representations of $H$. Then, it holds that

$$\left(\operatorname{Res}_H^G \circ \operatorname{Ind}_H^G\right)(\mathcal{V}, \Phi) \cong \bigoplus_{[g] \in G / H} {}^g(\mathcal{V}, \Phi), \tag{B.35}$$

where $g \in G$ labels (arbitrary) representatives of left $H$-cosets in $G$. In particular, if $G$ is abelian (in which case every subgroup of $G$ is normal), we have that

$$\left(\operatorname{Res}_H^G \circ \operatorname{Ind}_H^G\right)(\mathcal{V}, \Phi) \cong |G : H| \cdot (\mathcal{V}, \Phi). \tag{B.36}$$

## B.4  Simplicity

A useful way to study graded projective representations is again to study a particular subset that forms "building blocks" for all other graded projective representations:

**Definition B.17.** A graded projective representation of $G$ is *simple* if, up to isomorphism, it cannot be written as a direct sum of other graded projective representations.

More concretely, a graded projective representation $(\mathcal{V}, \Phi)$ is simple if the corresponding permutation action $\sigma : G \to S_n$ in its obstruction pair acts *transitively* on the base space $\langle n \rangle$ of $\mathcal{V}$. The labelling of simple graded projective representations as "building blocks" is then due to the following:

**Lemma B.18.** Every graded projective representation of $G$ is isomorphic to a direct sum of simple graded projective representations.

*Proof.* Let $(\mathcal{V}, \Phi)$ be a $n$-graded projective representation of $G$ with obstruction pair $(\sigma, c)$. Then, we can decompose

$$\langle n \rangle \;=\; \bigsqcup_{i \in I} O(i) \tag{B.37}$$

into orbits $O(i) \equiv \{\sigma_g(i) \,|\, g \in G\}$ of the permutation action $\sigma$ of $G$ on $\langle n \rangle$ with fixed representatives $i \in I \subset \langle n \rangle$, whose elements we label as

$$O(i) \;=:\; \left\{ i_1, \ldots, i_{n_i} \right\}, \tag{B.38}$$

where $i_1 \equiv i$ and $n_i \equiv |O(i)|$. On each orbit, we then obtain an induced permutation action $\sigma^i : G \to S_{n_i}$ coming from

$$\sigma_g(i_j) \;\overset{!}{=}\; i_{\sigma_g^i(j)}, \tag{B.39}$$

which we can use to decompose

$$(\mathcal{V}, \Phi) \;\cong\; \bigoplus_{i \in I} (\mathcal{V}^i, \Phi^i), \tag{B.40}$$

where the vector bundles $\mathcal{V}^i \to \langle n_i \rangle$ and bundle automorphisms $\Phi^i : \mathcal{V}^i \to \mathcal{V}^i$ are given by

$$\mathcal{V}^i_j \;:=\; \mathcal{V}_{i_j} \qquad \text{and} \qquad \Phi^i_g|_j \;:=\; \Phi_g|_{i_j}. \tag{B.41}$$

Since $G$ acts transitively on each orbit by construction, the $(\mathcal{V}^i, \Phi^i)$ form well-defined simple graded projective representations of $G$ for all $i \in I$. $\qquad\square$

Apart from their building-block nature, simple graded projective representations of $G$ are special since they can be obtained from 1-graded projective representations of subgroups $H \subset G$. Note that a 1-graded projective representation of $H \subset G$ is simply a pair $(V, \varphi)$, where

- $V$ is a complex vector space,

- $\varphi : H \to \mathrm{GL}(V)$ projective representation of $H$ on $V$.

We now state the following:

**Proposition B.19.** Any simple $n$-graded projective representation $(\mathcal{V}, \Phi)$ of $G$ is isomorphic to the induction of a 1-graded projective representation $(V, \varphi)$ of a sub-group $H \subset G$ of index $|G : H| = n$.

*Proof.* Let $(\mathcal{V}, \Phi)$ be a simple $n$-graded projective representation of $G$ with obstruction pair $(\sigma, c)$. We can construct a subgroup $H \subset G$ by setting

$$H \; := \; \mathrm{Stab}_\sigma(1) \; \subset \; G \,, \tag{B.42}$$

and obtain a projective representation $(V, \varphi)$ of $H$ by defining

$$V \; := \; \mathcal{V}_1 \qquad \text{and} \qquad \varphi \; := \; (\Phi|_H)|_{\mathcal{V}_1} \,. \tag{B.43}$$

One can then check that $(\mathcal{V}, \Phi) \cong \mathrm{Ind}_H^G(V, \varphi)$ as claimed. $\qquad\qquad\square$

**Remark B.20.** Note that the sub-group $H \subset G$ and the projective representation $(V, \varphi)$ of $H$ in Proposition B.19 are not unique, since inducing the conjugation ${}^g(V, \varphi)$ of $(V, \varphi)$ up to $G$ for any $g \in G$ will lead to a simple graded projective representation of $G$ isomorphic to $(\mathcal{V}, \Phi)$. Thus, we can label the isomorphism class of the simple graded projective representation $(\mathcal{V}, \Phi)$ of $G$ by the equivalence class of a triple $(H, V, \varphi)$ (with $H$, $V$ and $\varphi$ as above), where two triples $(H, V, \varphi)$ and $(H', V', \varphi')$ are considered equivalent if there exists a $g \in G$ such that

$$H' \; = \; {}^gH \,, \qquad V' \; = \; V \,, \qquad \varphi' \; = \; {}^g\varphi \,. \tag{B.44}$$

Having classified the simple graded projective representations of $G$, a natural question to ask is how simple graded projective representations fuse when taking tensor products:

**Proposition B.21.** Let $(V, \varphi)$ and $(W, \psi)$ be two projective representations of sub-groups $H$ and $K$ of $G$. Then, the tensor product of their inductions to $G$ is given by

$$\mathrm{Ind}_H^G(V, \varphi) \otimes \mathrm{Ind}_K^G(W, \psi) \; \cong \; \bigoplus_{[g] \in H\backslash G/K} \mathrm{Ind}_{H \cap {}^gK}^G\big((V, \varphi) \otimes {}^g(W, \psi)\big) \,, \tag{B.45}$$

where $g \in G$ labels (arbitrary) representatives of double $H$-$K$-cosets in $G$.

The proof is analogous to the proof of Proposition A.42 for the fusion of simple 2-representations, using the push-pull-formula for the tensor product of inductions. Note that, again, the choice of representatives $g \in G$ of double $H$-$K$-cosets $[g] \in H\backslash G/K$ in (B.45) matters only up to isomorphism. Furthermore, we again implicitly understand an appropriate restriction of graded projective representations on the right-hand side of (B.45).

### B.5 Primality

Just as simple graded projective representations form building blocks for general graded projective representations w.r.t. direct sums, we can introduce a notion of building blocks for graded projective representations w.r.t. tensor products:

**Definition B.22.** A graded projective representation of $G$ is said to be *prime* if, up to isomorphism, it cannot be written as a tensor product of other (non-trivial) graded projective representations.

Here, we think of the "trivial" graded projective representation as the the 1-graded projective representation with trivial fibre $\mathbb{C}$. From Proposition B.7 we know that if a graded projective representation $(\mathcal{V}, \Phi)$ factorises as a non-trivial tensor product, so does its obstruction pair $(\sigma, c)$. The converse need not be true in general, as the following construction shows:

**Definition B.23.** Two graded projective representations $(\mathcal{V}, \Phi)$ and $(\mathcal{V}', \Phi')$ of $G$ are said to be *composable* if their obstruction pairs $(\sigma, c)$ and $(\sigma', c')$ factorise as

$$(\sigma, c) \;=\; (\sigma_1, c_1)^{\#} \otimes (\sigma_2, c_2), \tag{B.46}$$

$$(\sigma', c') \;=\; (\sigma_2, c_2)^{\#} \otimes (\sigma_3, c_3). \tag{B.47}$$

for some permutation actions $\sigma_i : G \to S_{n_i}$ and 2-cocycles $c_i \in Z^2_{\sigma_i}(G, U(1)^{n_i})$.

Given two such composable graded projective representations $(\mathcal{V}, \Phi)$ and $(\mathcal{V}', \Phi')$, we can construct a new graded projective representation $(\mathcal{V}'', \Phi'')$ of $G$ with obstruction pair

$$(\sigma'', c'') \;=\; (\sigma_1, c_1)^{\#} \otimes (\sigma_3, c_3) \tag{B.48}$$

as follows: The fibre of $\mathcal{V}''$ at $(i, k) \in \langle n_1 \rangle \times \langle n_3 \rangle$ is given by

$$\mathcal{V}''_{(i,k)} \;:=\; \bigoplus_{j=1}^{n_2} \left( \mathcal{V}_{(i,j)} \otimes \mathcal{V}'_{(j,k)} \right), \tag{B.49}$$

which is acted upon by each $g \in G$ through the projective automorphism

$$\Phi''_g|_{(i,k)} \;:=\; \bigoplus_{j=1}^{n_2} \left( \Phi_g|_{(i,j)} \otimes \Phi'_g|_{(j,k)} \right). \tag{B.50}$$

One can check that this yields a well-defined $(n_1 \cdot n_3)$-graded projective representation of $G$, which we label as follows:

**Definition B.24.** The graded projective representations $(\mathcal{V}'', \Phi'')$ is called the *composition* of $(\mathcal{V}, \Phi)$ and $(\mathcal{V}', \Phi')$, and denoted by $(\mathcal{V}, \Phi) \circ (\mathcal{V}', \Phi')$.

One can then see from the construction in (B.49) that, even though its obstruction pair $(\sigma_1, c_1)^{\#} \otimes (\sigma_3, c_3)$ factorises, the composition of $(\mathcal{V}, \Phi)$ and $(\mathcal{V}', \Phi')$ itself does not factorise in general as a graded projective representation.

For later purposes, it is useful to understand how the support of graded projective representations behaves under composition. To see this, first note that given two subsets $A \subset \langle n_1 \cdot n_2 \rangle$ and $B \subset \langle n_2 \cdot n_3 \rangle$, we can define their composition $A \circ B \subset \langle n_1 \cdot n_3 \rangle$ by

$$A \circ B \; := \; \{ (i,k) \in \langle n_1 \cdot n_3 \rangle \mid \exists j \in \langle n_2 \rangle : \; (i,j) \in A \; \text{and} \; (j,k) \in B \}. \qquad (B.51)$$

It is then straightforward to check the following:

**Lemma B.25.** The support of the composition $(\mathcal{V}, \Phi) \circ (\mathcal{V}', \Phi')$ of two composable graded projective representations $(\mathcal{V}, \Phi)$ and $(\mathcal{V}', \Phi')$ is given by

$$\mathrm{Sup}(\mathcal{V} \circ \mathcal{V}') \; = \; \mathrm{Sup}(\mathcal{V}) \circ \mathrm{Sup}(\mathcal{V}'). \qquad (B.52)$$

## C The 2-Category of 2-Representations

We are now in the position to describe the 2-category $2\mathrm{Rep}(\mathcal{G})$ of 2-representations of the split 2-group $\mathcal{G}$ in more detail. In particular, we will try to describe the 1-morphism spaces of this 2-category, which are themselves categories, as well as their fusion and composition.

### C.1 Morphisms

In order to describe the category of 1-morphisms between two given 2-representations of $\mathcal{G}$, we recall that we denoted by

$$\mathrm{Rep}_S^{(\sigma, c)}(G) \qquad (C.1)$$

the category of graded projective representations of $G$ with fixed obstruction pair $(\sigma, c)$ and support $S$. It was shown in [19] that the 1-morphism spaces in $2\mathrm{Rep}(\mathcal{G})$ can then be described as follows:

**Theorem C.1.** Let $(\sigma, c, \chi)$ and $(\sigma', c', \chi')$ be two 2-representations of $\mathcal{G}$. Then, their 1-morphism space in $2\mathrm{Rep}(\mathcal{G})$ is given by the category of graded projective representations of $G$ with obstruction pair $(\sigma \otimes \sigma', \overline{c} \otimes c')$ and support $S(\overline{\chi} \otimes \chi')$, i.e.

$$\mathrm{Hom}\big( (\sigma, c, \chi), (\sigma', c', \chi') \big) \; \cong \; \mathrm{Rep}_{S(\overline{\chi} \otimes \chi')}^{(\sigma \otimes \sigma', \overline{c} \otimes c')}(G), \qquad (C.2)$$

where for any collection $\psi \in (A^\vee)^n$ of characters of $A$ we set $S(\psi) := \{i \in \langle n \rangle \mid \psi_i = 1\}$.

**Example C.2.** For each $n$-dimensional 2-representation $(\sigma, c, \chi)$ of $\mathcal{G}$, there exists an identity 1-endomorphism $\mathrm{Id}_{(\sigma, c, \chi)} \in \mathrm{End}(\sigma, c, \chi)$, which in the sense of Theorem C.1 is given by the $n^2$-graded projective representation of $G$ whose only non-vanishing fibres are given by $\mathbb{C}$ on the diagonal elements in $\langle n \rangle \times \langle n \rangle$.

We can visualize 1-morphisms pictorially by representing the 2-representations $(\sigma, c, \chi)$ and $(\sigma', c', \chi')$ by two-dimensional surfaces, and a 1-morphism $(\mathcal{V}, \Phi)$ between them by a one-dimensional line joining up the corresponding surfaces. This is shown in Figure 30.

A special class of 2-representations of $\mathcal{G}$ is given by the simple 2-representations. Recall from Proposition A.39 that every simple 2-representation is induced by a 1-dimensional 2-representations $(u, \alpha)$ of a sub-2-group $H \subset \mathcal{G}$, which is unique up to conjugation by group

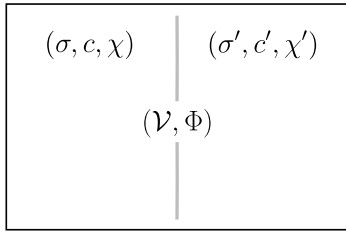

**Figure 30**.

elements $g \in G$. In the following, we will thus label simple 2-representations of $\mathcal{G}$ by triples $(H, u, \alpha)$ – thought of as 2-representations via the correspondence

$$(H, u, \alpha) \quad \longleftrightarrow \quad \mathrm{Ind}_H^{\mathcal{G}}(u, \alpha) \,. \tag{C.3}$$

A natural question to ask is whether the 1-morphism space between two simple 2-representations labelled by triples $(H, u, \alpha)$ and $(K, v, \beta)$ has an analogous simpler description in terms of the data $(H, u, \alpha)$ and $(K, v, \beta)$. This is answered as follows:

**Proposition C.3.** Consider two simple 2-representations of $\mathcal{G}$ labelled by triples $(H, u, \alpha)$ and $(K, v, \beta)$. Then, their 1-morphism space is equivalent to the $(H\backslash G/K)$-graded category

$$\mathrm{Hom}\big((H, u, \alpha),\, (K, v, \beta)\big) \;\cong\; \bigoplus_{\substack{[g]\, \in\, H\backslash G/K: \\ \overline{\alpha}\,\otimes\,^g\beta\, =\, 1}} \mathrm{Rep}^{\overline{u}\,\otimes\,^g v}(H \cap {}^g K)\,, \tag{C.4}$$

where $g \in G$ labels (arbitrary) representatives of double $H$-$K$-cosets in $G$.

*Proof.* Recall that, by definition, the induced simple 2-representations

$$(\sigma, c, \chi) \;:=\; \mathrm{Ind}_H^{\mathcal{G}}(u, \alpha) \tag{C.5}$$

$$(\sigma', c', \chi') \;:=\; \mathrm{Ind}_K^{\mathcal{G}}(v, \beta) \tag{C.6}$$

of $\mathcal{G}$ can be constructed by considering the left-coset spaces

$$G/H \;=\; \{[R_1]\,,...,\,[R_n]\} \tag{C.7}$$

$$G/K \;=\; \{[S_1]\,,...,\,[S_m]\} \tag{C.8}$$

with fixed representatives $R_i, S_j \in G$ (such that $[R_1] = H$ and $[S_1] = K$) and defining the permutation representations $\sigma : G \to S_n$ and $\sigma' : G \to S_m$ by

$$g \cdot R_i \;\overset{!}{=}\; R_{\sigma_g(i)} \cdot h_i(g)\,, \tag{C.9}$$

$$g \cdot S_j \;\overset{!}{=}\; S_{\sigma'_g(j)} \cdot k_j(g)\,, \tag{C.10}$$

where $h_i(g) \in H$ and $k_j(g) \in K$. The 2-cocycles $c \in Z_\sigma^2(G, U(1)^n)$ and $c' \in Z_{\sigma'}^2(G, U(1)^m)$ are then given by

$$c_i(g, g') \;:=\; u\big(h_{\sigma_g^{-1}(i)}(g)\,,\, h_{\sigma_{g\cdot g'}^{-1}(i)}(g')\big)\,, \tag{C.11}$$

$$c'_j(g, g') \;:=\; v\big(k_{\sigma'^{-1}_g(j)}(g)\,,\, k_{\sigma'^{-1}_{g\cdot g'}(j)}(g')\big)\,, \tag{C.12}$$

whereas the collections of characters $\chi \in (A^\vee)^n$ and $\chi' \in (A^\vee)^m$ are defined as

$$\chi_i(a) \; := \; \alpha(R_i^{-1} \triangleright_\rho a) \,, \tag{C.13}$$

$$\chi_j'(a) \; := \; \beta(S_j^{-1} \triangleright_\rho a) \,. \tag{C.14}$$

Let $(\mathcal{V}, \Phi)$ now be a 1-morphsim between $(\sigma, c, \chi)$ and $(\sigma', c', \chi')$, which according to Theorem C.1 is a $(n \cdot m)$-graded projective representation of $G$ with obstruction pair $(\sigma \otimes \sigma', \overline{c} \otimes c')$ and support $S(\overline{\chi} \otimes \chi')$. We decompose

$$\langle n \rangle \times \langle m \rangle \; = \; \bigsqcup_{l=1}^{p} O(i_l, j_l) \tag{C.15}$$

into orbits $O(i_l, j_l) \equiv \{\, (\sigma \otimes \sigma')_g(i_l, j_l) \mid g \in G \,\}$ of the $G$-action $\sigma \otimes \sigma'$ on $\langle n \rangle \times \langle m \rangle$ with fixed representatives $(i_l, j_l)$. According to Lemma B.18, $(\mathcal{V}, \Phi)$ then decomposes as

$$(\mathcal{V}, \Phi) \; \cong \; \bigoplus_{l=1}^{p} (\mathcal{V}_l, \Phi_l) \,, \tag{C.16}$$

where the $(\mathcal{V}_l, \Phi_l)$ are simple graded projective representations. According to Proposition B.19, they are induced by ordinary projective representations

$$(V_l, \varphi_l) \; := \; (\mathcal{V}_{(i_l, j_l)}, \, \Phi|_{(i_l, j_l)}) \tag{C.17}$$

of subgroups $L_l \subset G$ given by

$$L_l \; \equiv \; \mathrm{Stab}_{\sigma \otimes \sigma'}(i_l, j_l) \; = \; R_{i_l} \cdot (H \cap {}^{g_l}K) \cdot R_{i_l}^{-1} \,, \tag{C.18}$$

where we defined $g_l := R_{i_l}^{-1} \cdot S_{j_l} \in G$ for each $l \in \langle p \rangle$. One can check that the corresponding 2-cocycles $w_l$ of $(V_l, \varphi_l)$ are given by

$$w_l \; \equiv \; (\overline{c} \otimes c')_{(i_l, j_l)}|_{L_l} \; = \; {}^{R_{i_l}}(\overline{u} \otimes {}^{g_l}v) \,. \tag{C.19}$$

Using Lemma B.12 as well as the fact that the map $\langle p \rangle \to H \backslash G / K$ sending $l \mapsto [g_l]$ is a bijection, we thus see that the $(V_l, \varphi_l)$ are equivalent to a $(H \backslash G / K)$-graded family of ordinary projective representations of subgroups $H \cap {}^{g_l}K$ of 2-cocycle $\overline{u} \otimes {}^{g_l}v$. Furthermore, since the support of $(\mathcal{V}, \Phi)$ is $S(\overline{\chi} \otimes \chi')$, we know that $(V_l, \varphi_l)$ can only be (potentially) non-zero when

$$(\overline{\chi} \otimes \chi')_{(i_l, j_l)} \; \equiv \; {}^{R_{i_l}}(\overline{\alpha} \otimes {}^{g_l}\beta) \; \overset{!}{=} \; 1 \,. \tag{C.20}$$

Conversely, given a $(H \backslash G / K)$-graded family $(V_l, \varphi_l)$ of ordinary projective representations of $H \cap {}^{g_l}K \subset G$ with 2-cocycles $\overline{u} \otimes {}^{g_l}v$, one can check that

$$(\mathcal{V}, \Phi) \; := \; \bigoplus_{l=1}^{p} \mathrm{Ind}_{H \cap {}^{g_l}K}^{G}(V_l, \varphi_l) \tag{C.21}$$

is isomorphic to a 1-morphism between $(\sigma, c, \chi)$ and $(\sigma', c', \chi')$. $\qquad \square$

**Remark C.4.** Note that, on the right-hand side of (C.4), we regard two projective representations as equivalent if they are related by conjugation. Then, up to equivalence, the choice of representatives $g \in G$ of double $H$-$K$-cosets $[g] \in H\backslash G/K$ in (C.4) does not matter, since choosing different representatives $g' = h \cdot g \cdot k$ for some $h \in H$ and $k \in K$ leads to

$$\mathrm{Rep}^{\overline{u} \otimes ({}^{g'}v)}(H \cap {}^{g'}K) \; \cong \; {}^{h}\big(\mathrm{Rep}^{\overline{u} \otimes ({}^{g}v)}(H \cap {}^{g}K)\big). \tag{C.22}$$

As a useful by-product of Proposition C.3, we learn that the connected components of the 2-category $2\mathrm{Rep}(\mathcal{G})$ are labelled by $G$-orbits in $A^\vee$. Indeed, given two simple 2-representations of $\mathcal{G}$ labelled by $(H, u, \alpha)$ and $(K, v, \beta)$, formula (C.4) tells us that their 1-morphism space is non-vanishing if and only if there exists a $g \in G$ such that $\alpha = {}^{g}\beta$. The latter is equivalent to saying that the characters $\alpha$ and $\beta$ are in the same $G$-orbit inside $A^\vee$.

Proposition C.3 also tells us that we can think of a 1-morphism between two simple 2-representations of $\mathcal{G}$ labelled by triples $(H, u, \alpha)$ and $(K, v, \beta)$ as a collection

$$\varphi \; = \; \{\varphi_g\} \tag{C.23}$$

of ordinary projective representations $\varphi_g : H \cap {}^{g}K \to \mathrm{GL}(V_g)$ indexed by (representatives of) double $H$-$K$-cosets $[g] \in H\backslash G/K$. Pictorially, we can again visualize this by representing $(H, u, \alpha)$ and $(K, v, \beta)$ by two-dimensional surfaces that are joined up by a one-dimensional line representing $\varphi$, as shown in Figure 31.

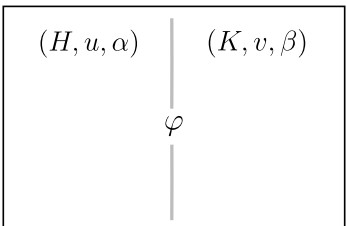

**Figure 31**.

The representation of 1-morphisms between simple 2-representations as in (C.23) furthermore allows us to associate an element $F(\varphi) \in \mathbb{Z}[H\backslash G/K]$ in the free abelian group generated by double $H$-$K$-cosets to each such 1-morphism $\varphi$ by setting

$$F(\varphi) \; := \; \sum_{[g] \in H\backslash G/K} \dim(\varphi_g) \cdot [g]. \tag{C.24}$$

In the following, we will call $F(\varphi)$ the *character* of the 1-morphism $\varphi$.

**Example C.5.** If we denote by $e \in G$ the neutral element of $G$, then the identity 1-endomorphism $\mathrm{Id}_{(H,u,\alpha)}$ of a simple 2-representation labelled by $(H, u, \alpha)$ can be seen as the family of projective representations indexed by (representatives of) double $H$-cosets whose only non-vanishing component is

$$\big(\mathrm{Id}_{(H,u,\alpha)}\big)_e \equiv 1 : \; H \; \to \; \mathbb{C}. \tag{C.25}$$

Consequently, its character is given by $F(\mathrm{Id}_{(H,u,\alpha)}) = 1 \cdot [e] \equiv H \in \mathbb{Z}[G//H]$.

**Example C.6.** In the special case where $G$ is abelian, the endomorphism category of a simple 2-representation $(H, u, \alpha)$ of $\mathcal{G}$ simplifies to

$$\text{End}(H, u, \alpha) \; \cong \; \bigoplus_{\substack{[g] \in G/H: \\ \overline{\alpha} \otimes {}^g\alpha = 1}} \text{Rep}(H) \; =: \; \text{Rep}(H)_{(G/H)_\alpha}, \tag{C.26}$$

where we denoted by $(G/H)_\alpha$ the subgroup of $G/H$ consisting of left $H$-cosets $[g] \in G/H$ for which ${}^g\alpha = \alpha$. The notation $\text{Rep}(H)_{(G/H)_\alpha}$ will be justified through the additional fusion structure on $\text{End}(H, u, \alpha)$ coming from composition, as we will describe later.

## C.2   Fusion

We know from Definition A.28 that there exists a well-defined notion of the tensor product of two 2-representations of $\mathcal{G}$. Similarly, we can use the tensor product for graded projective representations from Definition B.6 to obtain a well-defined tensor product operation on 1-morphisms between 2-representations of $\mathcal{G}$:

$$\text{Hom}\big((\sigma_1, c_1, \chi_1), (\sigma_2, c_2, \chi_2)\big) \; \times \; \text{Hom}\big((\sigma_3, c_3, \chi_3), (\sigma_4, c_4, \chi_4)\big)$$
$$\Big\downarrow \otimes \tag{C.27}$$
$$\text{Hom}\big((\sigma_1, c_1, \chi_1) \otimes (\sigma_3, c_3, \chi_3), (\sigma_2, c_2, \chi_2) \otimes (\sigma_4, c_4, \chi_4)\big)$$

Pictorially, the tensor product of 1-morphisms can be visualized as the fusion of two parallel surfaces, each of which consists of two 2-representations joined up by a 1-morphism. This is shown in Figure 32.

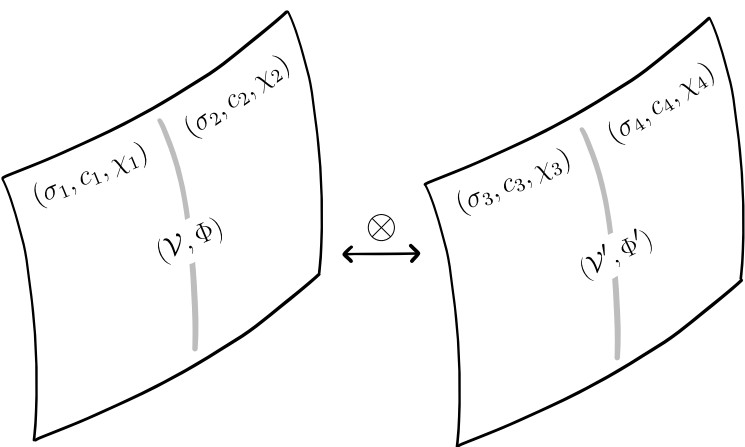

**Figure 32**.

A natural question to ask is how the tensor product acts on 1-morphisms between simple 2-representations labelled by triples $(H, u, \alpha)$ and $(K, v, \beta)$. To answer this question,

we note that the trivial 1-dimensional 2-representation $\mathbf{1}$ of $\mathcal{G}$ gives rise to a map

$$
\begin{array}{c}
\mathrm{Hom}\big((H,u,\alpha),(K,v,\beta)\big) \; \times \; \mathrm{End}(\mathbf{1}) \\[4pt]
\Big\downarrow \otimes \\[4pt]
\mathrm{Hom}\big((H,u,\alpha),(K,v,\beta)\big),
\end{array}
\qquad\text{(C.28)}
$$

where we used that $(\sigma,c,\chi)\otimes\mathbf{1}\cong(\sigma,c,\chi)$ for any 2-representation $(\sigma,c,\chi)$ of $\mathcal{G}$. According to Theorem C.1, the 1-endomorphism category of $\mathbf{1}$ is simply given by

$$
\mathrm{End}(\mathbf{1}) \;\cong\; \mathrm{Rep}(G),
\qquad\text{(C.29)}
$$

so that the objects of $\mathrm{End}(\mathbf{1})$ are ordinary representations $\psi : G \to \mathrm{GL}(W)$ of $G$ on a vector space $W$. The tensor product operation in (C.28) can then be described as follows:

**Proposition C.7.** Let $\varphi$ be a 1-morphism between two simple 2-representations $(H,u,\alpha)$ and $(K,v,\beta)$ and let $\psi : G \to \mathrm{GL}(W)$ be a 1-endomorphism of the trivial 2-representation $\mathbf{1}$. Then, their tensor product is given by the 1-morphism

$$
(\varphi\otimes\psi)_g \;\cong\; \varphi_g \,\otimes\, \mathrm{Res}^{G}_{H\cap{}^{g}K}(\psi),
\qquad\text{(C.30)}
$$

where $g \in G$ labels (arbitrary) representatives of double $H$-$K$-cosets $[g]\in H\backslash G/K$.

*Proof.* Recall that the 1-morphism $\varphi$ between $(H,u,\alpha)$ and $(K,v,\beta)$ can be thought of as a family $\{\varphi_g\}$ of ordinary projetive representations $\varphi_g : H\cap{}^{g}K \to \mathrm{GL}(V_g)$ indexed by double $H$-$K$-cosets $[g]\in H\backslash G/K$. We can construct a corresponding graded projective representation $(\mathcal{V},\Phi)$ of $G$ out of $\varphi$ by setting

$$
(\mathcal{V},\Phi) \;:=\; \bigoplus_{[g]\in H\backslash G/K} \mathrm{Ind}^{G}_{H\cap{}^{g}K}(V_g,\varphi_g).
\qquad\text{(C.31)}
$$

Similarly, we can regard the ordinary representation $\psi : G \to \mathrm{GL}(W)$ as the 1-graded projective representation $\mathrm{Ind}^{G}_{G}(W,\psi)$ induced from $G$ to itself. Then, using Proposition B.21, the tensor product of $(\mathcal{V},\Phi)$ and $(W,\psi)$ can be computed to be

$$
\begin{aligned}
(\mathcal{V},\Phi)\otimes(W,\psi) &\cong \bigoplus_{[g]\in H\backslash G/K} \mathrm{Ind}^{G}_{H\cap{}^{g}K\cap G}\big((V_g,\varphi_g)\otimes(W,\psi)\big) \\
&\cong \bigoplus_{[g]\in H\backslash G/K} \mathrm{Ind}^{G}_{H\cap{}^{g}K}\big((V_g,\varphi_g)\otimes\mathrm{Res}^{G}_{H\cap{}^{g}K}(W,\psi)\big).
\end{aligned}
\qquad\text{(C.32)}
$$

Thus, we again obtain a family of ordinary projective representations

$$
\varphi_g \,\otimes\, \mathrm{Res}^{G}_{H\cap{}^{g}K}(\psi) : \; H\cap{}^{g}K \;\to\; \mathrm{GL}(V_g\otimes W)
\qquad\text{(C.33)}
$$

indexed by double $H$-$K$-cosets $[g]\in H\backslash G/K$, which coincides with the family of projective representations given in (C.30). $\qquad\square$

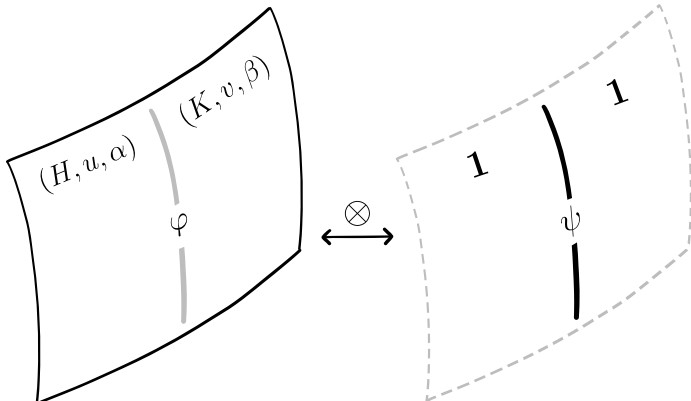

**Figure 33**.

Pictorially, the tensor product of a 1-morphism between simple 2-representations and a 1-endomorphism of **1** as in (C.28) can be visualized as shown in Figure 33.

In the special case where $(H, u, \alpha) = (K, v, \beta)$, we can choose $\varphi$ to be the identity morphism $\mathrm{Id}_{(H,u,\alpha)}$, which simplifies the tensor product operation in (C.28) as follows:

**Corollary C.8.** Let $\mathrm{Id}_{(H,u,\alpha)}$ be the identity 1-morphism of a 2-representation $(H, u, \alpha)$ and let $\psi : G \to \mathrm{GL}(W)$ be a 1-endomorphism of the trivial 2-representation **1**. Then, their tensor product is the 1-morphism whose only non-vanishing component is given by

$$\left(\mathrm{Id}_{(H,u,\alpha)} \otimes \psi\right)_e \;\cong\; \mathrm{Res}^G_H(\psi) . \tag{C.34}$$

## C.3 Composition

Using the notion of composition of graded projective representations from Definition B.24, we can introduce the composition of 1-morphisms between three 2-representations $(\sigma, c, \chi)$, $(\sigma', c', \chi')$ and $(\sigma'', c'', \chi'')$ of $\mathcal{G}$ as a map

$$\mathrm{Hom}\big((\sigma, c, \chi), (\sigma', c', \chi')\big) \;\times\; \mathrm{Hom}\big((\sigma', c', \chi'), (\sigma'', c'', \chi'')\big)$$
$$\Big\downarrow \circ \tag{C.35}$$
$$\mathrm{Hom}\big((\sigma, c, \chi), (\sigma'', c'', \chi'')\big) .$$

Pictorially, the composition of two 1-morphisms $(\mathcal{V}, \Phi)$ and $(\mathcal{V}', \Phi')$ can be visualized as the collision of two parallel lines joining up three surfaces labelled by the corresponding 2-representations. This is shown in Figure 34.

A natural question to ask is how composition acts on 1-morphisms between simple 2-representations labelled by $(H, u, \alpha)$, $(K, v, \beta)$ and $(L, w, \gamma)$, giving rise to a map

$$\mathrm{Hom}\big((H, u, \alpha), (K, v, \beta)\big) \;\times\; \mathrm{Hom}\big((K, v, \beta), (L, w, \gamma)\big)$$
$$\Big\downarrow \circ \tag{C.36}$$
$$\mathrm{Hom}\big((H, u, \alpha), (L, w, \gamma)\big) .$$

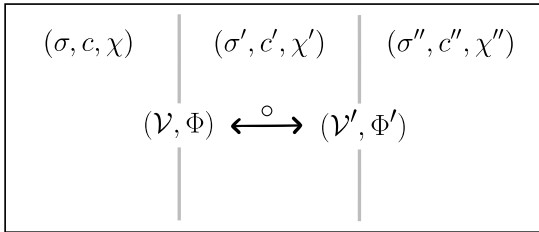

**Figure 34**.

We conjecture this question to be answered as follows:

**Proposition C.9.** Let $\varphi$ be a 1-morphism between $(H, u, \alpha)$ and $(K, v, \beta)$ and let $\varphi'$ be a 1-morphism between $(K, v, \beta)$ and $(L, w, \gamma)$. Then, their composition $\varphi \circ \varphi'$ is the 1-morphism between $(H, u, \alpha)$ and $(L, w, \gamma)$ whose components are given by

$$(\varphi \circ \varphi')_g \;\cong\; \bigoplus_{[\bar{g}] \,\in\, (H \cap {}^gL)\backslash G/K} \mathrm{Ind}_{H \cap {}^{\bar{g}}K \cap {}^gL}^{H \cap {}^gL} \big[\, \varphi_{\bar{g}} \otimes {}^{\bar{g}}\big(\varphi'_{\bar{g}^{-1}g}\big) \,\big]\,, \tag{C.37}$$

where $g \in G$ labels (arbitrary) representatives of double $H$-$L$-cosets $[g] \in H\backslash G/L$.

Pictorially, the composition of 1-morphisms between simple 2-representations as in (C.36) can be visualized as shown in Figure 35.

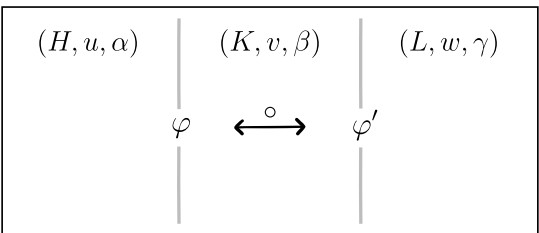

**Figure 35**.

**Remark C.10.** If we denote by $F(\varphi) \in \mathbb{Z}[H\backslash G/K]$ and $F(\varphi') \in \mathbb{Z}[K\backslash G/L]$ the characters of $\varphi$ and $\varphi'$ as in (C.24), one can check that the character of their composition is given by

$$F(\varphi \circ \varphi') \;=\; F(\varphi) * F(\varphi')\,, \tag{C.38}$$

where $*$ denotes the convolution product

$$*: \;\; \mathbb{Z}[H\backslash G/K] \,\times\, \mathbb{Z}[K\backslash G/L] \;\to\; \mathbb{Z}[H\backslash G/L]\,. \tag{C.39}$$

**Example C.11.** In the special case where $G$ is abelian, we know from Example C.6 that the 1-endomorphism category of a simple 2-representation $(H, u, \alpha)$ of $\mathcal{G}$ is given by $\mathrm{Rep}(H)_{(G/H)_\alpha}$. The notation of the latter is justified by the fact that, according to Proposition C.9, the composition of two 1-endomorphisms $\varphi$ and $\varphi'$ of $(H, u, \alpha)$ is given by

$$(\varphi \circ \varphi')_g \;=\; \bigoplus_{[g_1] \cdot [g_2] \,=\, [g]} \varphi_{g_1} \otimes \varphi'_{g_2}\,. \tag{C.40}$$

## C.4 Examples

Let us conclude by considering examples of the 2-category $2\text{Rep}(\mathcal{G})$ for different split 2-groups $\mathcal{G} = (G, A, \rho)$. For simplicity, we will only consider such $\mathcal{G}$ for which $G$ is abelian. Then, according to Remark A.41, the simple $n$-dimensional 2-representations of $\mathcal{G}$ are classified by triples $(H, u, \alpha)$, where

- $H \subset G$ is a subgroup of $G$ of index $|G : H| = n$,

- $u \in H^2(H, U(1))$ is a degree-2 cohomology class on $H$ with coefficients in $U(1)$,

- $\alpha \in A^\vee$ represents an equivalence class of $H$-invariant characters on $A$, where two such characters $\alpha_1$ and $\alpha_2$ are equivalent if there exists a $g \in G$ such that $\alpha_2 = {}^g\alpha_1$.

According to Proposition A.42, the tensor product of two such triples can be computed via

$$(H, u, \alpha) \otimes (K, v, \beta) \cong \bigoplus_{[g] \in H \backslash G / K} (H \cap K, \, u \otimes v, \, \alpha \otimes {}^g\beta), \tag{C.41}$$

whereas according to Proposition C.3 their 1-morphism category is given by

$$\text{Hom}\big((H, u, \alpha), (K, v, \beta)\big) \cong \bigoplus_{\substack{[g] \in H \backslash G / K: \\ \bar{\alpha} \otimes {}^g\beta = 1}} \text{Rep}^{\bar{u} \otimes v}(H \cap K). \tag{C.42}$$

Furthermore, according to Example C.6, the composition of 1-morphisms endows the category of 1-endomorphisms of any simple 2-representation $(H, u, \alpha)$ with the structure

$$\text{End}(H, u, \alpha) \cong \text{Rep}(H)_{(G/H)_\alpha}. \tag{C.43}$$

We will use the notation $\mathbf{n}$ for a $n$-dimensional simple 2-representation of $\mathcal{G}$ in what follows. In order to describe the 1-morphism categories between simple 2-representations $\mathbf{n}$ and $\mathbf{m}$, we use the diagrammatic notation

$$
\mathbf{n} \underset{\text{Hom}(\mathbf{m}, \mathbf{n})}{\overset{\text{Hom}(\mathbf{n}, \mathbf{m})}{\rightleftarrows}} \mathbf{m}
\tag{C.44}
$$

### C.4.1 $2\text{Rep}(\mathbb{Z}_2)$

Consider the 2-group $\mathcal{G} = (\mathbb{Z}_2, 1, 1)$. We denote the elements of the cyclic group $\mathbb{Z}_2$ by

$$\mathbb{Z}_2 = \{1, x\}. \tag{C.45}$$

Since $H^2(\mathbb{Z}_2, U(1)) = 1$, the simple 2-representations of $\mathcal{G}$ are completely determined by the choice of subgroup $H \subset \mathbb{Z}_2$, leaving us with

| | $H$ |
|---|---|
| $\mathbf{1}$ | $\mathbb{Z}_2$ |
| $\mathbf{2}$ | $\{1\}$ |

$$\tag{C.46}$$

Using (C.41), their fusion structure can be computed to be

$$
\begin{array}{c|cc}
\otimes & \mathbf{1} & \mathbf{2} \\
\hline
\mathbf{1} & \mathbf{1} & \mathbf{2} \\
\mathbf{2} & \mathbf{2} & \mathbf{2} \oplus \mathbf{2}
\end{array}
\tag{C.47}
$$

Furthermore, using (C.42), their 1-morphism categories can be described by the diagram

$$
\tag{C.48}
$$

### C.4.2   $2\mathbf{Rep}(\mathcal{D}_8)$

Consider the 2-group $\mathcal{G} = (\mathbb{Z}_2, \mathbb{Z}_2 \times \mathbb{Z}_2, \rho)$, where $\mathbb{Z}_2 = \{1, x\}$ acts on $\mathbb{Z}_2 \times \mathbb{Z}_2$ via

$$
x \triangleright_\rho (a, b) := (b, a). \tag{C.49}
$$

We denote the elements of the Pontryagin dual of $\mathbb{Z}_2$ by

$$
\mathbb{Z}_2^\vee =: \{1, \lambda\}, \tag{C.50}
$$

where the non-trivial character $\lambda$ is defined by $\lambda(x) = -1$. Since $H^2(\mathbb{Z}_2, U(1)) = 1$, the simple 2-representations of $\mathcal{G}$ are completely determined by the choices of subgroup $H \subset \mathbb{Z}_2$ and $H$-invariant character $\alpha \in \mathbb{Z}_2^\vee \times \mathbb{Z}_2^\vee$, leaving us with[12]

$$
\begin{array}{c|cc}
 & H & \alpha \\
\hline
\mathbf{1}_+ & \mathbb{Z}_2 & (1, 1) \\
\mathbf{1}_- & \mathbb{Z}_2 & (\lambda, \lambda) \\
\mathbf{2}_+ & \{1\} & (1, 1) \\
\mathbf{2}_0 & \{1\} & (\lambda, 1) \\
\mathbf{2}_- & \{1\} & (\lambda, \lambda)
\end{array}
\tag{C.51}
$$

---

[12]Note that $(1, \lambda) = {}^x(\lambda, 1)$, which gets rid of the additional 2-dimensional simple 2-representation of $\mathcal{G}$ we could have written down naively.

Using (C.41), their fusion structure can be computed to be

| $\otimes$ | $\mathbf{1}_+$ | $\mathbf{1}_-$ | $\mathbf{2}_+$ | $\mathbf{2}_0$ | $\mathbf{2}_-$ |
|---|---|---|---|---|---|
| $\mathbf{1}_+$ | $\mathbf{1}_+$ | $\mathbf{1}_-$ | $\mathbf{2}_+$ | $\mathbf{2}_0$ | $\mathbf{2}_-$ |
| $\mathbf{1}_-$ | $\mathbf{1}_-$ | $\mathbf{1}_+$ | $\mathbf{2}_-$ | $\mathbf{2}_0$ | $\mathbf{2}_+$ |
| $\mathbf{2}_+$ | $\mathbf{2}_+$ | $\mathbf{2}_-$ | $\mathbf{2}_+ \oplus \mathbf{2}_+$ | $\mathbf{2}_0 \oplus \mathbf{2}_0$ | $\mathbf{2}_- \oplus \mathbf{2}_-$ |
| $\mathbf{2}_0$ | $\mathbf{2}_0$ | $\mathbf{2}_0$ | $\mathbf{2}_0 \oplus \mathbf{2}_0$ | $\mathbf{2}_+ \oplus \mathbf{2}_-$ | $\mathbf{2}_0 \oplus \mathbf{2}_0$ |
| $\mathbf{2}_-$ | $\mathbf{2}_-$ | $\mathbf{2}_+$ | $\mathbf{2}_- \oplus \mathbf{2}_-$ | $\mathbf{2}_0 \oplus \mathbf{2}_0$ | $\mathbf{2}_+ \oplus \mathbf{2}_+$ |

$$\text{(C.52)}$$

Furthermore, using (C.42), their 1-morphism categories can be described by the diagram

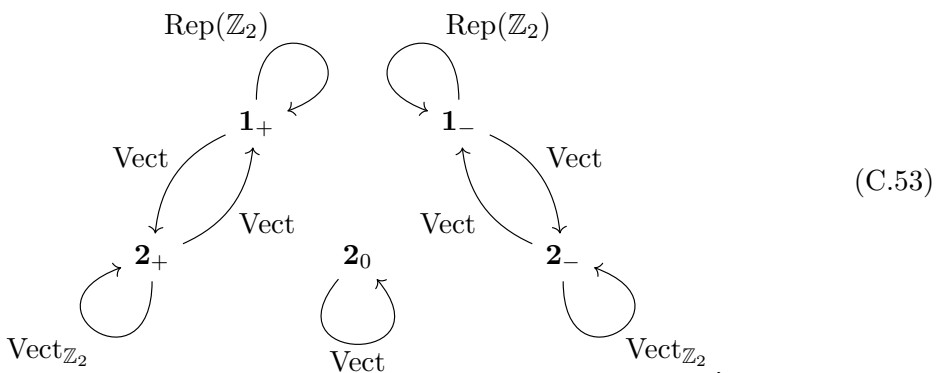

$$\text{(C.53)}$$

As expected, there are three connected components, corresponding to the three $\mathbb{Z}_2$-orbits inside $(\mathbb{Z}_2 \times \mathbb{Z}_2)^\vee$.

### C.4.3  $\mathbf{2Rep}(\mathbb{Z}_2 \times \mathbb{Z}_2)$

Consider the 2-group $\mathcal{G} = (\mathbb{Z}_2 \times \mathbb{Z}_2, 1, 1)$, where we again denote the elements of $\mathbb{Z}_2$ by $\mathbb{Z}_2 = \{1, x\}$. Using that $H^2(\mathbb{Z}_2 \times \mathbb{Z}_2, U(1)) = \mathbb{Z}_2$ and $H^2(\mathbb{Z}_2, U(1)) = 1$, the simple 2-representations of $\mathcal{G}$ are completely determined by the choices of subgroup $H \subset \mathbb{Z}_2 \times \mathbb{Z}_2$

and a corresponding cohomology class $u \in H^2(H, U(1))$, leaving us with

$$
\begin{array}{c|cc}
 & H & u \\
\hline
\mathbf{1}^\alpha & \mathbb{Z}_2 \times \mathbb{Z}_2 & \alpha \in \mathbb{Z}_2 \\
\mathbf{2}_L & \mathbb{Z}_2^L & 1 \\
\mathbf{2}_D & \mathbb{Z}_2^D & 1 \\
\mathbf{2}_R & \mathbb{Z}_2^R & 1 \\
\mathbf{4} & \{1\} & 1
\end{array}
\qquad (C.54)
$$

where we denoted the non-trivial subgroups of $\mathbb{Z}_2 \times \mathbb{Z}_2$ by

$$
\begin{aligned}
\mathbb{Z}_2^L &:= \{(1,1), (x,1)\}, \\
\mathbb{Z}_2^D &:= \{(1,1), (x,x)\}, \\
\mathbb{Z}_2^R &:= \{(1,1), (1,x)\}.
\end{aligned}
\qquad (C.55)
$$

Using (C.41), their fusion structure can be computed to be

$$
\begin{array}{c|ccccc}
\otimes & \mathbf{1}^\alpha & \mathbf{2}_L & \mathbf{2}_D & \mathbf{2}_R & \mathbf{4} \\
\hline
\mathbf{1}^\beta & \mathbf{1}^{\alpha+\beta} & \mathbf{2}_L & \mathbf{2}_D & \mathbf{2}_R & \mathbf{4} \\
\mathbf{2}_L & \mathbf{2}_L & \mathbf{2}_L \oplus \mathbf{2}_L & \mathbf{4} & \mathbf{4} & \mathbf{4} \oplus \mathbf{4} \\
\mathbf{2}_D & \mathbf{2}_D & \mathbf{4} & \mathbf{2}_D \oplus \mathbf{2}_D & \mathbf{4} & \mathbf{4} \oplus \mathbf{4} \\
\mathbf{2}_R & \mathbf{2}_R & \mathbf{4} & \mathbf{4} & \mathbf{2}_R \oplus \mathbf{2}_R & \mathbf{4} \oplus \mathbf{4} \\
\mathbf{4} & \mathbf{4} & \mathbf{4} \oplus \mathbf{4} & \mathbf{4} \oplus \mathbf{4} & \mathbf{4} \oplus \mathbf{4} & \mathbf{4} \oplus \mathbf{4} \oplus \mathbf{4} \oplus \mathbf{4}
\end{array}
\qquad (C.56)
$$

Furthermore, using (C.42), their 1-morphism categories can be described by the diagram

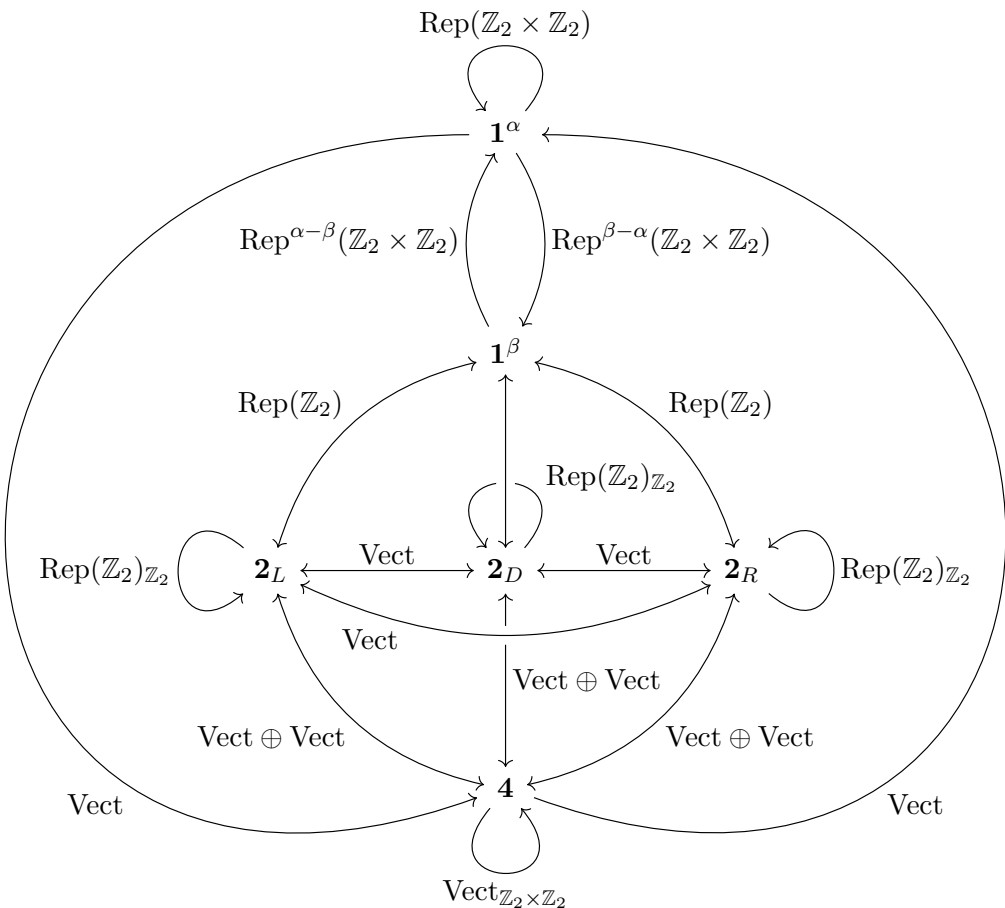

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
