# Peer review of "Non-invertible Symmetries and Higher Representation Theory I"

_SciPost Physics_

## Round 2 · Referee Report · Anonymous (Referee 1) · 2023-11-30

Strengths

1) The main finding of the paper is interesting because it generalizes a known fact from 2d to the 3d context in a nice and concise manner. The identification of the full dual symmetry in 3d as 2 Rep(G) was not explicitly stated before and is an important result.

2) The paper connects known mathematical facts (such as the classification of 2Reps) with physical expectations in a cohesive manner, providing convincing evidence for its claims.

3) The paper is pedagogical and detailed in nature. Hence, it may serve as a useful reference, especially for facts that were already known but not so clearly explained in other references. An example is the derivation the authors use to demonstrate that the category after gauging G is Rep(G) in 2D.

Weaknesses

1) The paper is quite lengthy and repeats the same procedure multiple times, increasing in complexity. Some sections delve into pedantic details, causing the main idea to become obscured within highly technical content. It would be beneficial to refine certain parts for better clarity.

2) While the paper maintains mathematical precision, the authors have not attempted to provide a physical motivation for the relevance of their results. This in itself is not a drawback, as many within the generalized symmetries field appreciate nearly pure mathematical results. However, readers approaching the paper from a more physical perspective might struggle to find strong motivation to delve into its detailed content.

3) Although the introduction summarizes the main findings, most of the notations used are not explained there, rendering it somewhat incomprehensible before delving into the main body of the paper. For instance, the notation U(1)^O is introduced for the first time on page 2 but isn't explained until page 25. The authors should undoubtedly address these issues to enhance the introduction's usefulness, especially in the context of such an extensive and technical paper.

Report

The paper explores the structure of the 2-category symmetry resulting from the gauging of finite 0-form symmetry groups in 3D, showing that this category corresponds to 2-Representations of the group. This extends the well-known 2D fact that gauging G leads to the dual symmetry Rep(G). It aligns with physical expectations, considering the emergence of condensation defects beyond topological Wilson lines of G, introduced in [4], and fitting these expectations into a known mathematical structure, which the authors review comprehensively. The paper then generalizes this concept to 2-groups.

Therefore I recommend the paper for the publication, provided that the authors address several points.

Here is a list of comments, questions, typos I noted, and curiosities. The most important issues which should be addressed are then reported in the requested changes.

1) The labeling with G-orbits at pag. 2 would require further explain the notation, and maybe some finer specification. This is more or less addressed later on in sec. 4.2, but It would really help the reader understating the main fining from the introduction. Concretely on “orbit” could refer the the subset of a given set on which G acts in a given way, but this is not what the authors have in mind. It seems they have in mind the family of finite sets endowed with a transitive action of G. More precisely I think they should talk about isomorphism classes of such sets to have a finite number of them. The notation U(1)^O is also not explained here but at pag. 25.

2) At pag. 2, and repeatedly in all the paper the authors claim that both condensation defects and SPT stacked on some submanifold are condensations from a mathematical perspective. This would require an explanation, or at least a reference (side note: the main reference in math that I know for condensations is 1905.09566 which is not even cited in the paper). Then a question: from a physical perspective, shouldn’t SPT phases on submanifold be identified with the identity operator? They don’t act on anything, and moreover the authors seem not to include decoupled TQFTs on submanifold which I would say are on the same footing from the physical viewpoint. If this is not true from a mathematical perspective the authors should explain this difference.

3) Section 2.2 is a review of section 4.5 of [17] with a slightly different derivation. It’s definitely more pedagogical and hence useful, but maybe the authors could improve the usefulness by commenting on the relation between this abstract derivation and the physical intuition that the category after gauging in Rep(G) just because is generated by Wilson lines.

4) A picture of eq. (2.11) and similar after could help the reader.

5) In section 2.3, before considering the general situation of interfaces maybe it’s worth mentioning that in the simplest case the symmetry category is again just Rep(G).

6) Curiosity. The authors point out that the Rep^c(G), the projective representations with fixed projective cocycle don’t have a fusion structure. Given c\in H^2(G,U(1)) however one could construct a central extension \widetilde{G}_c of G by U(1) determined by c, and construct Rep( widetilde{G}_c) which definitely a fusion category and it’s the symmetry arising by gauging the symmetry extension. Is there some relation between these two objects?

7) Typo in fig.9 \hat(T)--- > T in left upper box.

8) It is not clear what is the motivation of the (quite long) sec. 3.3: the already showed the category is Rep(G) so obviously the simple objects will be the same.

9) End of sec. 3: the authors mention TY based on Z_2xZ_2 but they don’t state bicharacter and FS indicator.

10) Eq. 4.3: why don’t the authors allow the coefficients to be partition functions of TQFTs, as is by now established in the physics literature (e.q. in [4]).

11) Can the authors give an argument in favor of their belief that (4.18) exhaust all the conditions?

12) The derivation at pag. 32 is one of the fundamental ones in the paper since it connects the known mathematical classification of 2Reps with the physical expectation that condensates are labeled by a subgroup H (to be gauged from the “full” condensate) and a discrete torsion for it. However many steps are not very clear and there are few points I’m confused about. (I) As I already pointed out in 1) it should be more correct to talk about “isomorphism” classes of orbits. (II) The phrase “conjugacy classes of subgroups” could be ambiguous. One may think that you have to consider H as a group and take its set of conjugacy classes, while the authors means the set of subgroups H\subset G with the identification that H_1 \sim H_2 if they are conjugated. That’s ok but it’s worth explaining the terminology (III) It’s easy to check that to get a 1-1 correspondence G-orbits < ---- > conjugacy classes of subgroups one really have to consider “isomorphism classes of orbits”, supporting my previous point. But then it’s not clear why the authors at the end label simple objects with (H,c) instead of equivalence classes. (IV) However there is a clear physical motivation (that also the authors explain, when they interpret H as the unhiggsed subgroup) why the label should be a subgroup instead of an equivalence class of them. Why is there is this mismatch between what they seems to obtain and what they claim (which is the physical expectation)?

13) Note on the end paragraph of section 4.3.2 : the same picture with the general case of H\neq 1 was also explained in [11].

13) In (4.52) the fusion coefficients are positive integers, while in many references (e.g. [4]) it is pointed out they should be TQFT.

14) As explained in [4] the fusion (4.77) is only correct on T^2, while generically the fusion=2 should be replaced by the partition function of a Z_2 gauge theory.

15) At the end of sec. 4 it would be extremely useful to discuss in some detail also a non-abelian example (where there can be non-normal subgroups) since this could help clarifying the issue of subgroups/conjugacy classes of them in labeling the defect, that I rised in 12) (III-IV).

16) Pag. 40 below fig. 24 typo: Rep(G)-- > Rep(\hat{G})

17) Beginning of sec. 5.1: The presence of condensation defects is known in the math literature as Karoubi completion (see e.g. 1905.09566) and in physics was emphasized in the second version of [5].

Requested changes

1) Better explanation of the notation and some of the notions in the introduction, as suggested in point 1) of the REPORT.

2) Address the issue of TQFT coefficients that the authors don’t account for in their formalism, although these coefficients are known to appear in the fusion of condensation in many other references. If the categories 2Rep don’t include these data, the mismatch between the physical findings and the mathematical results would require at least some comment.

3) Resolve the issue of labeling with subgroups versus conjugacy classes of them, as pointed out in the REPORT.

4) Discuss a non-abelian example (the simplest case could be S_3) at the end of sec. 4 and attempt to use it to clarify the aforementioned issue.

---

## Round 2 · Referee Report · Anonymous (Referee 2) · 2024-3-22

Report

This paper deals with non-invertible symmetries that can be obtained via gauging a more standard group-like symmetry. The main result is a detailed description of the categorical structure of the topological defects that arise from gauging a split 2-group in 3d. In particular, it is argued that the symmetry category of the gauged theory, which is a fusion 2-category, is the category of 2-representations of the 2-group. This is an important result that generalizes to 3d the well-known fact that gauging a symmetry $G$ in 2d results in a dual symmetry Rep$(G)$ in the gauged theory.

Some general comments are the following:

  1. In section 3.2.2 (and in the 3d cases as well) the product structure in the gauged theory $\widehat{\mathcal{T}}/H$ is simply stated to follow from the tensor product of graded vector spaces in $\widehat{\mathcal{T}}$. It is not completely clear, however, and would be interesting to see, if the product structure in the gauged theory can be obtained more explicitly from the bimodule construction as a projection of the product in the original theory $\widehat{\mathcal{T}}$;

  2. Section 4.3 contains important results but it is technical and difficult to read without delving into the appendices. I think providing more details for the induction and restriction of 2-representations in the main text would help the reader;

  3. Connection to condensation defects is only made briefly on top of page 33 for groups in 3d, but it would be beneficial for the physics audience to expand it more in terms of the usual picture of gauging (a subset) of the Rep$(G)$ symmetry lines on a codimension-1 submanifold. For example, the sub-category of morphisms Vect$(G/H)$ on a surface labelled by $H$ has a very clear interpretation in this picture as the dual symmetry lines;

  4. Regarding the general structure: the paper is 100 pages long, and almost half of it consists of appendices. These are quite technical, mathematical and slightly hard to digest for the physically minded reader, so I personally think the paper would benefit from a more pedagogical and streamlined discussion of the more mathematical aspects, as well as a more explicitly stated connection to physics whenever possible.

Some minor typos/comments are the following:

  1. middle of page 3, point 2 in the labelling of simple objects, should be $H^2(H,U(1)^O)$ instead of $H^2(G,U(1)^O)$;

  2. bottom of page 3, should be 'class' instead of 'lass';

  3. figure 9 on page 10, in the left upper box $\mathcal{T}$ should have no hat;

  4. eq. 3.14 on page 12, should be $V_{\chi,e}$ instead of $V_{h,e}$ in the second line;

  5. in section 4.2.1. it is used sometimes $e$, sometimes $1$ for the identity (e.g. eq. 4.13 compared to figure 16);

  6. should specify also in eq. 4.18 that $\rho_g$, $\rho_{h_1}$ etc. denote the identity 2-morphisms on the corresponding 1-morphism (and similarly in other sections);

  7. below 4.22 should be $j=1,\dots,n$ instead of $i$;

  8. eq. 4.25, should it be $\sigma_{g}'(j-n)$ instead of $\sigma_{g}'(n-j)$?

  9. top of page 28, should be $\Phi_{g,\sigma_h(j)}$ instead of $\Phi_{g,j}$;

  10. above eq. 5.1 on page 39, the homomorphism should be $\varphi: H \rightarrow \text{Aut}(A)$;

  11. after equation 7.5 on page 54 there seems to be a missing reference to some figure. The discussion about extension to 2-groups with non-trivial Postnikov class is a bit vague. Is it meant to say gauge $H$ after 7.7, as gauging $\widehat{A}$ would just give back $\mathcal{T}$?

Overall, the paper is well-written and contains original and important results, therefore I recommend the paper for publication provided the above comments are taken into account.

---

## Editorial Decision

resubmitted